# How Classifiers Extract General Features for Downstream Tasks: An Asymptotic Analysis in Two-Layer Models

## Abstract

Neural networks learn effective feature representations through intermediate layers, enabling feature transfer without additional training for new tasks. However, the conditions for successful feature transfer remain underexplored. In this paper, we investigate feature transfer in classifier-trained networks, focusing on clustering in unseen distributions. In binary classification, we find that higher similarity between training and unseen distributions improves Cohesion and Separability, while Separability further requires unseen data to be assigned to different training classes. In multi-class classification, our analysis shows that the feature extractor maps input point based on their similarity to training classes, i.e. that unrelated training classes to input have negligible impact on feature extraction. We validate our theoretical findings in synthetic dataset and demonstrate practical applicability utilizing ResNet and variations of CAR, CUB, SOP, ISC, and ImageNet datasets.

## 1. Introduction

Neural networks have the remarkable ability to adapt to specific tasks, learning representations through penultimate layers. Training these intermediate layers is crucial for neural network generalization (Damian et al., 2022). Also, these layers can extract semantically meaningful and transferable features from new data, enabling feature transfer for new tasks (Yosinski et al., 2014; Kornblith et al., 2019). A wide range of techniques, from open set clustering (Roth et al., 2020; Huang et al., 2024) to vision-language models (Li et al., 2023) and language models (Brown et al., 2020; Kojima et al., 2023), leverage feature transfer for downstream tasks. However, the specific conditions where features can be effectively transferred remain underexplored.

[1] Anonymous Institution, Anonymous City, Anonymous Region, Anonymous Country. Correspondence to: Anonymous Author <anon.email@domain.com>.

Preliminary work. Under review by the International Conference on Machine Learning (ICML). Do not distribute.

Among various applications, classification based visual open-set clustering (Musgrave et al., 2020) serves as a fundamental benchmark for evaluating whether a feature extractor can generalize to unseen data. Typically, this task involves classifier training on one set of classes and then testing it on disjoint classes to assess whether the extracted features form cohesive and separable class-wise clusters on unseen data (Wang et al., 2018; Seidenschwarz et al., 2021; Deng et al., 2022). Given this context, we aim to investigate feature clustering with the following research questions:

> Can we **capture the presences of feature learning in classification** and **identify the conditions where features** cluster effectively on new distributions?

To address this question, we analyze a two-layer nonlinear network network trained with a single large gradient descent step on a mean-squared classification loss in the *proportional regime* (in section 2). The proportional regime intuitively represents a scenario where the network width and the size of the dataset are of similar scales, aligning with common practices in model scaling (Ba et al., 2022), and they are known to effectively capture the phenomena occurring during the actual training process, as demonstrated in studies such as Mei & Montanari (2020); Moniri et al. (2024). We capture that the dominant part of the trained feature is composed of random initialization and *spikes* (Def. 3.4) associated with the training classes (section 3). Leveraging dominant features, we identify conditions for effective clustering on new distributions (section 4).

In a binary classification setting, we assess the intra-class *cohesion* and inter-class *separability* of trained features in a numerical-analytical manner representing the clustering population risks (Def. 4.3) (Clémençcon, 2011; Papa et al., 2015; Li & Liu, 2021) and goals for clustering performance (Liu et al., 2017). As a result, *Cohesion* increases as the *train-unseen similarity* (in Def. 4.1) grows larger. Meanwhile, for *Separability*, if classes classes are *assigned* (Notes 4.2, E.1) to different training classes, *Separability* increases as the *train-unseen similarity* grows larger; otherwise, it decreases, as illustrated in Figure 1.

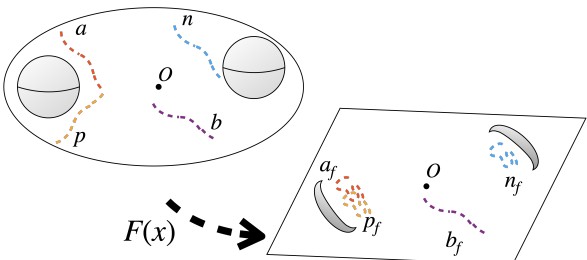

Figure 1: Mapping data from the input space (left) to the learned feature space (right). Training classes are shown as balls, and unseen classes as dashed lines $(a, b, p, n)$. *Cohesion*: Strong *cohesion* occurs for $a, p, n$, which have high similarity to the training classes compared to $b$. *Separability of $a, n$*: $a$ and $n$, *assigned* to different training class, demonstrate high Separability. *Separability of $a, p$*: $a$ and $p$, *assigned* to the same training class, exhibit low Separability.

In the multi-class classification setting, we analyze the *spikes* of features and find that *spikes* map new inputs based on a linear combination of randomly initialized classifier heads' weight with coefficients that represent the similarity of the training classes. Therefore, the more *spikes* aligned with the input data the greater their contribution to feature extraction, enhancing the expressiveness of the features.

In the experiments, we empirically observe *train-unseen similarity*, *cohesion*, *Separability*, and *recall@1* under our theoretical assumptions in synthetic datasets. As a result, we confirm that the theoretical interpretation aligns with the actual findings (subsection 5.2). Additionally, we explore practical metric learning settings and find evidence supporting the validity of our analysis results in a practical setup (subsection 5.4). In most cases, we observe that clustering performance is higher when the unseen classes share the same sementic domain as the training classes. Moreover, adding semantically relevant training classes improves performance, whereas adding unrelated training classes does not lead to performance improvement.

Our contributions are summarized into following:

- We analyze the classifier feature, providing insights into how feature extractors operate:
  - Higher *train-unseen similarity* increases *cohesion*.
  - Higher *train-unseen similarity* increases *separability* between data *assigned* to different classes but reduces it otherwise.
  - Expressiveness of feature improves with an increased number of *spikes* non-orthogonal to input.

- We generalize the distribution assumption of prior works and present novel proof techniques for classifier analysis.

- The theoretical results are validated through diverse experiments, including synthetic and real-world datasets.

## 1.1. Related Works

**Metric Learning and Open Set Clustering** Metric learning is proposed to cluster visually similar unseen classes using classification or triplet loss (Movshovitz-Attias et al., 2017; Zhai & Wu, 2019; Boudiaf et al., 2021). Several recent approaches have focused on increasing the number of classes in the training data to improve clustering. One approach adds virtual classes (Chen et al., 2018; Qian et al., 2020; Gu et al., 2021). Another approach suggested leveraging a larger number of classes induced from Schuhmann et al. (2021) to achieve state-of-the-art performance (An et al., 2023). This aligns with our analysis, which suggests that performance improves as the number of relevant classes in clustering increases.

**Neural Collapse (NC) and Unconstrained Layer-Peeled Model (ULPM)** Recent studies have introduced the concept of Neural Collapse (Papyan et al., 2020) to explain the emergence of intra-class features and feature-weight alignment in trained neural networks. Several studies propose the ULPM to understand training dynamics of NC treating features and weights as unconstrained free variables (Fang et al., 2021; Zhu et al., 2021; Ji et al., 2022; Tirer & Bruna, 2022). However, ULPM, unlike the two layer network model we use, assumes the free variable features, which limits analyzability about input distribution and, consequently, prevents studying feature transferability.

**Feature Learning in Two-Layer Networks** Many works (Louart et al., 2017; Goldt et al., 2020; Hu & Lu, 2022) study the Conjugate Kernel (CK), which enables the analysis of the structure of the first layer in two-layer networks. Ba et al. (2022); Moniri et al. (2024); Ba et al. (2023) argue that feature learning aids in reducing the population risk when evaluated on distributions same to the training data. Unlike these studies, we claim that the CK feature learning model not only explains this generalization but also enables the analysis of features from non-identical distributions, facilitating a deeper understanding of feature transfer.

Additional related works are provided in Appendix A.

## 2. Problem Statement

**Notations** Let $\|\cdot\|$ be $L^2$ or the operator norm. Let $\odot$ be the Hadamard product. Let $A^{\circ k}$ be the Hadamard power. Let $C, c > 0$ and $\kappa \in \mathbb{R}$ be constants that may change from line to line. Define $[d] \triangleq \{1, 2, \cdots, d\}$. For $o, O, \Theta$ notations we follow Moniri et al. (2024)

**Training Data** We define data for one vs. one classification with $\#_{cls}$ classes. The number of problem $\#_P \triangleq \frac{\#_{cls}(\#_{cls}-1)}{2}$. Let $\#_{cls}$ be the number of training classes, and let $\mathscr{C}_1, \cdots, \mathscr{C}_{\#_{cls}}$ represent the class-conditional distri-

butions of the training data. Define the training dataset as $\mathcal{D} = (X, Y)$, where $X \in \mathbb{R}^{\mathbf{n} \times \mathbf{d}}$, $Y \in [\#_{cls}]^{\mathbf{n}}$, $X = (\{x \sim \mathscr{C}_1\} \times m \cup \cdots \cup \{x \sim \mathscr{C}_{\#_{cls}}\} \times m)$, where $\#_{cls} m = \mathbf{n}$ and $m$ is the number of instances per class. Let $\tilde{\mathcal{D}} = (\tilde{X}, \tilde{Y})$ an i.i.d. copy of $\mathcal{D}$.

**Network Structure**  We consider two-layer networks. The initial weight of the first layer, $W_0 \in \mathbb{R}^{\mathbf{d} \times \mathbf{N}}$, is initialized as $W_0[i] \sim Unif(\mathbb{S}^{\mathbf{d}-1})$ for $i \in [\mathbf{d}]$. We denote $W$ obtained via a single step of gradient descent. The initial weights of the second layer, $a_{ij} \in \mathbb{R}^{\mathbf{N}}$ for $i, j \in [\#_{cls}]$ s.t. $i < j$, are initialized as $a_{ij} \sim \mathbf{N}(0, \frac{1}{\mathbf{N}}I)$. We define the initialized feature as $F_0(x) \triangleq \sigma(W_0^\top x)$ and the one-step trained feature as $F(x) \triangleq \sigma(W^\top x)$. The network output is defined as the following $\#_P$-dimensional vector: $(F(x)^\top a_{ij})|_{ij}$.

**Proportional Regime**  We consider the two-layer neural networks in the proportional regime. $\mathbf{n}$, $\mathbf{d}$, and $\mathbf{N}$ are sample size, data and feature dimension, respectively. We perform our analysis under $\mathbf{d}/\mathbf{n}, \mathbf{N}/\mathbf{n} \to c$ as $\mathbf{n}, \mathbf{d}, \mathbf{N} \to \infty$.

**Optimization Problem**  Denote the set of all network parameters as $\theta = \{W, a_{12}, \cdots, a_{\#_P-1, \#_P}\}$. Let $X_{ij}$ be a matrix in $\mathbb{R}^{2m \times d}$, where the first $m$ rows contain samples $x \sim c_i$ and the last $m$ rows contain samples $x \sim c_j$. Let $y \triangleq [1, 1, \ldots, 1, -1, \ldots, -1]^\top \in \mathbb{R}^{2m}$ be a vector consisting of $m$ ones followed by $m$ negative ones. To classify the given data, we use the Mean Squared Error,

$$L(x, y; \theta) = \frac{1}{2n} \sum_{i<j}^c \|y - \sigma(X_{ij}W)a_{ij}\|^2. \quad (1)$$

The weight update formula for the first layer is given by $W = W_0 + G$, where $G \triangleq -\frac{\partial L}{\partial w} = \sum_{i<j} G_{ij}$, s.t.

$$G_{ij} = -\frac{1}{n}\left[X_{ij}^T[(\sigma(X_{ij}W)a_{ij} - y)a_{ij}^T \odot \sigma'(X_{ij}W)]\right]. \quad (2)$$

Now, we introduce the assumptions for theoretical analysis.

**Assumption 2.1** (Activation Function). Let $\sigma(x)$ be an element-wise activation s.t. $\sigma, \sigma', \sigma''$ is bounded by $\lambda_\sigma$ almost surely. It admits a Hermite decomposition i.e. $\sigma(z) = \sum_{k=0}^\infty c_k H_k(z)$, where $c_k = \frac{1}{k!}\mathbb{E}[\sigma(z)H_k(z)]$ for standard gaussian $z$. We assume $c_0 = 0, c_1 > 0$ and $c_k^2 k! \leq Ck^{-3/2-w}$, for constants $C, w > 0$. For example, Shifted ReLU $\max(x, 0) - \frac{1}{\sqrt{2\pi}}$ satisfies this condition.

**Assumption 2.2** (Training Data). Let the class-conditional training data distributions $\mathscr{C}_i$ be non-centered Sub-Gaussians (Vershynin, 2018; Cao et al., 2021; Cole & Lu, 2024). This distribution family is suitable for classification, including distributions with limited support that are separable. It is an extension of the Gaussian assumption of Ba et al. (2022).

## 3. Feature Decomposition

This section analyzes the learning dynamics during a single gradient descent step. First, we demonstrate that the gradient with respect to the $W_0$ exhibits an almost Rank-$\#_P$ property within the proportional regime. Subsequently, we prove that the learned features can be predominantly expressed as Rank-$\#_P$ components, establishing the dominant components for subsequent analyses.

**Gradient Decomposition**  We decompose the gradient (equation 2) using Hermite decomposition, which allows us to extract the essential rank-one matrix structure for each $ij$-th classification problem. Note that $\sigma' = c_1 + \sigma'_\perp$.

$$\begin{aligned} G_{ij} &= \frac{c_1}{n}X_{ij}^T ya_{ij}^T + \frac{1}{n}X_{ij}^T ya_{ij}^T \odot \sigma'_\perp(X_{ij}W_0) \\ &\quad - \frac{1}{n}X_{ij}^T \sigma(X_{ij}W_0)(a_{ij}a_{ij}^T) \odot \sigma'(X_{ij}W_0) \\ &\triangleq \mathbb{A}_{ij} + \mathbb{B}_{ij} + \mathbb{C}_{ij}. \end{aligned} \quad (3)$$

We derive the norm bound for the terms $\mathbb{A}_{ij}$, $\mathbb{B}_{ij}$, and $\mathbb{C}_{ij}$ in Lemma I.1. Using these bounds, we establish the following Theorem 3.1. For the proof, please refer to Appendix I

**Theorem 3.1** (Approximation of Gradient). *Under the assumptions in section 2, and when $\mathbf{n}$ satisfies $\frac{1}{2} > \kappa \frac{\log^2 \mathbf{n}}{\sqrt{\mathbf{n}}}$, the following holds w.p. $1 - C(\mathbf{n}e^{-c\log^2 \mathbf{n}} + e^{-c\mathbf{n}})$:*

$$\left\|G - \sum_{i<j}\mathbb{A}_{ij}\right\| \leq \kappa \frac{\log^2 \mathbf{n}}{\mathbf{n}}. \quad (4)$$

**Feature Decomposition**  Now we utilize $\sum_{i<j} \mathbb{A}_{ij}$ to decompose the feature extractor. We decompose the one-step trained feature function $F(x) = \sigma((W_0 + G)^\top x)$, which serves as a key step in deriving our main analysis. For the proof, please refer to Appendix J.

**Definition 3.2** (Data-Label Covariance). Data-Label Covariance for $X_{ij}$ is defined as $\beta_{ij} = \frac{1}{n}X_{ij}^\top y \in \mathbb{R}^{\mathbf{d}}$.

**Theorem 3.3** (Decomposition of Trained Features). *Under the assumptions in section 2, let $F_0 = \sigma(\tilde{X}W_0)$, $L \triangleq \log n$, $F_0^L = \sum_{k=1}^L c_k H_k(\tilde{X}W_0)$, and $spike_L = \sum_{k=1}^L c_1^k c_k(\tilde{X}\sum_{i<j}\beta_{ij}a_{ij}^T)^{ok}$. With probability $1 - o(1)$,*

$$F = F_0^L + spike_L + \Delta. \quad (5)$$

*Moreover, $\|spike_L\|$ is greater than $\sqrt{n}$, $\|F_0^L\| = \Theta(\sqrt{\mathbf{n}})$, and $\|\Delta\| = o(\sqrt{n})$.*

Based on these results, we analyze the feature representation using the approximation $F_L$, which dominates the residual term $\|\Delta\| = o(\sqrt{\mathbf{n}})$ with probability $1 - o(1)$.

(a) Cohesion      (b) Separability

Figure 2: **Numerical Observation of Cohesion and Separability**. Plot of *Cohesion* and Heatmap of *Separability* calculated by adjusting $\beta^\top \mu_1$ and $\beta^\top \mu_2$.

**Definition 3.4** (Dominant Feature $F_L = F_0^L + \text{spike}_L$).

$$F_L(x) \triangleq \sum_{k=1}^{L} c_k [H_k(\tilde{X} W_0) + c_1^k (\sum_{i<j} (\beta_{ij}^\top x) a_{ij}^T)^{\circ k}]. \quad (6)$$

Using the feature decomposition conducted so far, the next section analyzes clustering risk and explores the conditions for effective clustering of unseen data.

## 4. Feature Analysis

### 4.1. Clustering Risk Analysis in binary classification

In this section, we analyze clustering risks. We show **train** ($\beta$)-**unseen** ($\mu$) similarity governs the the clustering population risk i.e. *Cohesion* and *Separability* of $F_L$ from Definition 3.4 under condition 4.4. We derive *cohesion* and *separability* of $F_L$ for two "unseen" class-conditional distributions.

**Definition 4.1** (*Train-Unseen Similarity*). Given Train Data-Label Covariance $\beta$ in Definition 3.2 and mean of Unseen distribution $\mu$, *Train-Unseen Similarity* is defined as $\beta^\top \mu$.

*Note* 4.2 (Explanation of *assignment* and $\beta^\top \mu$). $\beta_{ij}$ represents the normal vector of the linear decision boundary, i.e. the direction determining class $i$ vs. $j$ based on the sign of its inner product with data. Therefore, the sign of $\beta^\top \mu$ indicates the class *assignment* of unseen data with $\mu$.

**Definition 4.3** (*Cohesion* and *Separability*). We define the clustering risks based on similarity between feature vectors using inner products.

*Cohesion* measures the expected similarity between i.i.d. features of the same class over network parameters $\theta$ and data $x, x' \sim c_1$, i.e.

$$\mathbb{E}_\theta[\mathbb{E}_{x \sim c_1} F(x)^T \mathbb{E}_{x' \sim c_1} F(x')].$$

*Separability* measures the expected dissimilarity between independent features of different classes over $\theta$, $x \sim c_1$ and $x' \sim c_2$ i.e.

$$-\mathbb{E}_\theta[\mathbb{E}_{x \sim c_1} F(x)^T \mathbb{E}_{x' \sim c_2} F(x')].$$

*Condition* 4.4. We fix $\mathbf{n}, \mathbf{d}, \mathbf{N}$ large enough. Under assumptions 2.1, 2.2, let $c_i = \mathcal{N}(\mu_i, I_{\mathbf{d}})$ for $i \in [2]$ be the class conditional distributions. Define $\rho_{k,k'}^{(1)} > 0, \rho_{k,k'}^{(2)}(cos(\mu_1, \mu_2)), \rho_{k,k',r}^{(3)} > 0, \rho_{k,k',r,r'}^{(4)} > 0$ as functions of $N, d$. Note that $\rho_{k,k'}^{(2)}$ increases as $cos(\mu_1, \mu_2)$ grows. Exact definitions are in Def. K.1. The Shifted ReLU, as stated in Assumption 2.1, is used as the activation.

**Proposition 4.5** (Cohesion). *Following condition 4.4, the Cohesion of $F_L$ for $c_i$, $i \in [2]$ is given by:*

$$\sum_{k=1,k'=1}^{L} c_k c_{k'} \begin{bmatrix} \rho_{k,k'}^{(1)} \|\mu\|^{k+k'} \\ +2\sum_{r'=0}^{k'} \rho_{k,k',r'}^{(3)} |\mu^T \beta|^{k'-r'} \|\beta\|^{r'} \|\mu\|^k \\ +\sum_{r,r'=(0,0)}^{(k,k')} \rho_{k,k',r,r'}^{(4)} |\mu^T \beta|^{k+k'-r-r'} \|\beta\|^{r+r'}. \end{bmatrix} \quad (7)$$

**Proposition 4.6** (Separability). *Following condition 4.4, the Separability of $F_L$ for $c_1$, $c_2$ is given by:*

$$-\sum_{k=1,k'=1}^{L} c_k c_{k'} \begin{bmatrix} \rho_{k,k'}^{(2)}(cos(\mu_1, \mu_2)) \|\mu_1\|^k \|\mu_2\|^{k'} \\ +\sum_{r=0}^{k} \rho_{k,k',r}^{(3)} |\mu_1^T \beta|^{k-r} \|\beta\|^{r'} \|\mu_2\|^{k'} \\ +\sum_{r'=0}^{k'} \rho_{k,k',r'}^{(3)} |\mu_2^T \beta|^{k'-r'} \|\beta\|^{r'} \|\mu_1\|^k \\ +\sum_{r,r'=(0,0)}^{(k,k')} \rho_{k,k',r,r'}^{(4)} (\mu_1^T \beta)^{k-r} (\mu_2^T \beta)^{k'-r'} \|\beta\|^{r+r'}. \end{bmatrix} \quad (8)$$

The proofs of Propositions 4.5 and 4.6 are provided in Appendix K. We numerically analyze the results of propositions 4.5 and 4.6 to investigate *Cohesion* and *Separability* further. For this numerical observations, we set $\|\mu_1\| = \|\mu_2\| = \|\beta\| = 1$, $\mu_1 = -\mu_2 \in \mathbb{R}^{320000}$ and $L = \log_{10} \mathbf{n}$. We calculate equation 7 and equation 8 by adjusting $\mu_1^T \beta$ and $\mu_2^T \beta$, as shown in Figure 2, which demonstrates the *Cohesion* and *Separability* of $F_L$. *Cohesion* increases when the $|\mu^T \beta|$ increases. *Separability* increases when $\mu_1^T \beta$ and $\mu_2^T \beta$ grow with opposite signs and decreases when they grow with the same sign. Moreover, we observe that this phenomenon is governed by the last term of equation 7, 8 (related to $\rho^{(4)}$), as shown by separately computing this term and the others numerically in Appendix B. Additionally, under the theoretical setup, we observe that our hypothesis tends to hold over a wider range as $\mathbf{n}$ increases (please refer to Appendix B).

The analytical results in equation 7 and equation 7 can be explained as follows. With $\rho^{(4)} > 0$, the last term inside the bracket of *Cohesion* in equation 7 increases in value as *Train-Unseen Similarity* grows. The last term of *Separability* is influenced by $(\mu_1^T \beta)^{k-r} (\mu_2^T \beta)^{k'-r'}$. Provided that $k - r$ and $k' - r'$ are odd, this term implies that if the *Train-Unseen Similarities* have opposite signs and increase, then this term improves; otherwise, if the signs are the same and increase, *Separability* decreases. According to the analysis

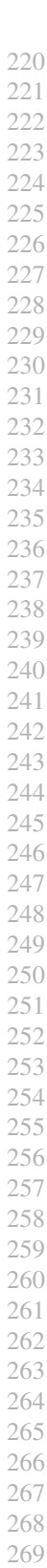

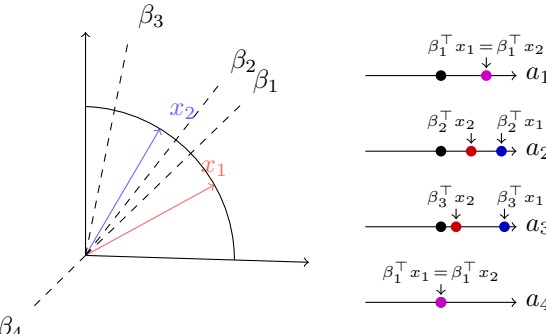

Figure 3: As shown in equation 6, after one step of training with spike $\beta_1, \beta_2, \beta_3, \beta_4$, the inner product between input $x_i$ and $\beta_i$ acts as the coefficient in the linear combination of $a_i$, forming the *spikes* structure of the feature.

in Appendix H, the first coefficient $c_1$ of Shifted ReLU is a large positive value, and subsequent Hermite coefficients approach zero while oscillating around it. Thus, we hypothesize that the positive part is likely to dominate $\sum c_k c_{k'}$, but further work is needed to confirm this.

### 4.2. Spike Component Analysis

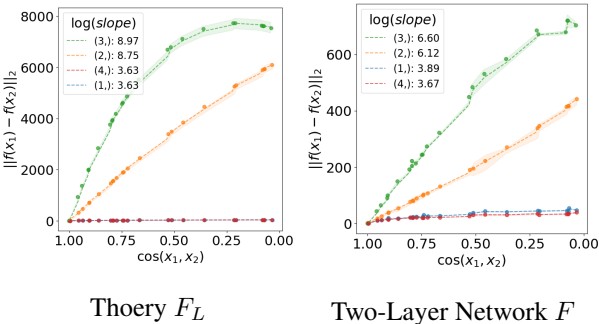

Thoery $F_L$      Two-Layer Network $F$

Figure 4: When trained along the directions $\beta_2$ and $\beta_3$, we observe significant changes in the feature space distance as $x_1$ and $x_2$ vary, compared to $\beta_1, \beta_4$.

In this section, based on the previous feature decomposition and extend it to examine the impact of a multi-class classifier's spike structure on unseen data clustering. We examine the spike structure in $F_L = F_0^L + \text{spike}_L$ and its influence on feature mapping. This examination allows us to explore the impact of the training data's structure $\beta$ on the feature generation of unseen data. The spike structure inside the Hadamard power involves the linear combination coefficient $\beta_{ij}^\top x$ and the random initialized classifier head $a_{ij}$ (equation 3.4). Thus, the feature extraction is closely linked to the inner product between $\beta_{ij}$ and the input point $x$. If the direction of $x$ is not orthogonal to $\beta_{ij}$, then spike of $\beta_{ij}$ involve feature extraction.

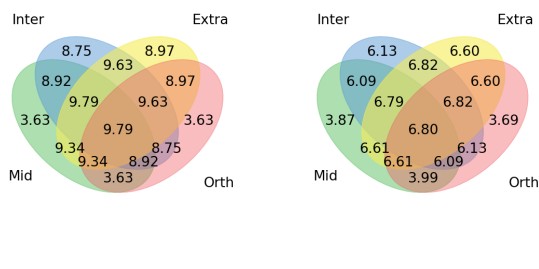

(a) Theory      (b) Two-layer Networks

Figure 5: Comparison of log average slope between Theory and Two-layer Networks. ■ Midpoint ($\beta_1$) ■ Interpolation ($\beta_2$) ■ Extrapolation ($\beta_3$) ■ Orthogonal ($\beta_4$). The intersection implies learning intersecting $\beta$.

Conversely, when $x$ is orthogonal to $\beta_{ij}$, the impact of spike $\beta_{ij}$ is eliminated. To validate this, we define following four spikes, given test input $x_1, x_2 \in \mathbb{S}^{d-1}(\sqrt{\mathbf{d}})$, $\beta_1 = \frac{x_1+x_2}{2}, \beta_2 = \frac{x_1+3x_2}{4}, \beta_3 = \frac{-x_1+5x_2}{4}$ and $\beta_4$, a random vector orthogonal to $x_1, x_2$. Then, the magnitudes are adjusted to $\sqrt{\mathbf{d}}$. By definition, $\beta_1, \beta_4$ cannot contribute to feature extraction because they are Midpoint or Orthogonal, while $\beta_2$ and $\beta_3$ can distinguish the two inputs. For illustration see Figure 3.

Now, we demonstrate this explanation using the approximated features $F_L$ and the two layer neural network $F$ with the four disjoint sub-classification problem [1] defined as follows: We generated four classification problems by creating Gaussian training data with means $\beta_i$ and $-\beta_i$, and a covariance of $0.1I$ for $\mathbf{n}, \mathbf{d}, \mathbf{N} = 2^{11}$, enabling the networks to learn $\beta_i$ as their *spike*. $F$ is trained by this data and $F_L$ is calculated by its definition. We observed the feature distance between $F(x_1), F(x_2)$ and between $F_L(x_1), F_L(x_2)$ for $\binom{4}{k}$ combinations of $\beta_i$ in this problem by varying the angle between $x_1, x_2$. Please refer to Figure 4 and 21 for results. It can be observed that the feature from $\beta_1$ and $\beta_4$ hardly captures variations in the angle of test input $x_1, x_2$ within the data space. In contrast, the feature from $\beta_2$ and $\beta_3$ is highly sensitive to such variations, suggesting that it effectively preserves the structural changes in the input data. Both $F_L(x_1)$ and $F(x_1)$ exhibit the same trends, which supports the validity of our feature approximation. To aggregate these combinatorial results, we measure the log of the average slope, which indicates that features with sensitive changes tend to have larger values, as shown in Figure 5.

As a result in Figure 5, we observe that when multiple $\beta$s are used in training, features are more sensitive to changes

---

[1]Instead of studying all combinations for 8 classes classification, we simplify the task by grouping four pairs, performing only four combinations of classifications.

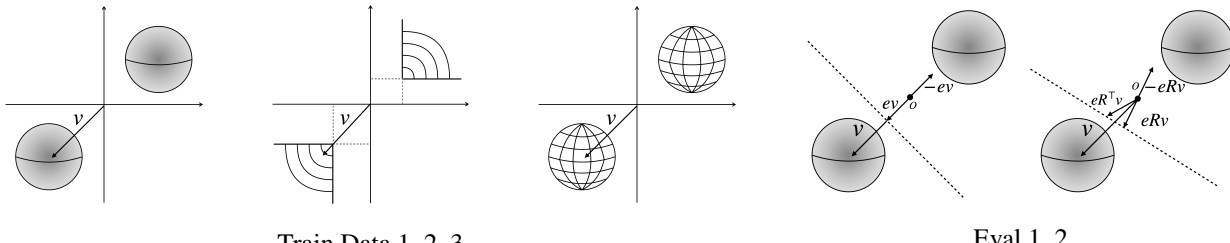

Train Data 1, 2, 3                                                Eval 1, 2

Figure 6: Examples of training datasets (Data 1, 2, 3) and evaluation data Eval 1, 2.

in distance within the data space. Meanwhile, the Midpoint $\beta_1$ and Orthogonal spike $\beta_4$ seem ineffective for feature extraction, even when learned alongside other spikes. Experiments show that learning representations with unrelated classes limits expressiveness, while related classes enhance the model's ability to capture fine-grained features of unseen data. This trend is consistently observed in real-world datasets in Expr V, VI at subsection 5.4. Additionally, to clarify the effect of the spikes, we compute $F_0^L$ and spike$_L$ separately as shown in Figure 22. The results show that the spike$_L$ created by $\beta_1$ and $\beta_4$ embeds $x_1$ and $x_2$ as the same feature. Therefore, it confirms that the distinction between $x_1$ and $x_2$ created by the model trained with $\beta_1$ and $\beta_4$ is due to the random feature $F_0^L$.

## 5. Experiments

*Remark* 5.1. *recall@1* $\triangleq \mathbb{E}_{x_i, y_i} \mathbf{1}_{y_i = \hat{y}_{i,1\text{-NN}}}$. $\hat{y}_{i,1\text{-NN}}$ is class of the closest feature to $x_i$. This is a feasible measure for evaluating whether new classes form clusters.

In this section, we conduct seven experimental setups to validate our theoretical results. First, in Experiments I, II and III, we utilize a synthetic dataset to confirm that, as discussed in subsection 4.1, *Cohesion*, *Separability* are determined by the *Train-unseen similarity*. Second, to demonstrate how our theoretical explanations can provide intuition in practical settings, we conduct Experiments IV, V, VI, and VII. For this purpose, we analyze the open-set clustering problem using fine-grained real image datasets.

### 5.1. Setup for Theory Vaildation: Expr. I, II, III

We use three types of different non-centered Sub-Gaussian distributions as training datasets that are symmetric about the origin. For the evaluation, we introduce two distribution i.e. Eval 1, Eval 2 with translation parameter $e$ and rotation parameter $R \in \mathcal{R} \subseteq \mathrm{SO}(n)$ to control the *train-unseen similarity* $\beta^\top \mu$. e.g. as $e$ increases from 0 towards 1, $\beta^\top \mu$ increases, and as $R$ approaches the identity matrix $I$, $\beta^\top \mu$ increases. For illustration of the data, see Figure 6. For detail, refer to subsection D.1. We follow the condition described in section 2 and subsection 4.1.

Now we explain Expr. I, II, III. For each experiment, we utilize all datasets $1, 2, 3$, with distinct Eval data usage. Expr. I uses two Eval 1 data with translation parameter $e_1 \in [-0.9, 0.9]$ and $e_2 = -e_1$, so they are *assigned* to opposite training classes (say pos-neg). Experiments II and III are based on two Eval 2 data distributions, each parameterized by a small-angle random rotation matrix $R \in \mathcal{R}$. In Experiment II, considering the case where the datasets are *assigned* to opposite classes, the first distribution uses $R$ and the second distribution is origin symmetry of the first distribution. In Experiment III, considering the situation where the datasets are *assigned* to the same class (say pos-pos), the first distribution uses $R$ and the second uses $R^\top$ to slightly rotate given means.

### 5.2. Results of Theory Vaildation: Expr. I, II, III

In this experiment, we examine the relationships between the *train-unseen similarity*( i.e. $\beta^\top \mu$), *Cohesion*, *Separability* that we discussed in subsection 4.1 and *Recall@1* to evaluate performance using practical measures. All test data are generated symmetrically, so for simplicity in visualization, we report the measurement for a single class. For Expr I, we present a summary of the results in Figure 8. We observe that for large values of $|\beta^\top \mu|$, strong *Cohesion* and *Separability* occur across all datasets. For Expr II and III, in accordance with the *Separability* structure observed in subsection 4.1, when the signs of $\beta^\top \mu_1, \beta^\top \mu_2$ are opposite (Expr II), we observed an increase in *Separability*, whereas in the other case (Expr III), we observed a decrease Figure 7. For *recall@1*, we observed a similar trend as *Separability*. These results correspond to our theoretical findings. For individual graphs, refer to Appendix D.

### 5.3. Setup for Practical Vaildation: Expr. IV, V, VI, VII

We designed experiments to examine whether these insights are also applicable to clustering performance in image datasets and practical neural networks. In these scenarios, we utilize *train-unseen similarity* to conceptualize semantic similarity between training and unseen classes (Expr. IV). The number of non-orthogonal *spikes* is interpretable as the number of semantically similar or dissimilar training classes

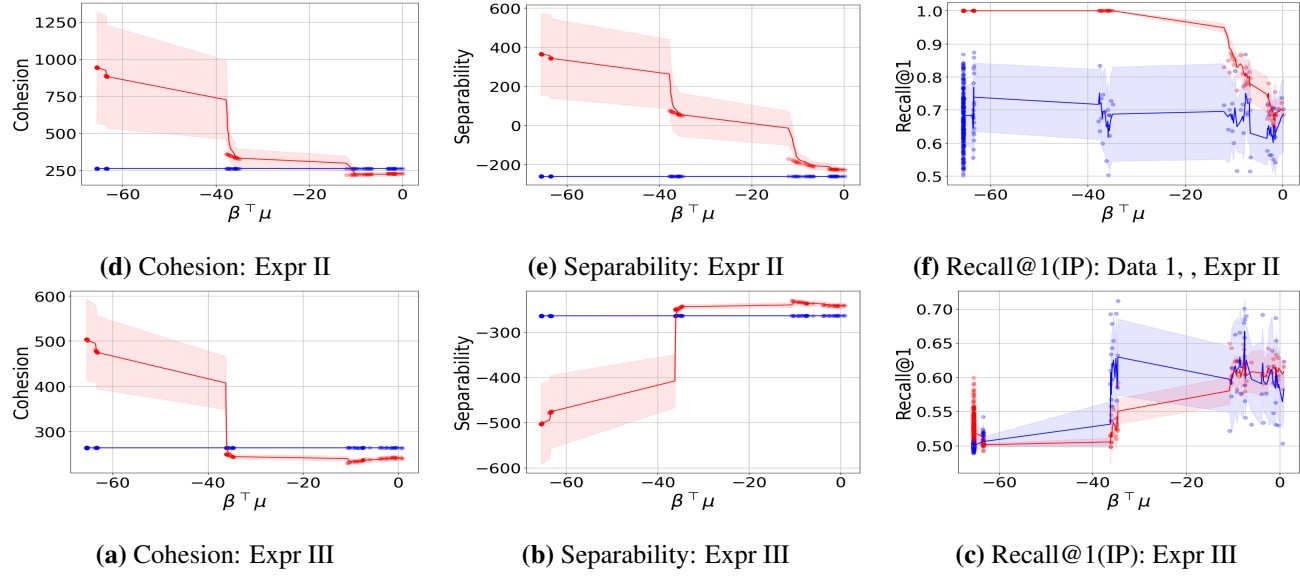

**(d)** Cohesion: Expr II  **(e)** Separability: Expr II  **(f)** Recall@1(IP): Data 1, , Expr II

**(a)** Cohesion: Expr III  **(b)** Separability: Expr III  **(c)** Recall@1(IP): Expr III

Figure 7: Data 1 evaluated in the Eval 2 setup. Upper row: In Expr II, all metrics increase as $|\beta^\top \mu|$ increases. Lower row: In Expr III, where two test classes are *assigned* to a single train class, recall@1 and Separability tend to decrease as $|\beta^\top \mu|$ increases. This aligns with our predictions. The red line — represents the values after one step training. Tje blue line — represents the values from initialization.

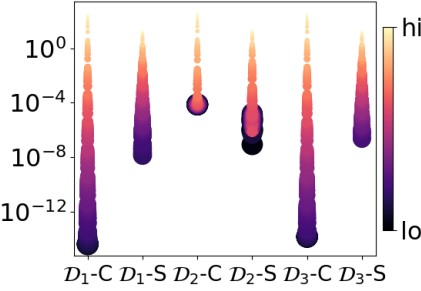

Figure 8: Summary of Expr. I. $D_i$ denotes Data $i$ and C, S denote *Cohesion* and *Separability*. Dark and large points indicate low $|\beta^\top \mu|$ values, while the opposite indicates high values. All measurements increase with respect to $|\beta^\top \mu|$. We scaled using the absolute value at the 85th percentile.

(Expr. V, VI). Additionally, we validate whether removing the duplicatively *assigned* unseen classes improve clustering risk compared to random removal, as suggested by the results of *Separability* (Expr. VII).

For this investigation, we used the benchmark datasets CAR(Vehicle) (Krause et al., 2013), CUB(Bird) (Wah et al., 2011), SOP(Product) (Song et al., 2015), and ISC (Clothing) (Liu et al., 2016), referred to as *Domain*. Additionally, we utilized ImageNet subsets corresponding to the domains Vehicle, Bird, Product, and Clothing, denoted as I(V), I(B), I(P), and I(C), referred to as *sub In1k* for extra classes. Also,

we performed experiments on the whole classes ImageNet by sampling 100 instances per class (say *subsampled whole In1k*). Details are in Appendix N. The objective function and most experimental configurations followed the approach outlined in Zhai & Wu (2019), which is a seminal baseline. We use ResNet18 and ResNet50 (He et al., 2015). In addition to the randomly initialized networks in the main text, we conducted experiments with pre-trained networks common in feature learning, and results are included in Appendix E. The two setups exhibited similar trends.

### 5.4. Results of Practical Vaildation: Expr. IV, V, VI, VII

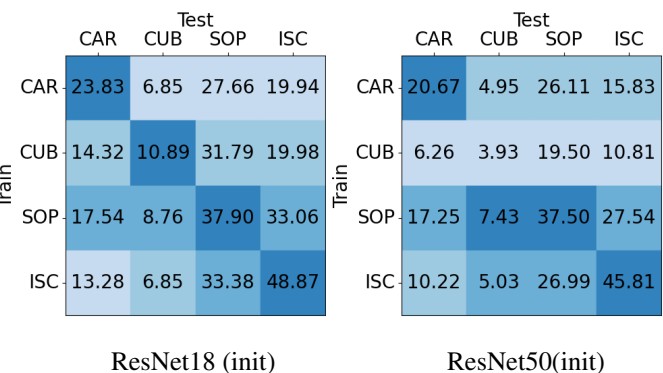

Figure 9: Expr. IV, *recall@1* measurements. Most cases show the highest performance when the domain of the Train and Test corresponds.

For **Expr. IV**, we trained with each *Domain* dataset (CAR, CUB, SOP, and ISC train datasets) and *Domain+sub In1k* dataset (CAR+I(V), CUB+I(B), SOP+I(P), and ISC+I(C)), and then measured how each model well cluster on all of the test datasets (CAR, CUB, SOP, ISC test datasets). As shown in Figure 9, we verify whether clustering the test dataset related to the train classes is more effective than clustering unrelated data, analogous to result in subsection 4.1.

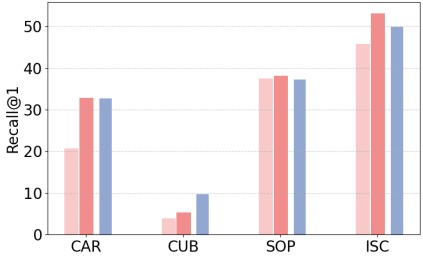

Figure 10: Expr V in ResNet50(init). The pink ▇, red ▇, and blue ▇ bars represent *Domain*, *Domain+sub In1k*, *Domain+subsampled whole In1k*, respectively.

In **Expr. V**, we measured the clustering performance for corresponding test datasets after learning the *Domain*, *Domain+sub In1k*, and *Domain+subsampled whole In1k*. We find that adding classes from the entire ImageNet dataset during training, rather than including only related classes, does not significantly improve clustering (Figure 10, 32).

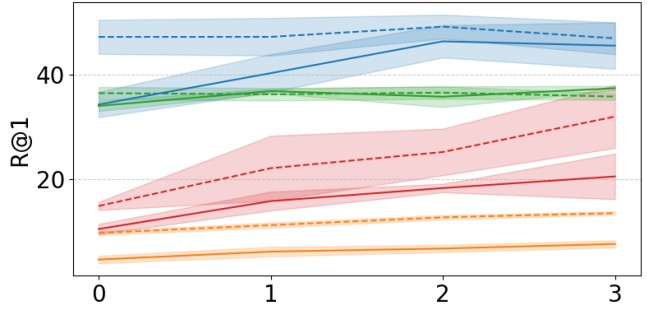

Figure 11: Expr VI, Recall@1 values for the CAR, CUB, SOP, and ISC datasets are shown with dashed lines - - for ResNet18 and solid lines — for ResNet50.

In **Expr. VI**, experiments are conducted by dividing the *Domain* datasets into four steps to observe the impact of increasing the number of related classes on *recall@1* performance (Figure 11). From Step 0 to Step 3, 25%, 50%, 75%, and 100% of the *Domain* dataset classes are sequentially added for training. The added classes are randomly selected, and each experiment is repeated three times. For the number of classes, refer to Table 6. Furthermore, we observed that some results of Expr. V align with those of Expr. VI, as discussed in detail in subsection E.1.

For **Expr. VII**, in evaluation, removing duplicatively *assigned* of unseen classes resulted in a $1.73 \pm 2.87\%$ improvement in recall@1 compared to random removal of same amount of unseen classes, with max improve: 13.65%, min decrease: -3.28%, a success rate: 79% and $p = 9.40 \times 10^{-7}$. This suggest that duplicate *assignments* hinder clustering, which aligns with our theory. Details are in subsection E.2.

## 6. Conclusion

In this study, we explored the feature learning dynamics of a two-layer classifier in the proportional regime to uncover the mechanisms underlying feature transferability. Specifically, we analyzed the conditions where the learned features of unseen classes form cohesive and separable cluster. Our theoretical analysis extends the Conjugate Kernel framework to classification tasks. As a result, our numerical-analytical theory demonstrated that feature *cohesion* increases with greater similarity between training and unseen data, while feature *separability* is influenced not only by similarity but also by avoiding duplicate class *assignments* in binary classification. Additionally, we showed that only when the *spikes* are non-orthogonal to the input, do they get involved in feature extraction. In addition to validation on synthetic datasets, we observed that our theory offers valuable insights even when applied to real-world datasets.

Our empirical findings suggest that clustering performance improves when the test data share the same semantic domain as the training data. Furthermore, adding semantically relevant classes to the training set leads to performance gains, whereas introducing unrelated classes has little effect. Contrary to existing research that focuses on performance improvement through large-scale learning on broad domains (Brown et al., 2020; An et al., 2023), our study provides evidence that only certain relevant knowledge, closely related to the domain, influences feature transfer. This evidence mirrors classical problems in the field of artificial intelligence, such as the frame problem and the installation problem. Specifically, AI agents do not require all available knowledge to solve a given problem; only specific, detailed knowledge is necessary. Dennett (1984) states about this as follows: "People in AI ... take the shortcut of installing all that an agent has to know to solve a problem. This may, of course, be a dangerous shortcut." We hope that our study may remind the AI community of the longstanding principle that it may not be the scale of the data that matters. We have also discussed the limitations and future research directions related to the Hermite expansion approximation and general results for *cohesion* and *separability* in Appendix F.

## Impact Statement

This paper presents work aimed at advancing the field of Machine Learning. In this research, we analyze the potential for clustering performance improvement through the classification training of a large number of highly granular classes. Such an approach may lead to a reduction in the level of personal data masking required for fine-grained data differentiation, which could trigger new ethical discussions regarding privacy protection. Additionally, to effectively implement this approach, there may be a tendency to collect more data, which can have significant implications for the scale and scope of data collection, as well as data management practices.

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

## A. Additional Related Works

**Feature Transferability in Deep Metric Learning**    The explanation for how Deep Metric Learning learns transferable features towards unseen data remains insufficient. Chopra et al. (2005) suggested that CNNs' robustness to geometric distortions enables the creation of generalizable features. This explanation has been replaced in transformer-based research by the idea that, without the inductive biases of CNNs, transformers are less constrained and thus capable of extracting generalizable features (El-Nouby et al., 2021; Caron et al., 2021). Additionally, following the manifold hypothesis (Chang et al., 2003; Lee et al., 2003; Talwalkar et al., 2008; Goodfellow et al., 2016), Liu et al. (2018); Ermolov et al. (2022) explained that normalized softmax for metric learning works well because hyperspherical/hyperbolic feature space and the data lies on a manifold. However, these studies do not provide a detailed analysis of how features are learned and transferred through classification.

**Neural Collapse (NC) and Features learned by Classifiers**    There exist studies exploring Neural Collapse (NC) and features learned by classifiers that cannot be explained under the free variable assumption. Hui et al. (2022) argue that NC does not manifest on test data. Sohoni et al. (2020); Yang et al. (2023) claim that even on training data, NC is not fully realized, with critical fine-grained structures concealed. Notably, Yang et al. (2023) utilized a two-layer network to analyze training data features. Regarding NC on novel data, Galanti et al. (2022) statistically analyze NC in transfer learning, suggesting that NC generalizes not only to new samples within training classes but also to unseen classes with empirical observations. However, their analysis is constrained by focusing on general function spaces rather than specific neural network architectures.

**MSE for Classification**    Utilizing MSE in classification is as well-established as using softmax-cross entropy, especially in theoretical analyses of classification problems (Han et al., 2022; Zhou et al., 2022).

**Generalization Bound for Metric Learning**    Research on the generalization bounds of metric learning related to the U-process we use is also ongoing (Bellet & Habrard, 2015; Huai et al., 2019; Zhou et al., 2024). However, these studies do not analyze the exact feature learning structure.

## B. Empirical Insights into High-Dimensional Asymptotics

In asymptotic analysis, $\mathbf{n}, \mathbf{d}, \mathbf{N} \to \infty$ is crucial for observe result. Please see Figure 12, Figure 13 for the cohesion and Separability in $\mathbb{R}^{2000}, \mathbb{R}^{20000}, \mathbb{R}^{320000}$. As the dimension increases, the range where cohesion and Separability align with our expectations expands.

For component analysis, please see Figure 14, Figure 15, Figure 16 , Figure 17, Figure 18, Figure 19

## C. Additional Observation of Multi Classes Feature Analysis

See Figure 21 for multi-directional training result. For $F_0^L$, and spike$_L$ term depiced in Figure 22, Figure 23.

## D. Additional Results of two-classes Experiments

### D.1. Additional setup for Experiment I, II, III

We set $\mathbf{d} = \mathbf{n} = \mathbf{N} = 2^{11}$ and use Shifted ReLU. We repeat each experiment with 3 different initializations of the neural network parameters.

**Training Datasets**    (Data 1) two uniform distributions over a radius-$\sqrt{\mathbf{d}}$ ball, (Data 2) two multi-dimensional element-wise truncated Gaussian distributions, and (Data 3) two uniform distributions over a radius-$\sqrt{\mathbf{d}}$ sphere, symmetric about the origin [2]. The two means of training class are denoted as $v$ and $-v$, respectively. For Data 1, 3 $v \triangleq 2r \cdot \mathbf{u}$, with $\mathbf{u} \sim \text{Unif}\left(\mathbb{S}^{\mathbf{d}-1}\right)$. For Data 2, one class has support on $[1, \infty)$ across all dimensions, while the other class has support on $(-\infty, -1]$.

**Evaluation Datasets**    Eval 1, 2 use the projected Gaussian distribution, which is projected onto the mean direction of one training data $v$, as defined in equation 9. For Eval 1, we translate mean of projected Gaussian distribution with $e$, and

---

[2]The Sub-Gaussian property is proven for Data 1 and 3 in Vershynin (2018), and for Data 2 in Lemma L.1.

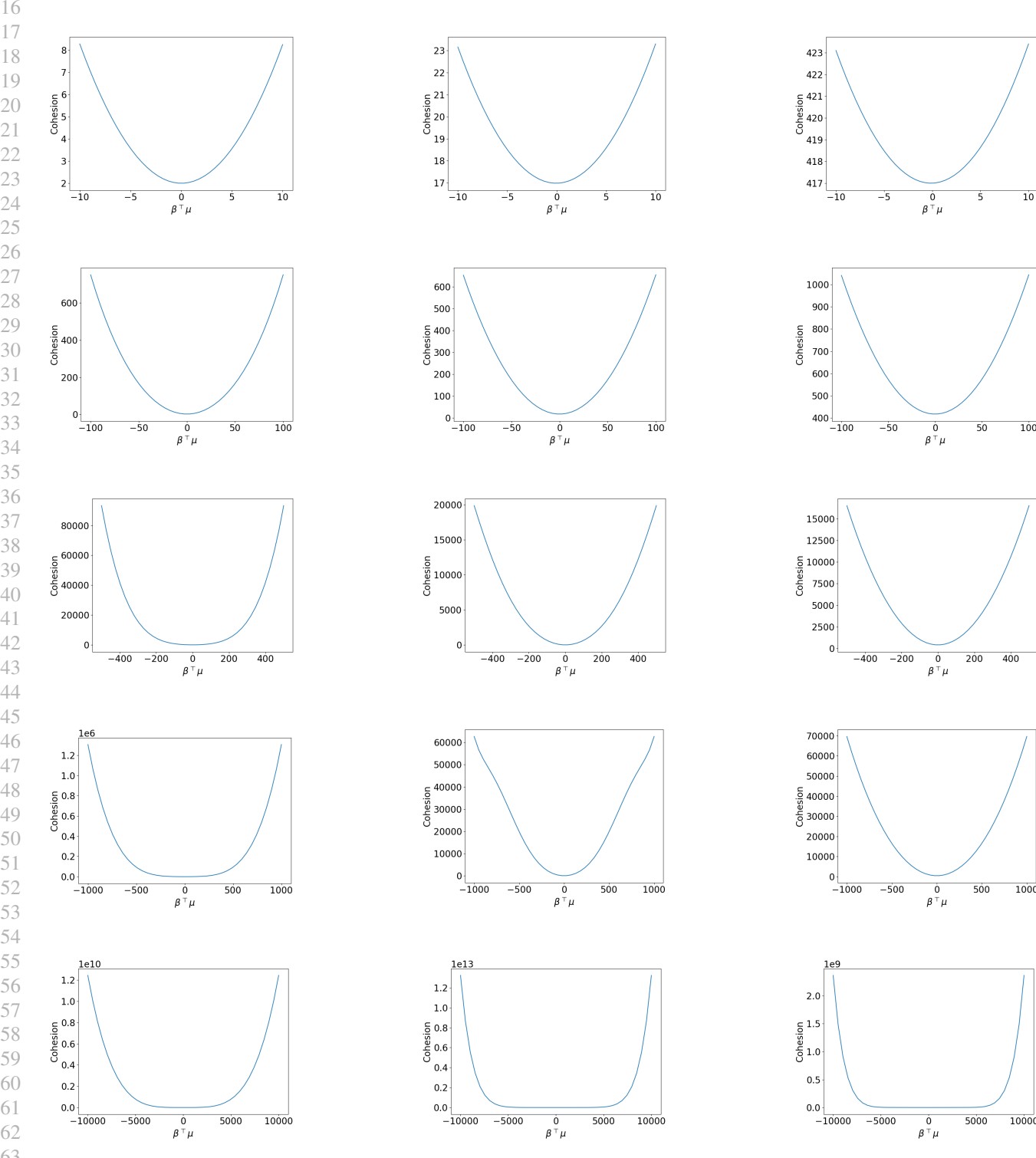

Figure 12: Cohesion in $\mathbb{R}^{2000}, \mathbb{R}^{20000}, \mathbb{R}^{320000}$ (left to right), with the computed range expanding from top to bottom.

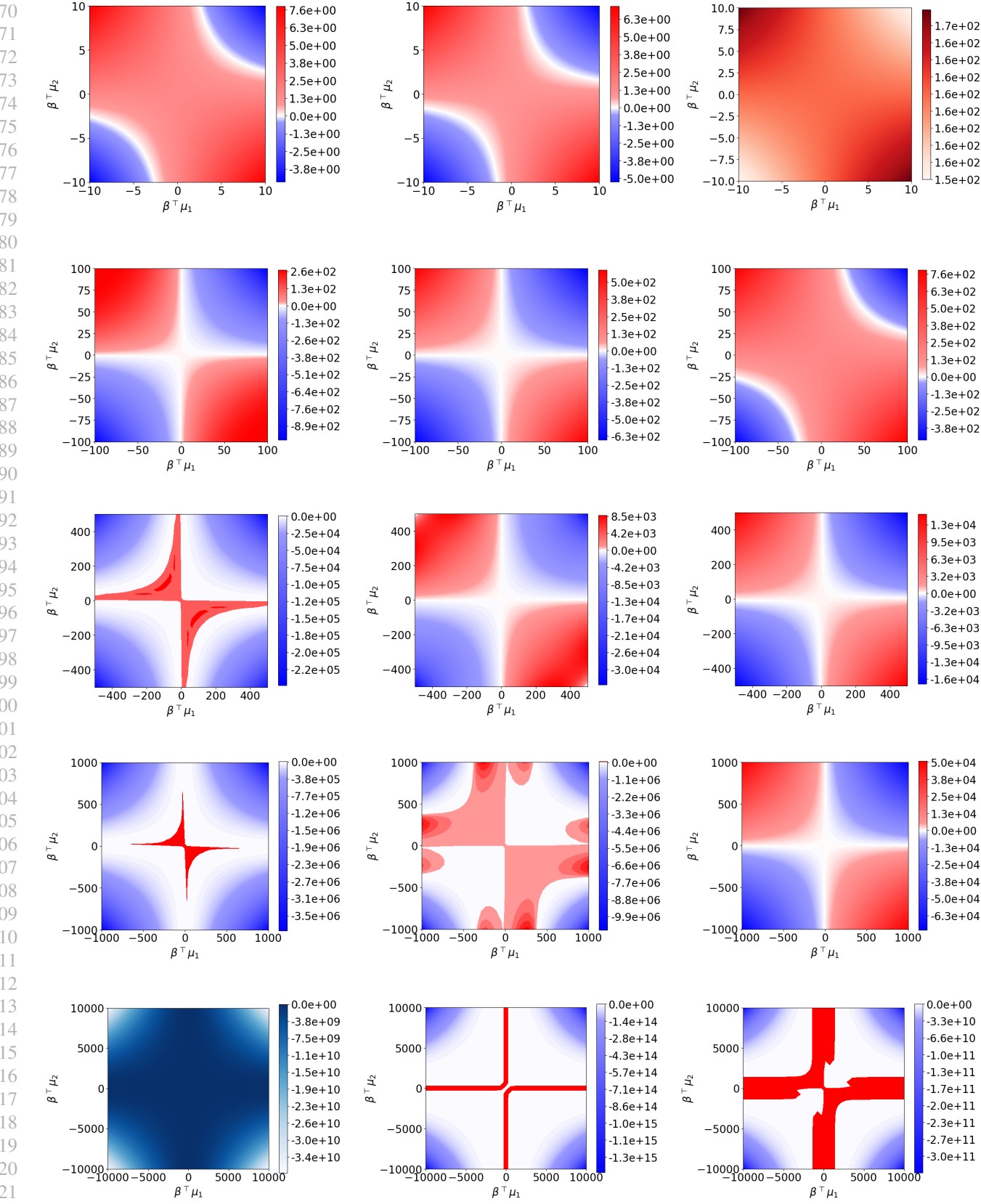

Figure 13: Separability in $\mathbb{R}^{2000}, \mathbb{R}^{20000}, \mathbb{R}^{320000}$ (left to right), with the computed range expanding from top to bottom.

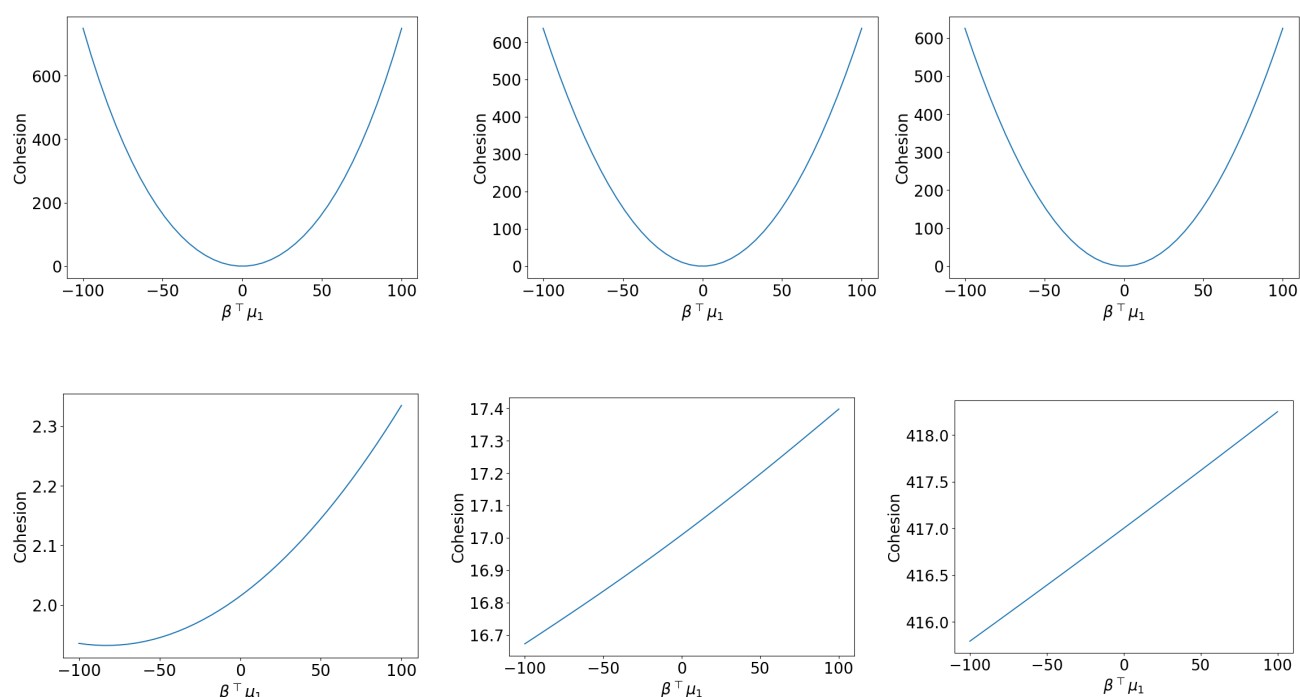

Figure 14: Component analysis of Cohesion in $\mathbb{R}^{2000}, \mathbb{R}^{20000}, \mathbb{R}^{320000}$ (left to right) in range $[-100, 100]$, top: the dominant last component, bottom: sum of the other terms.

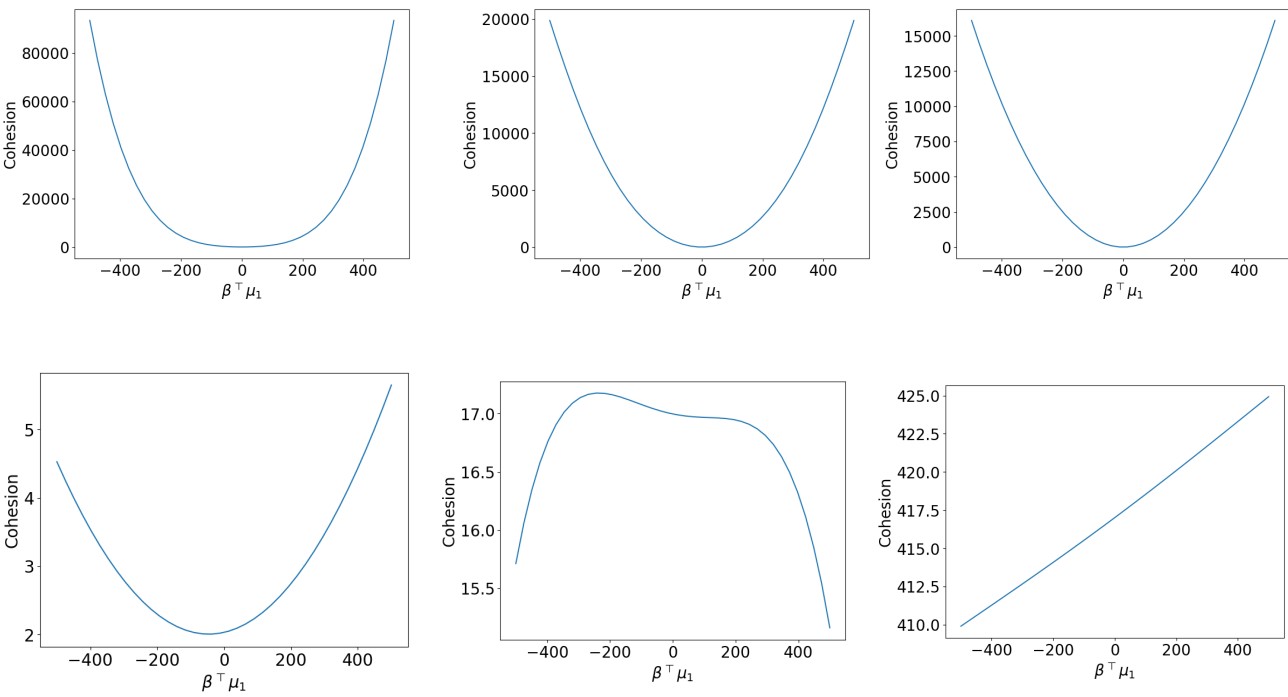

Figure 15: Component analysis of Cohesion in $\mathbb{R}^{2000}, \mathbb{R}^{20000}, \mathbb{R}^{320000}$ (left to right) in range $[-500, 500]$, top: the dominant last component, bottom: sum of the other terms.

for Eval 2, we Rotate mean of projected Gaussian distribution with $R \in \mathcal{R}$ and fixed $e$. We generate 300 distinct rotation matrices $\mathcal{R}$ using the process in Appendix O. The projected gaussian distribution is sampled as follows,

$$z - \frac{z^\top \nu \nu}{\|\nu\|^4} + \nu, \quad \text{where} \quad z \sim \mathcal{N}(0, cI). \tag{9}$$

For Eval 1, $\nu \triangleq ev$, $c = 1$ and for Eval 2, $\nu \triangleq Rev$, $c = 10^{-1}$ with $e = 0.01$ for Data 2 experiment and $e = 0.008$ for Data 1 and 3 experiments, $R \in SO(d)$.

### D.2. Comprehensive Results of All Experiments

The overall experimental results for *Cohesion* and *Separability* are shown in Figure 24. The results for Eval 1 experimental settings are presented in linear scale in Figure 25 and in logarithmic scale in Figure 26. Additionally, as presented in Figure 7, experiments for Eval 2 settings on Data 2 and 3 are shown in linear scale in Figure 27, with results for *Cohesion*, *Separability*, and Recall@1 (IP). Furthermore, results for Recall@1 (cos) are presented in linear scale in Figure 28. All observed results align with the theoretical predictions.

# E. Additional Results of Real-world dataset Experiments

Figure 29 summarizes the experimental results and the purpose of the experiment. Expr. IV is in Figure 30, 31, 1. Expr. V is e in Figure 32, Table 2. Expr. VI is in Figure 33. Expr. VII is in Figure 34, 35, 36, 37, Table 3, and 4.

### E.1. Relation between Expr. V and VI

On the other hand, certain results from Expr. V align with those from Expr. VI. As shown in Table 5, for datasets such as CAR and CUB, the number of additional classes introduced by the *sub In1k* dataset is significantly larger compared to SOP. For these data, inclusion of the additional *sub In1k* dataset contributes to improved *recall@1* performance when trained using a Random Initialized Network. Meanwhile, the performance of the pre-trained network is not significantly affected by the additional dataset. We attribute this to the fact that the pre-trained model is additionally re-trained using the same ImageNet dataset *sub In1k*. These findings suggest that further research on the behavior of pre-trained networks is necessary.

### E.2. Expr. VII: Removing Duplicately *Assigned* Eval Classes

In **Expr. VII**, as suggested by the theoretical results on *Separability*, we validated whether eliminating duplicate in the *assignments* improves performance. To clarify, we will provide an example of duplicate *assignment* at Note E.1.

*Note* E.1 (Example of duplicate *assignment*). For two train classes $\mathscr{C}_1^{(train)}, \mathscr{C}_2^{(train)}$ and two test classes $c_1^{(test)}, c_2^{(test)}$, if most instances of $c_1^{(test)}$ and $c_2^{(test)}$ are classified as $\mathscr{C}_1^{(train)}$, both test classes are assigned to $\mathscr{C}_1^{(train)}$, resulting in duplication. Conversely, if $c_1^{(test)}$ is classified as $\mathscr{C}_2^{(train)}$ and $c_2^{(test)}$ as $\mathscr{C}_1^{(train)}$, they are assigned without duplication.

To validate, we introudce treatment and control groups. For treatment group, we eliminate duplicate in the textitassignments for the train classes, i.e. , for each unseen class, the most frequently classified training class is aggregated, and the classes are randomly removed to ensure that the selected training classes become unique (2). For the control group, we performed random selection of the same number of classes of treatment group (1). These two groups are evaluated using *recall@1*. This process was repeated five times, and the average was reported. The experimental results are presented in 34, 35, 36, 37, Table 3, and 4. A total of 64 experiments are conducted, of which 51 demonstrated performance improvements: the estimated success rate is 79%. There is a $1.73\% \pm 2.87\%$ average improvement in recall@1, with a maximum improvement of 13.65%, a minimum decrease of -3.28%. These findings suggest that the duplicate reduction treatment group outperforms the randomly removed group with a binomial test p-value of $9.40 \times 10^{-7}$.

# F. Limitations and Future Work

While our study provides valuable insights into feature learning and transferability, several important directions remain for future research. First, while the Hermite approximation aided our feature analysis, it posed numerical challenges due to the discrepancy between polynomials and nonlinear neural networks. Specifically, the need for extremely high-dimensional approximations Figure 2 and the lack of precise scaling alignment between the approximation and the neural networks in

---

**Algorithm 1** Random Sampling

---

**Input:** Number if unseen classes $u$, number of classes $|L|$
**Output:** Sampled class set $S_{\text{random}}$
Set $S_{\text{random}} \leftarrow \text{random.sample}(\{0, 1, \dots, u-1\}, |L|)$
**return** $S_{\text{random}}$

---

---

**Algorithm 2** Duplicated *assignment* reduction sampling

---

**Input:** Model $f$, unseen data loader $\mathcal{D}$, number of train classes $C_{\text{train}}$, number of unseen classes $C_{\text{unseen}}$
**Output:** Sampled class set $S_{\text{nondup}}$
Initialize counter matrix $\texttt{counter} \leftarrow \mathbf{0}^{C_{\text{unseen}} \times C_{\text{train}}}$
**for** $(\texttt{img}, \texttt{label})$ in $\mathcal{D}$ **do**
  $\texttt{pred} \leftarrow f(\texttt{img})$                                       *Predicted class indices*
  Update counter: $\texttt{counter}[\texttt{label}, \texttt{pred}] \mathrel{+}= 1$
**end for**
$\texttt{top1\_index} \leftarrow \text{argsort}(\texttt{counter}, \dim = 1, \text{descending} = \texttt{True})[..., 0]$
$\texttt{unique\_label} \leftarrow \text{unique}(\texttt{top1\_index})$
Initialize $S_{\text{nondup}} \leftarrow \emptyset$
**for** each label $\ell$ in $\texttt{unique\_label}$ **do**
  $I_\ell \leftarrow \{i \mid \texttt{top1\_index}[i] = \ell\}$               *Indices corresponding to label $\ell$*
  $i_{\text{sample}} \leftarrow \text{random.sample}(I_\ell, 1)$             *Select one random index*
  $S_{\text{nondup}} \leftarrow S_{\text{nondup}} \cup \{i_{\text{sample}}\}$
**end for**
**return** $S_{\text{nondup}}$

---

finite dimensions Figure 4.

These limitations highlight the need for alternative approximation techniques or analytical approaches. Second, the relationship between semantic similarity and train-unseen similarity requires further theoretical exploration. Third, an important direction for future research is expanding the concepts of cohesion and Separability to multi-class softmax classification problems, incorporating normalization and temperature scaling to better align with practical settings or Neural Collapse research. Finally, recently Zavatone-Veth et al. (2023) suggest neural networks tend to compress the feature space around training data while expanding the regions between decision boundaries. We consider this phenomenon appears closely related to the train-unseen similarity-driven cohesion and Separability observed in our study. Investigating this connection through the lens of Riemannian geometry could yield novel insights into the fundamental structure of learned representations.

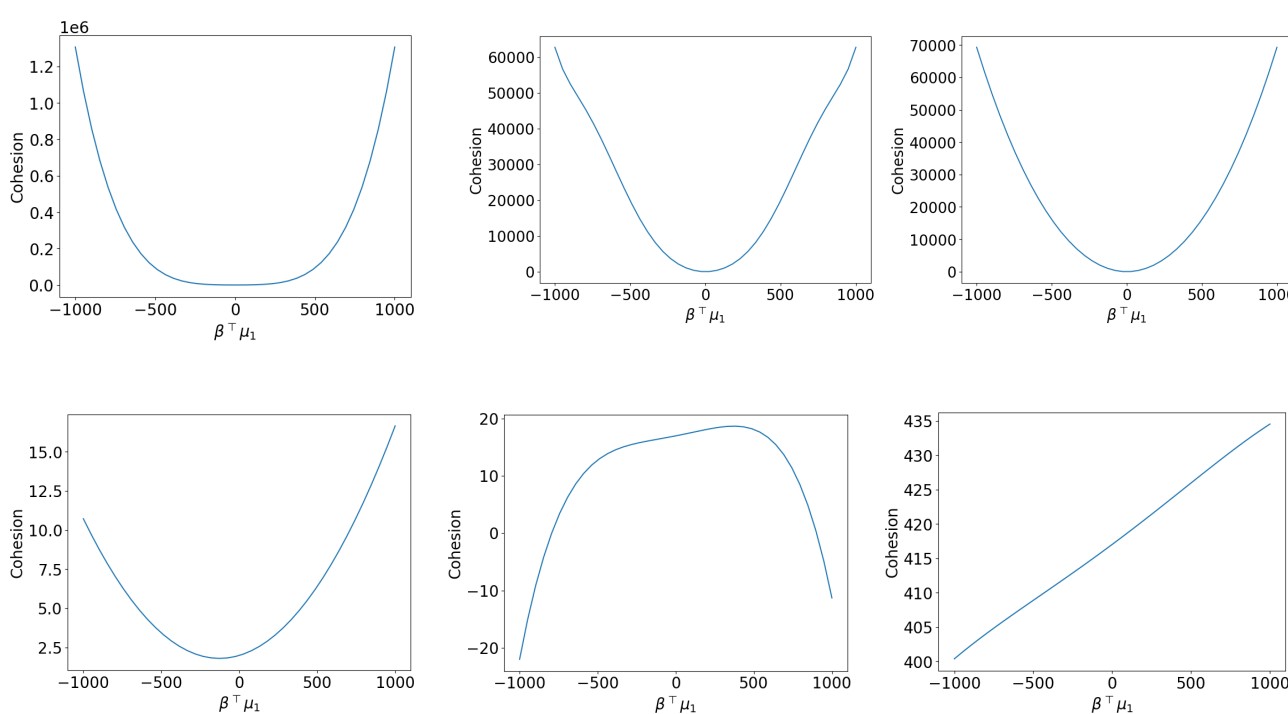

Figure 16: Component analysis of Cohesion in $\mathbb{R}^{2000}, \mathbb{R}^{20000}, \mathbb{R}^{320000}$ (left to right) in range $[-1000, 1000]$, top: the dominant last component, bottom: sum of the other terms.

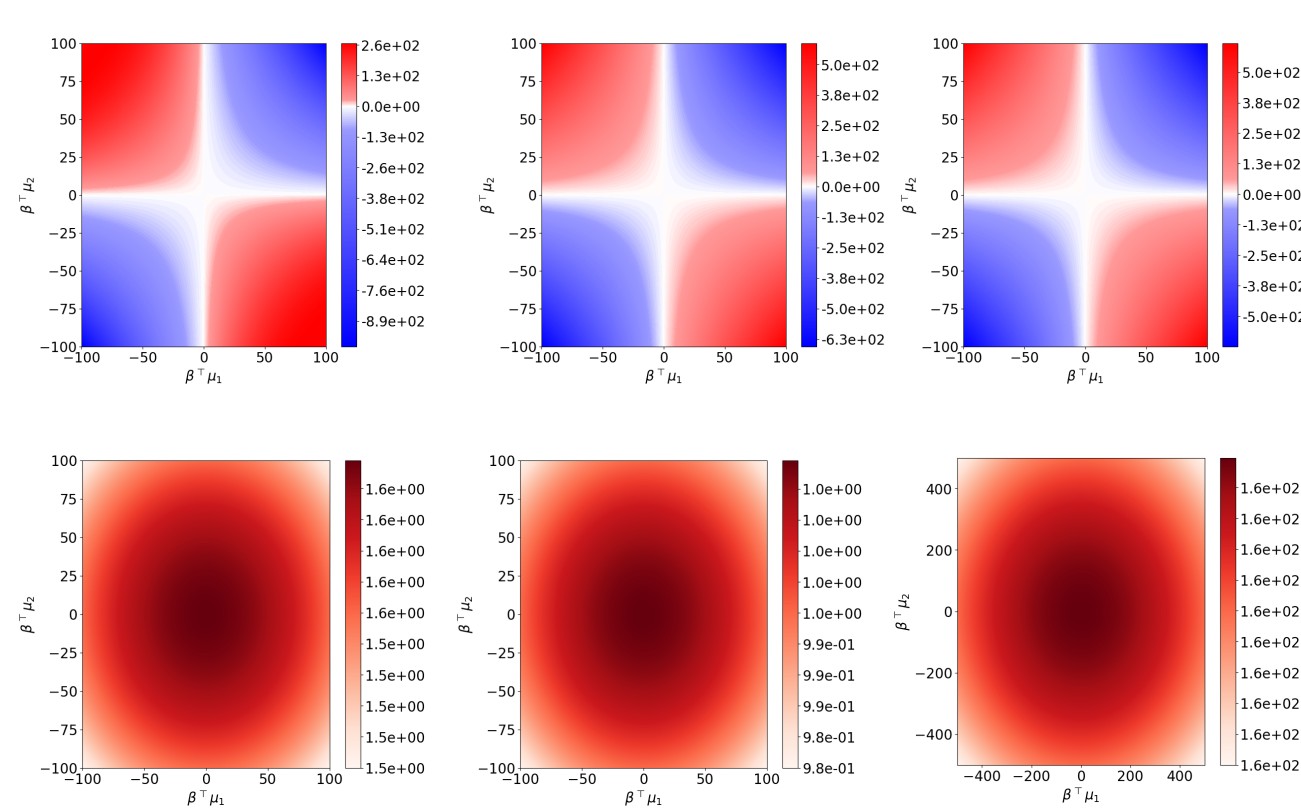

Figure 17: Component analysis of Separability in $\mathbb{R}^{2000}, \mathbb{R}^{20000}, \mathbb{R}^{320000}$ (left to right) in range $[-500, 500]$, top: the dominant last component, bottom: sum of the other terms.

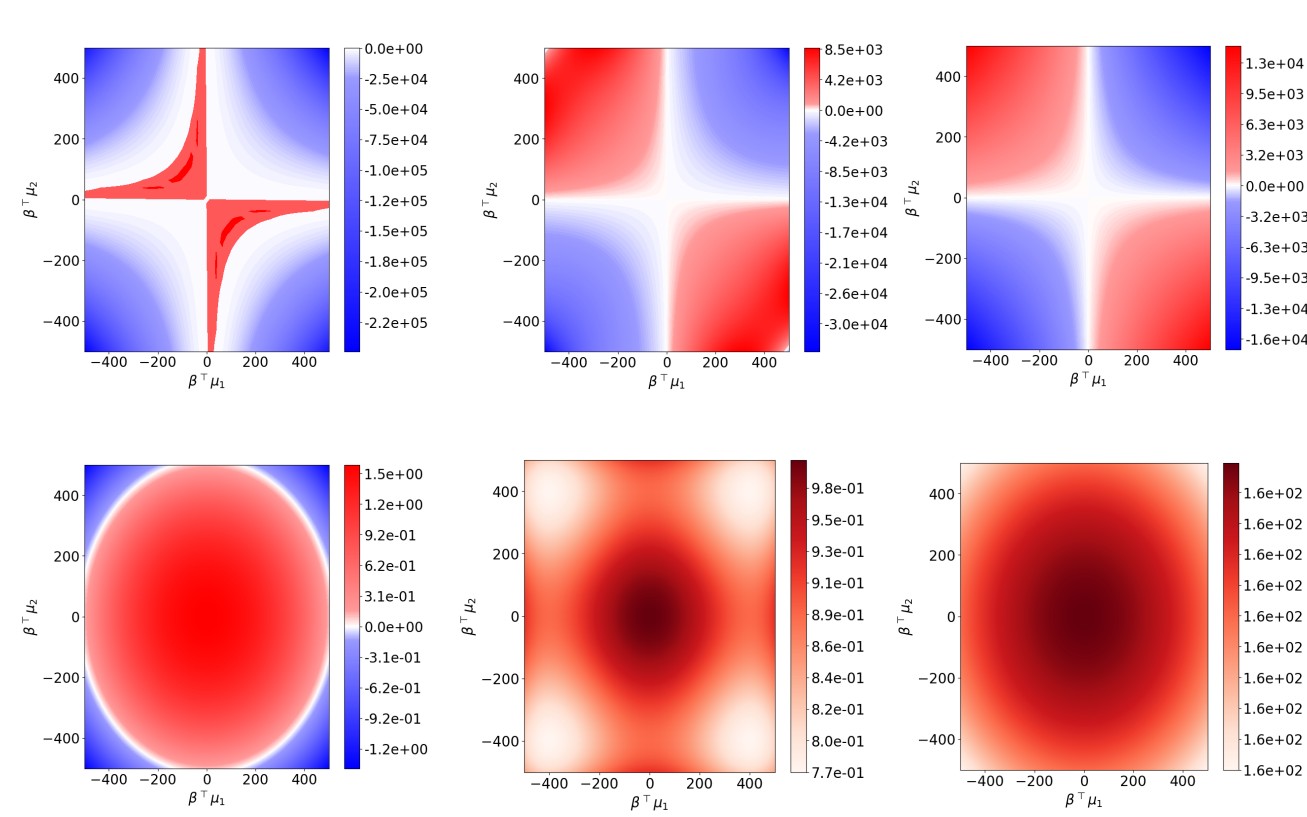

Figure 18: Component analysis of Separability in $\mathbb{R}^{2000}, \mathbb{R}^{20000}, \mathbb{R}^{320000}$ (left to right) in range $[-500, 500]$, top: the dominant last component, bottom: sum of the other terms.

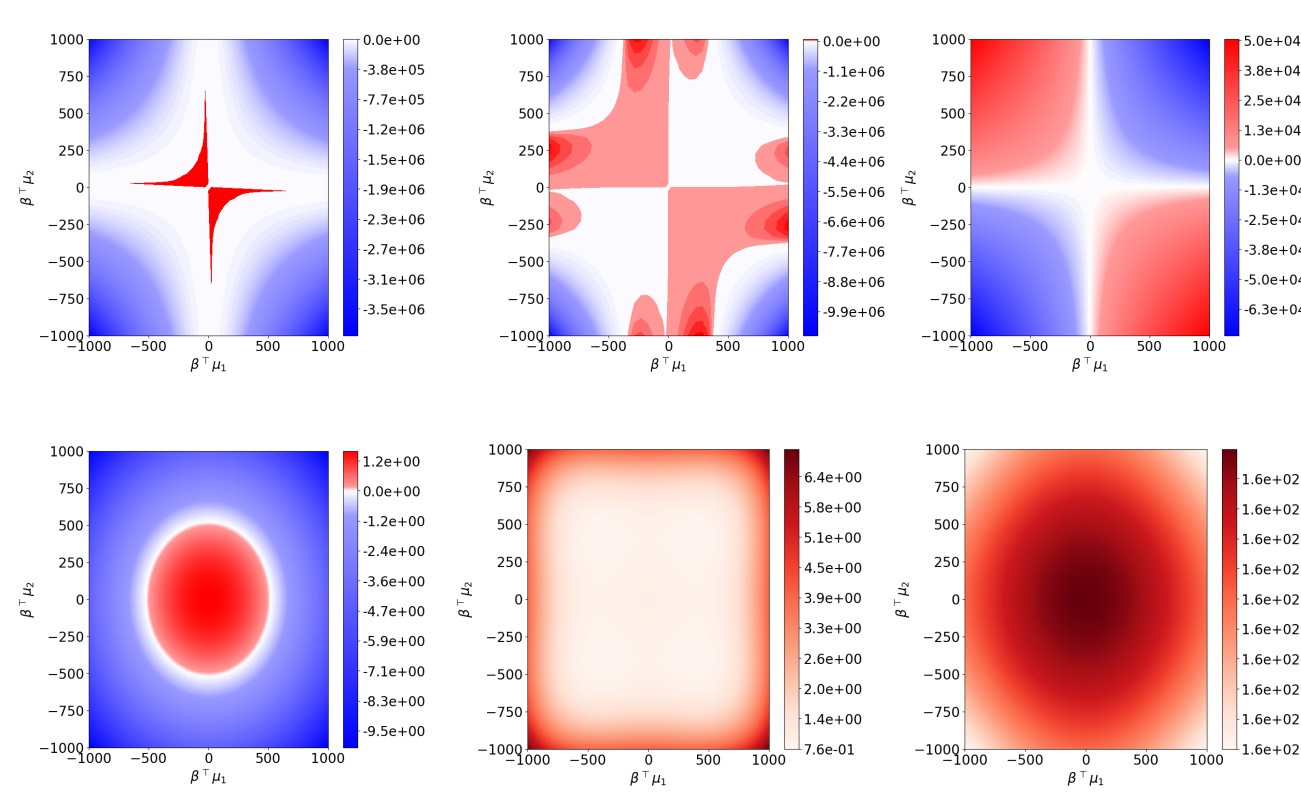

Figure 19: Component analysis of Separability in $\mathbb{R}^{2000}, \mathbb{R}^{20000}, \mathbb{R}^{320000}$ (left to right) in range $[-1000, 1000]$, top: the dominant last component, bottom: sum of the other terms.

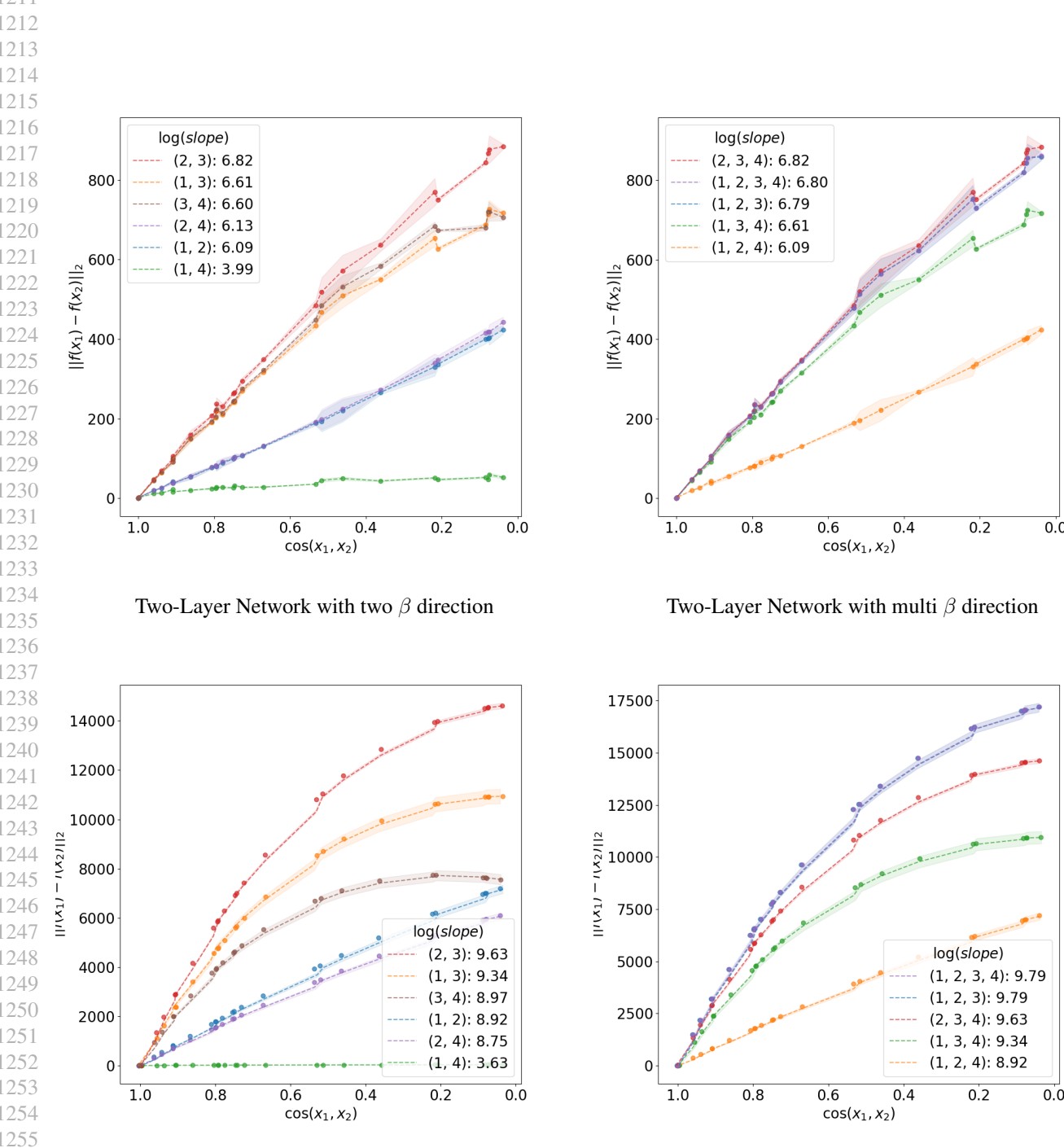

Two-Layer Network with two $\beta$ direction

Two-Layer Network with multi $\beta$ direction

Theory estimation with two $\beta$ direction

Figure 20: Theory estimation with multi $\beta$ direction

Figure 21: Extra results of subsection 4.2 experiments for multiple $\beta_i$ direction

$F_0^L$ with single $\beta$ direction $\qquad$ $F_0^L$ with two $\beta$ direction $\qquad$ $F_0^L$ with multi $\beta$ direction

spike$_L$ with single $\beta$ direction $\qquad$ spike$_L$ with two $\beta$ direction $\qquad$ spike$_L$ with multi $\beta$ direction

Figure 22: Extra results of subsection 4.2 experiments for seperate term $F_0^L$, and spike$_L$.

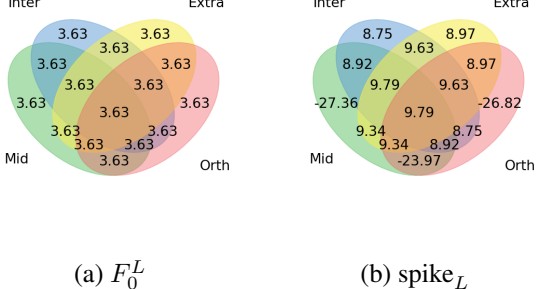

(a) $F_0^L$ $\qquad\qquad$ (b) spike$_L$

Figure 23: Comparison of log average slope between $F_0^L$, and spike$_L$. ■ Midpoint ($\beta_1$) ■ Interpolation ($\beta_2$) ■ Extrapolation ($\beta_3$) ■ Orthogonal ($\beta_4$) $F_0^L$ is not influenced by spikes and generates random features in all cases. spike$_L$ is influenced only by spikes, so when using only the $\beta_1$ or $\beta_4$ spikes, the two features are always mapped to the same position.

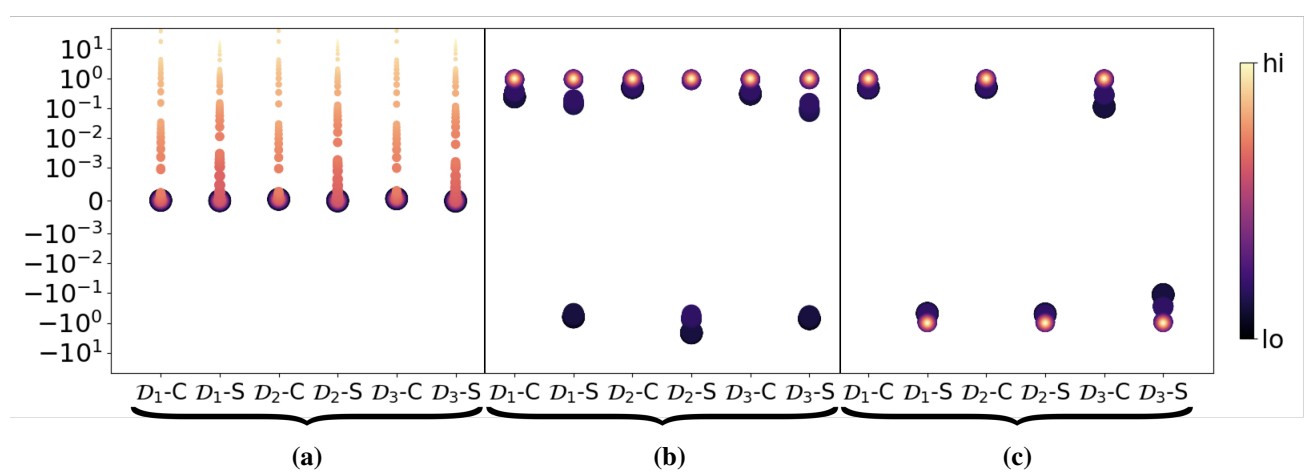

Figure 24: Summary of the synthetic data experiments: The large and dark circles represent low *train-unseen similarity*, while the small and light circles indicate high *train-unseen similarity*. The datasets $\mathcal{D}_1$, $\mathcal{D}_2$, and $\mathcal{D}_3$ correspond to synthetic Data 1, 2, and 3, respectively. C denotes *Cohesion*, and S denotes *Separability*. In panels (a) and (b), the two unseen classes are *assigned* to different training classes (i.e., a positive-negative), and as the *train-unseen similarity* increases, both *Separability* and *Cohesion* increase accordingly. In contrast, in panel (c), the two unseen classes are *assigned* to the same training class (i.e., a positive-positive), leading to a decrease in *Separability*. These observations are consistent with our theoretical predictions. We scaled all measurement using the absolute value at the 85th percentile.

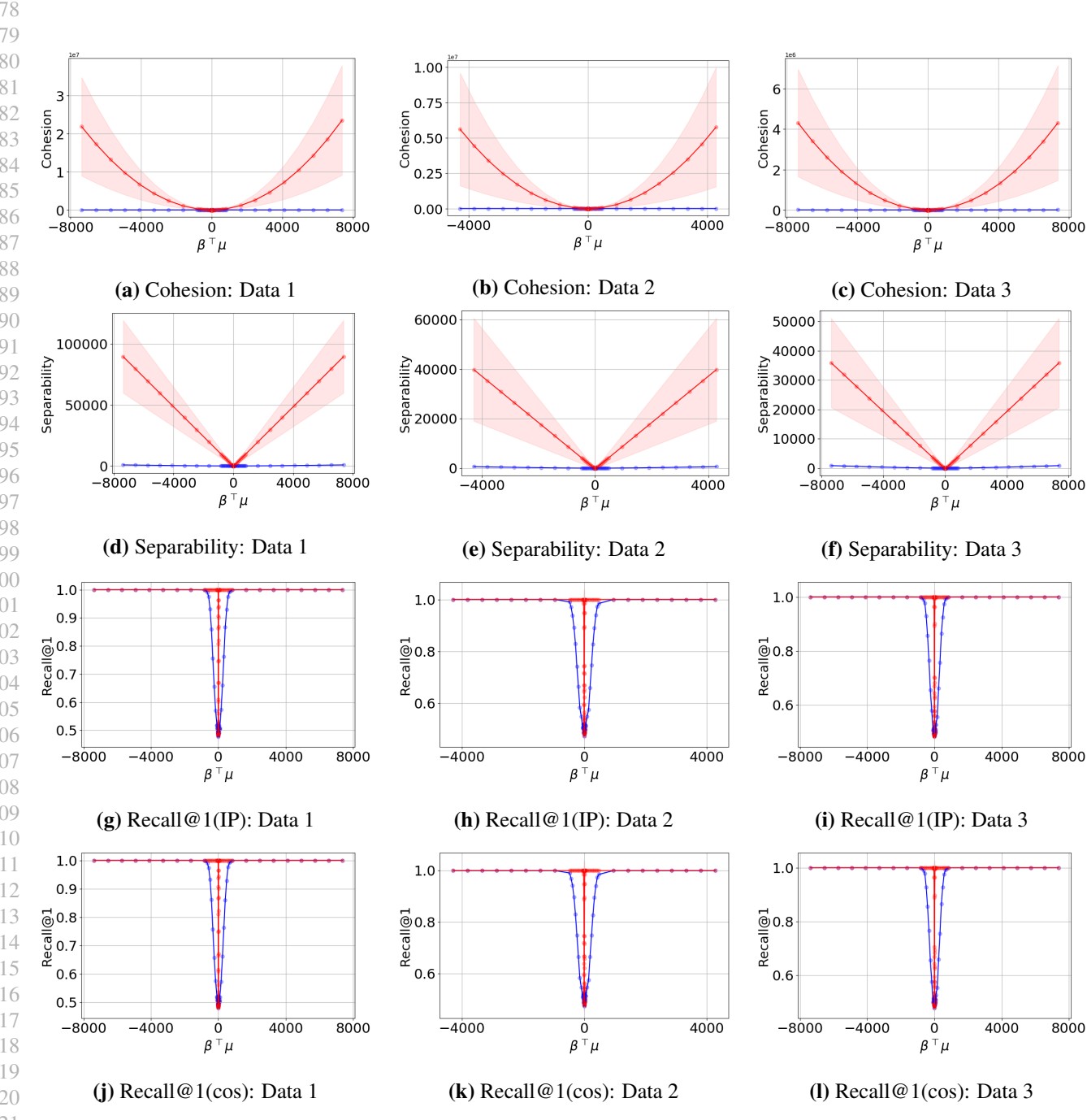

Figure 25: Expr. I: translation($e$) variation case (linear scale). — is after one step training. — is from initialization. As the *train-unseen similarity* increases, both cohesion and Separability become larger due to pos-neg setup.

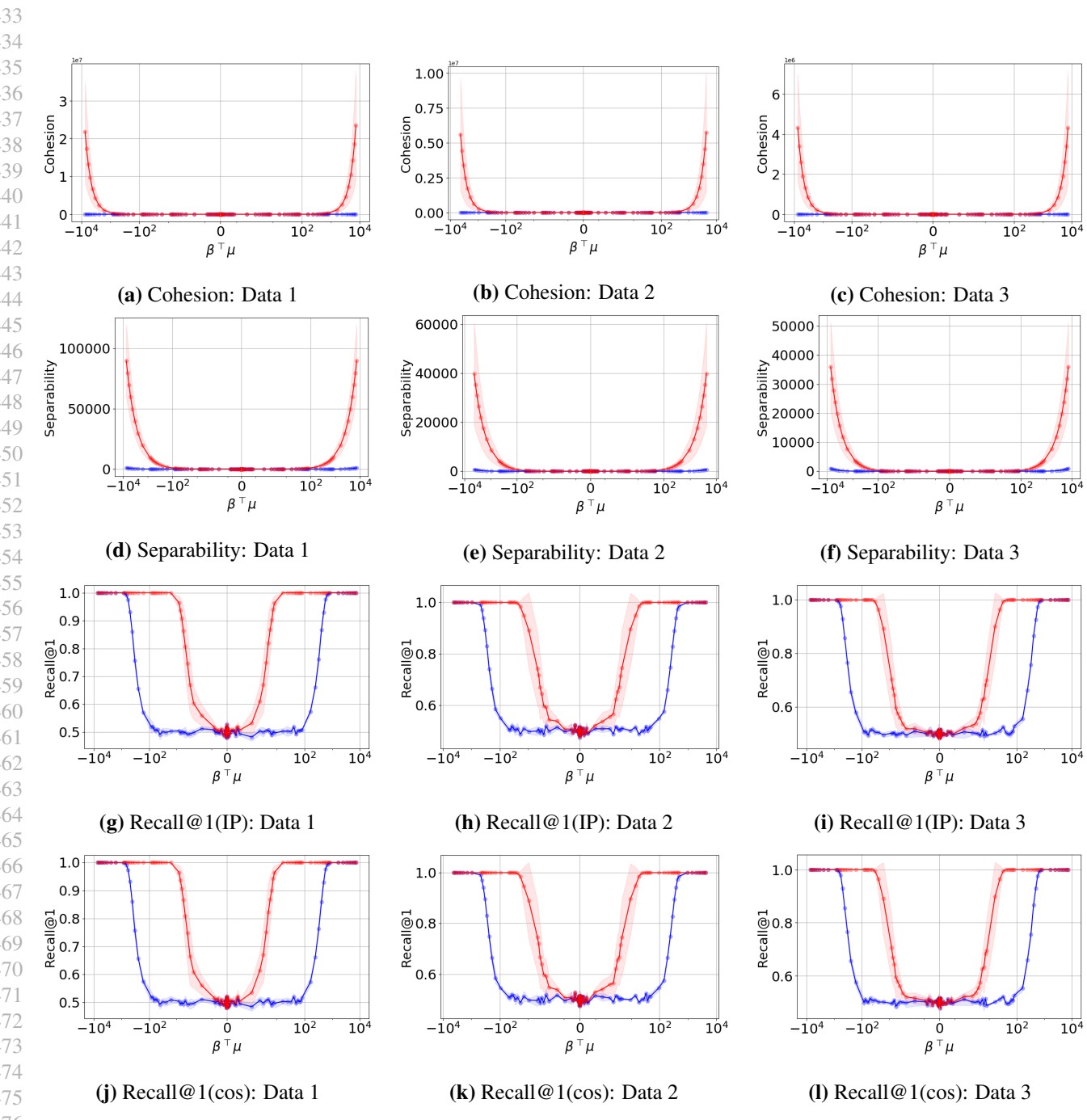

**(a)** Cohesion: Data 1      **(b)** Cohesion: Data 2      **(c)** Cohesion: Data 3

**(d)** Separability: Data 1      **(e)** Separability: Data 2      **(f)** Separability: Data 3

**(g)** Recall@1(IP): Data 1      **(h)** Recall@1(IP): Data 2      **(i)** Recall@1(IP): Data 3

**(j)** Recall@1(cos): Data 1      **(k)** Recall@1(cos): Data 2      **(l)** Recall@1(cos): Data 3

Figure 26: Expr. I: translation($e$) variation (log scale). — is after one step training. — is from initialization. As the *train-unseen similarity* increases, both cohesion and Separability become larger due to pos-neg setup.

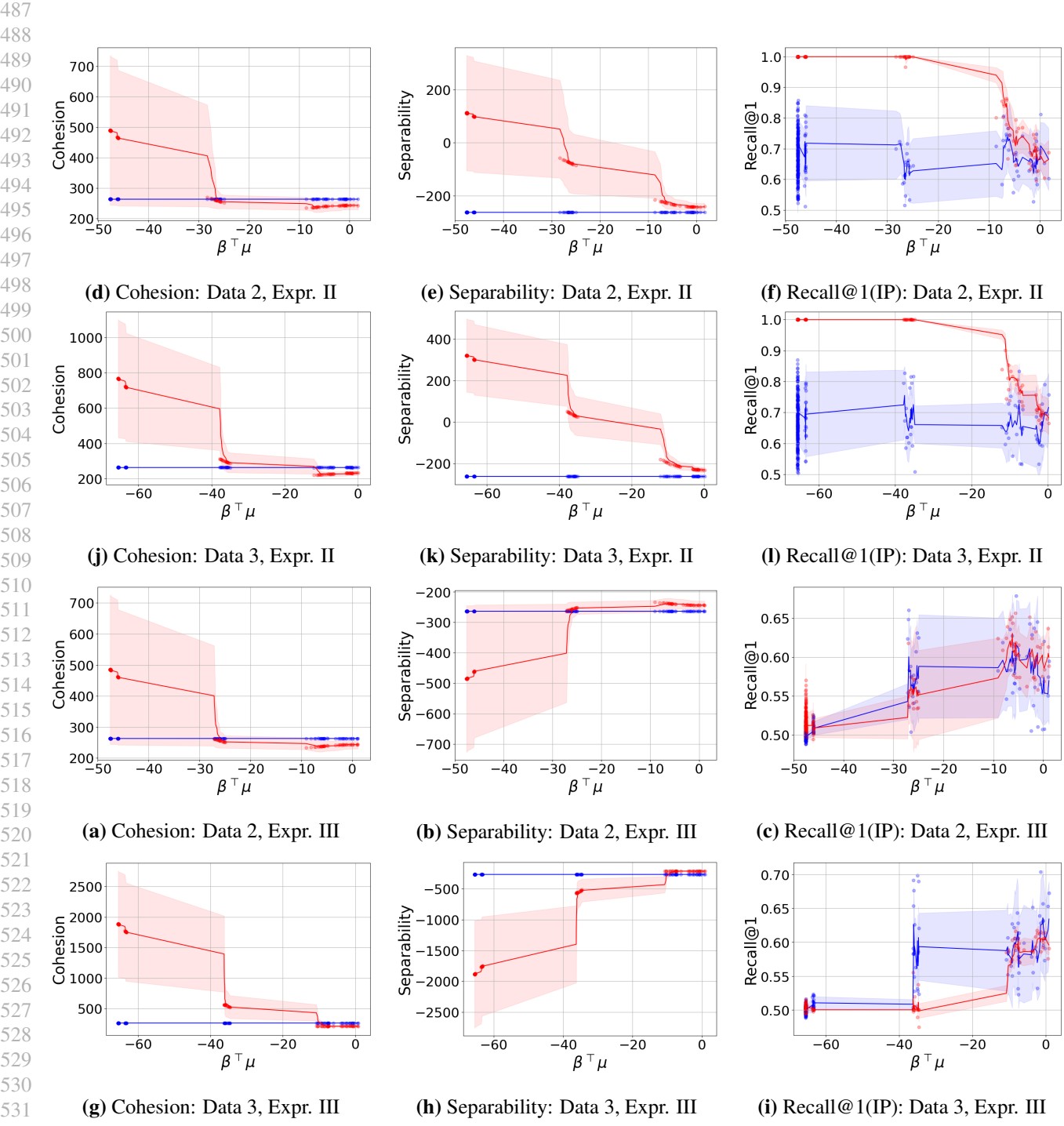

**(d)** Cohesion: Data 2, Expr. II    **(e)** Separability: Data 2, Expr. II    **(f)** Recall@1(IP): Data 2, Expr. II

**(j)** Cohesion: Data 3, Expr. II    **(k)** Separability: Data 3, Expr. II    **(l)** Recall@1(IP): Data 3, Expr. II

**(a)** Cohesion: Data 2, Expr. III    **(b)** Separability: Data 2, Expr. III    **(c)** Recall@1(IP): Data 2, Expr. III

**(g)** Cohesion: Data 3, Expr. III    **(h)** Separability: Data 3, Expr. III    **(i)** Recall@1(IP): Data 3, Expr. III

Figure 27: Expr. II, Expr. III: rotation($R$) variation (linear scale). — is after one step training. — is from initialization. Expr. II is pos-neg. Expr. III is pos-pos.

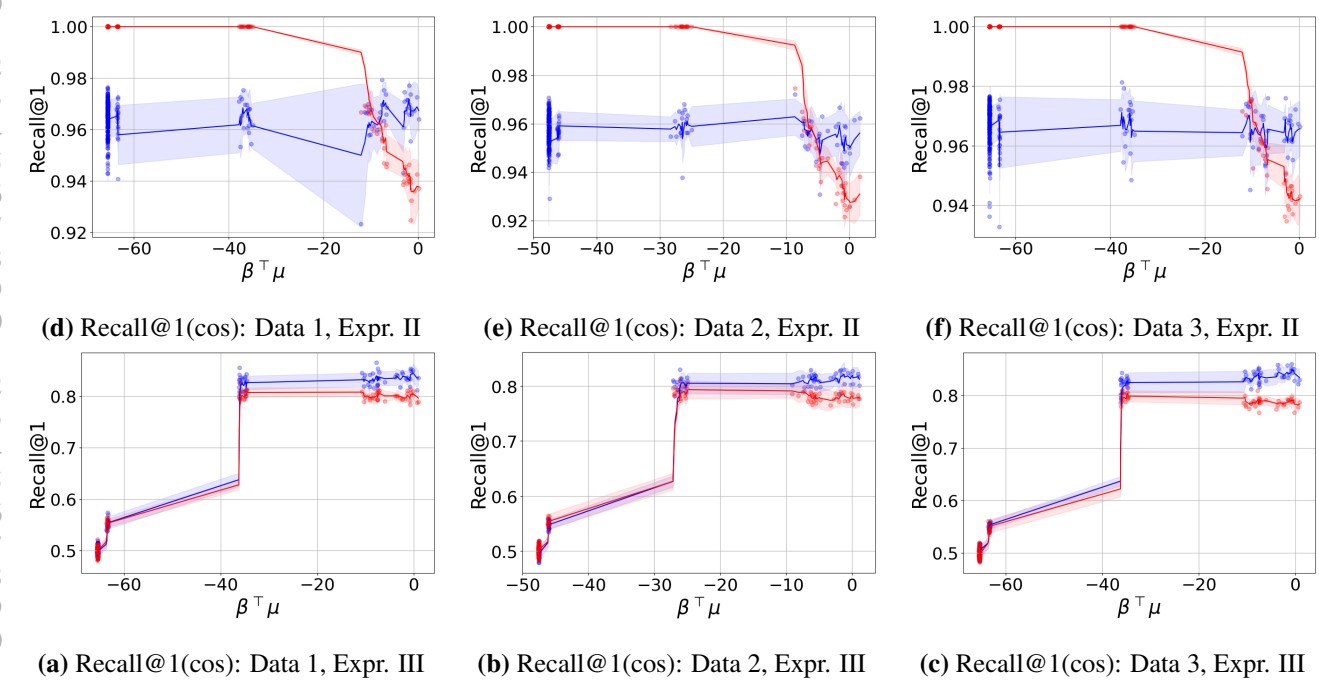

**(d)** Recall@1(cos): Data 1, Expr. II    **(e)** Recall@1(cos): Data 2, Expr. II    **(f)** Recall@1(cos): Data 3, Expr. II

**(a)** Recall@1(cos): Data 1, Expr. III    **(b)** Recall@1(cos): Data 2, Expr. III    **(c)** Recall@1(cos): Data 3, Expr. III

Figure 28: Recall@1 with cosine similarity of Expr. II, Expr. III: rotation($R$) variation (linear scale). —— is after one step training. —— is from initialization. Expr. II is pos-neg. Expr. III is pos-pos.

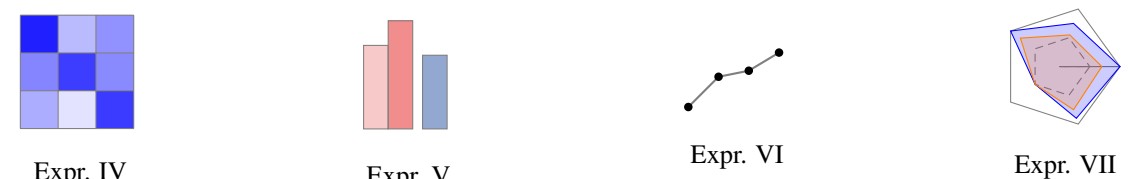

Expr. IV          Expr. V          Expr. VI          Expr. VII

Figure 29: Expr. IV: High clustering performance with same train-unseen domain. Expr. V: Extra unrelated training classes do not affect *recall@1* performance. Expr. VI: Extra related training classes improve *recall@1* performance. Expr. VII: Removing duplicately *assigned* eval classes improves performance over random removal.

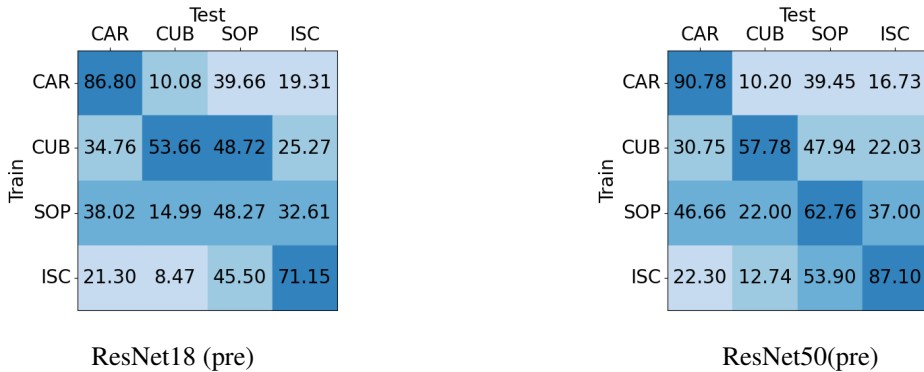

ResNet18 (pre)          ResNet50(pre)

Figure 30: Expr. IV on ResNet18 with *Domain* datasets (CAR, CUB, SOP, ISC)

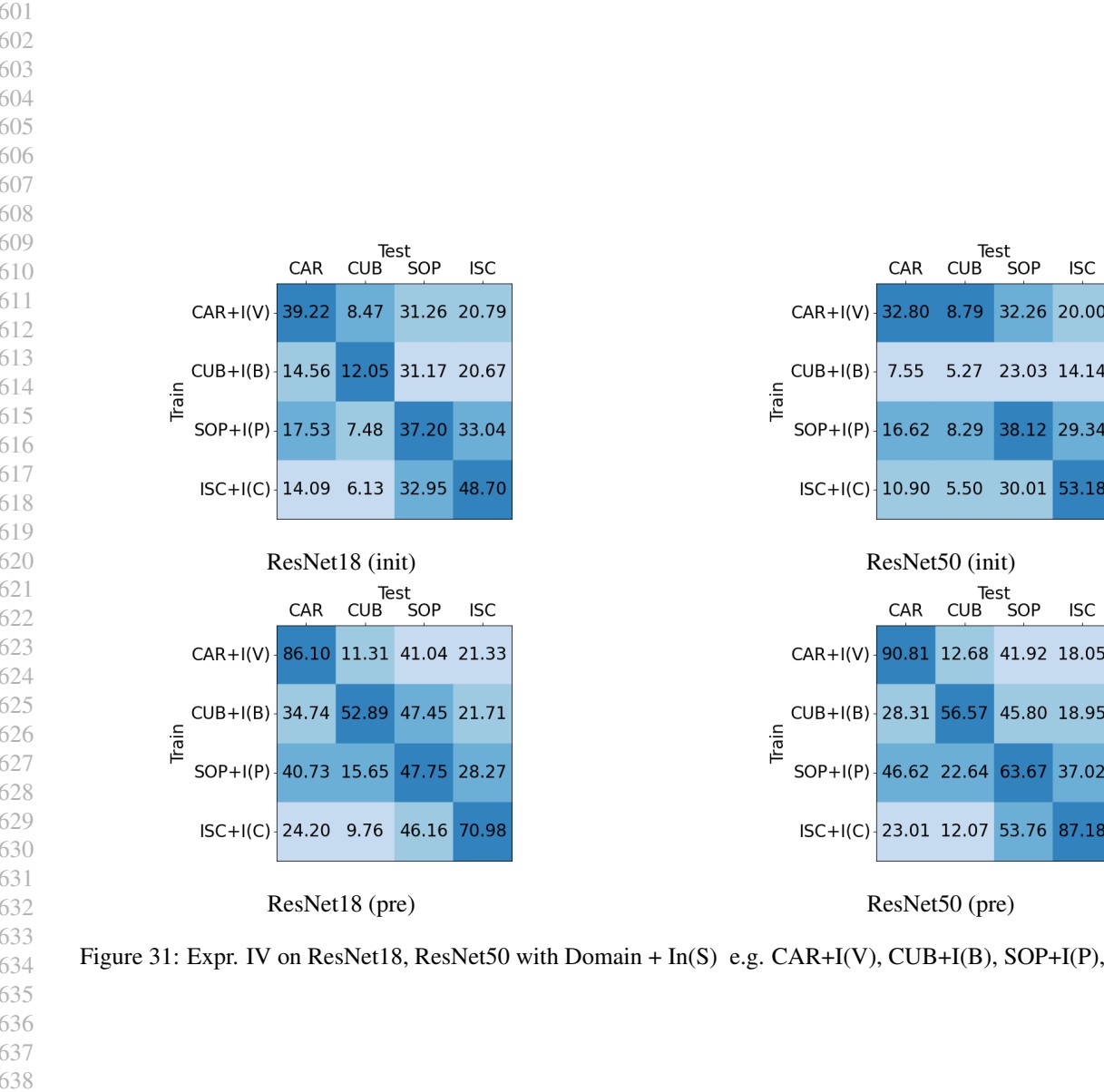

Figure 31: Expr. IV on ResNet18, ResNet50 with Domain + In(S)  e.g. CAR+I(V), CUB+I(B), SOP+I(P), ISC+I(C)

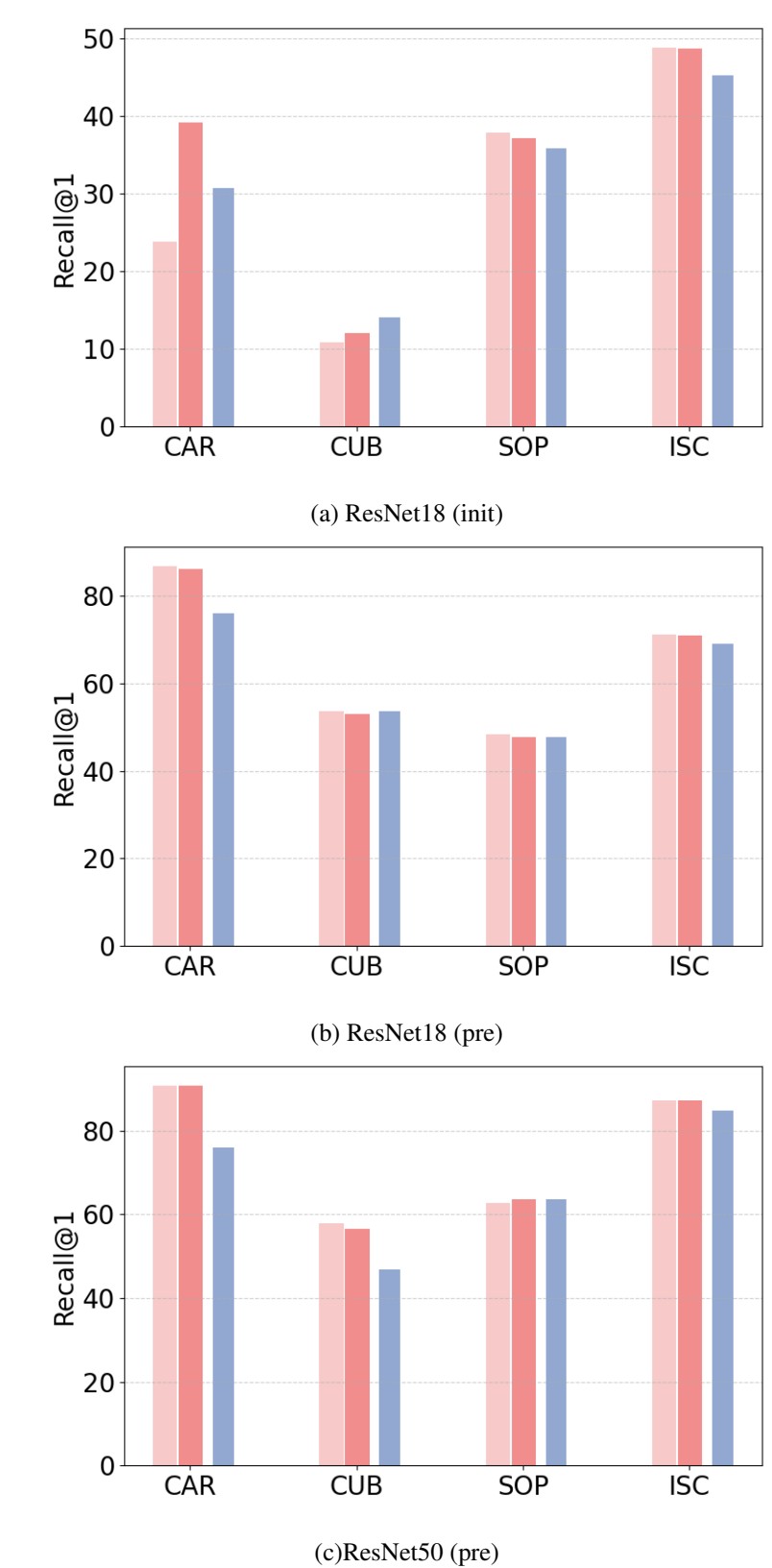

(a) ResNet18 (init)

(b) ResNet18 (pre)

(c)ResNet50 (pre)

Figure 32: Expr. V, additional results, it is represented as follows ▬ Domain ▬ Domain + Related Subset of In1k ▬ Domain + Whole In1k subsampled Adding unrelated classes for training does not significantly affect the performance.

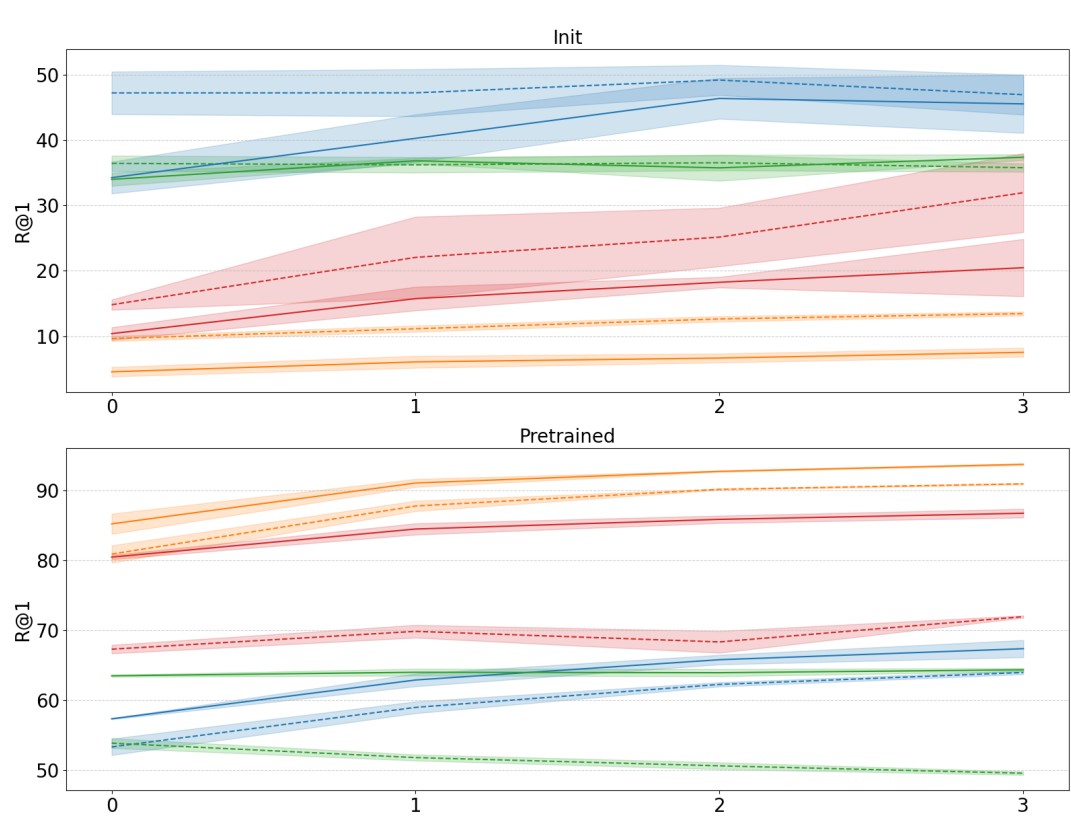

Figure 33: Expr VI, it is represented as follows: ResNet18 - - , ResNet50 —, Dataset car, cub, sop, isc. As the steps increased and related classes were added, performance generally improved consistently.

Figure 34: Expr. VII, ResNet18 (Init), depending on the experimental setup, there are three cases: ▬ removing redundancy, ▬ randomly selecting the same number of classes as those with redundancy removed, and ▬ using all test classes. For dataset we use, we denote as 'Train data(Test data)'. 'In' denote using whole classes of subsampled ImageNet.

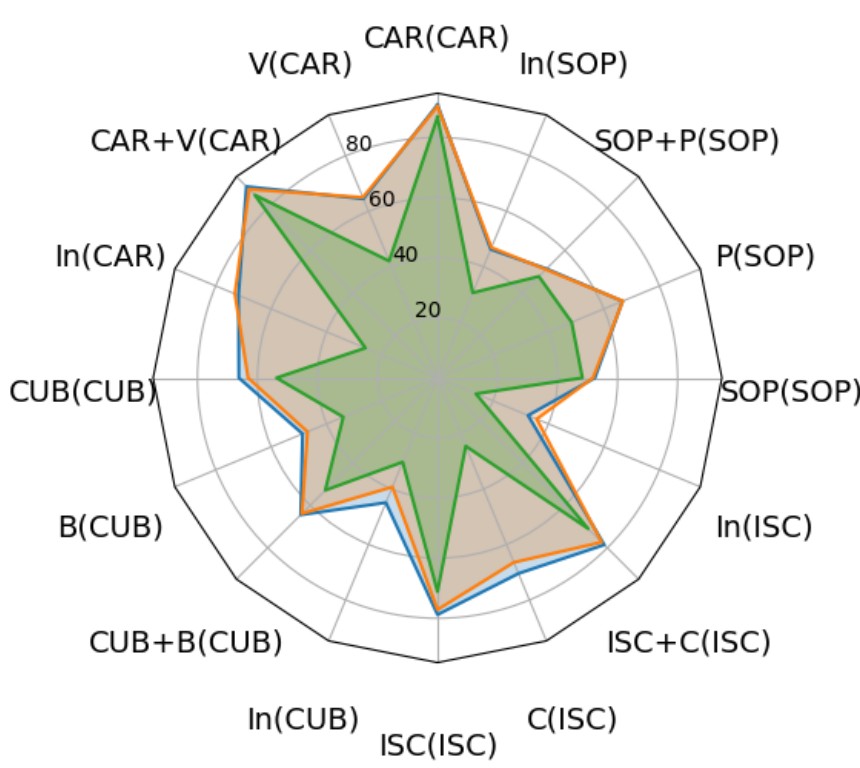

Figure 35: Expr. VII, ResNet18 (Pre), depending on the experimental setup, there are three cases: ▬ removing redundancy, ▬ randomly selecting the same number of classes as those with redundancy removed, and ▬ using all test classes. For dataset we use, we denote as 'Train data(Test data)'. 'In' denote using whole classes of subsampled ImageNet.

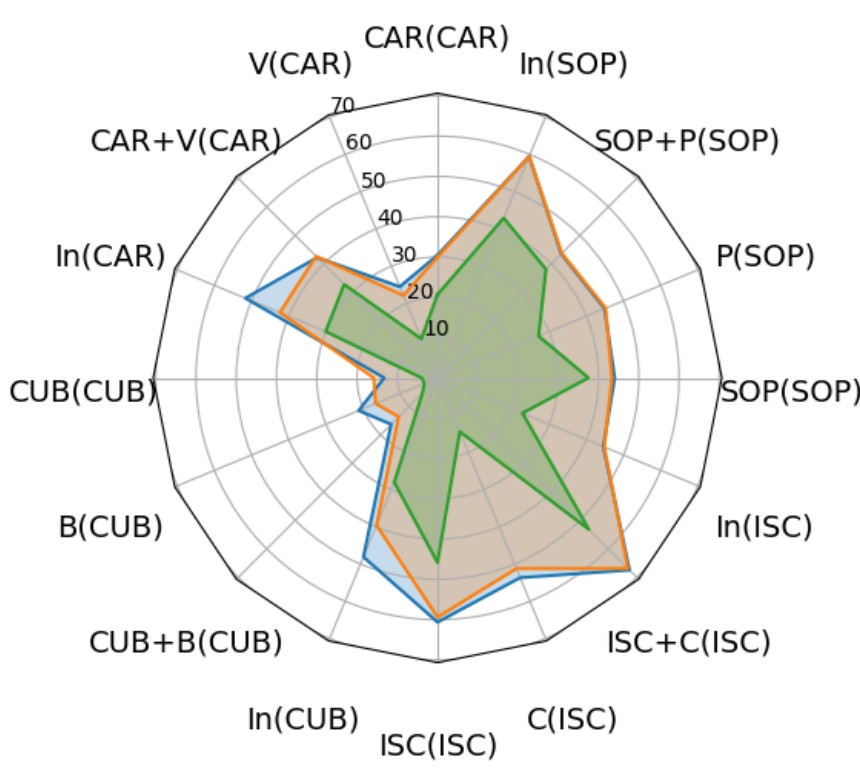

Figure 36: Expr. VII, ResNet50 (Init), depending on the experimental setup, there are three cases: ▬ removing redundancy, ▬ randomly selecting the same number of classes as those with redundancy removed, and ▬ using all test classes. For dataset we use, we denote as 'Train data(Test data)'. 'In' denote using whole classes of subsampled ImageNet.

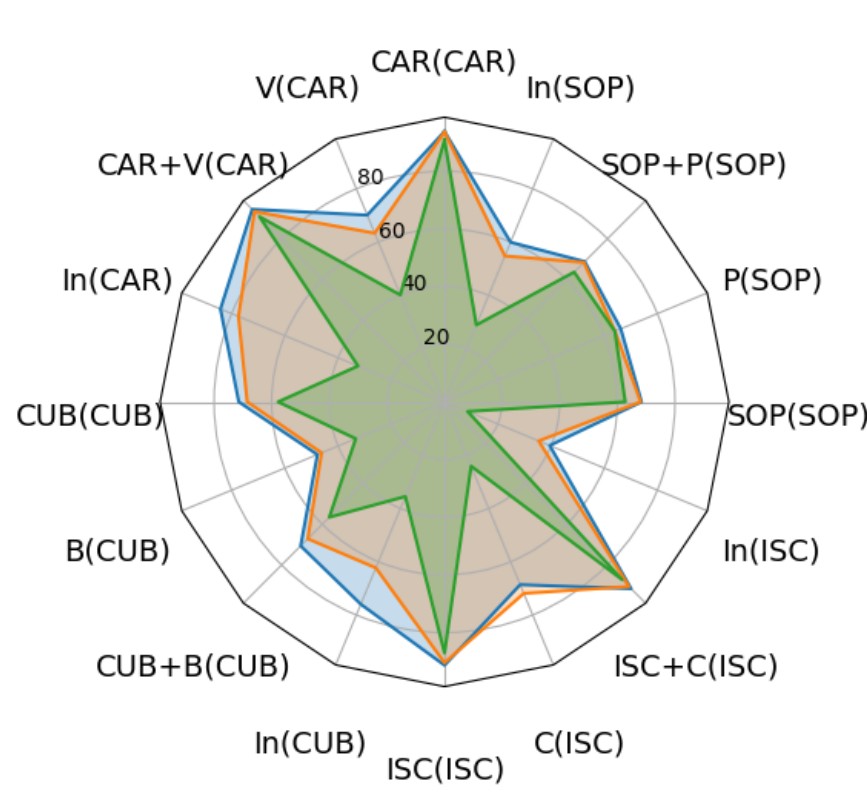

Figure 37: Expr. VII, ResNet50 (Pre), depending on the experimental setup, there are three cases: ▬ removing redundancy, ▬ randomly selecting the same number of classes as those with redundancy removed, and ▬ using all test classes. For dataset we use, we denote as 'Train data(Test data)'. 'In' denote using whole classes of subsampled ImageNet.

Table 1: Table results for Expr. IV

ResNet18 (Randomly Initialized)

|            | CAR    | CUB    | SOP    | ISC    |
|------------|--------|--------|--------|--------|
| CAR+I(V)   | **0.3922** | 0.0847 | 0.3126 | 0.2079 |
| CAR        | 0.2383 | 0.0685 | 0.2766 | 0.1994 |
| I(V)       | 0.1117 | 0.0618 | 0.2610 | 0.1793 |
| CUB+I(B)   | 0.1456 | **0.1205** | 0.3117 | 0.2067 |
| CUB        | 0.1432 | 0.1089 | 0.3179 | 0.1998 |
| I(B)       | 0.0973 | 0.0640 | 0.2658 | 0.1703 |
| SOP+I(P)   | 0.1753 | 0.0748 | 0.3720 | 0.3304 |
| SOP        | 0.1754 | 0.0876 | **0.3790** | 0.3306 |
| I(P)       | 0.1405 | 0.0586 | 0.3129 | 0.2327 |
| ISC+I(C)   | 0.1409 | 0.0613 | 0.3295 | 0.4870 |
| ISC        | 0.1328 | 0.0685 | 0.3338 | **0.4887** |
| I(C)       | 0.0908 | 0.0471 | 0.2485 | 0.1823 |

ResNet50 (Randomly Initialized)

|            | CAR    | CUB    | SOP    | ISC    |
|------------|--------|--------|--------|--------|
| CAR+I(V)   | **0.3280** | **0.0879** | 0.3226 | 0.2000 |
| CAR        | 0.2067 | 0.0495 | 0.2611 | 0.1583 |
| I(V)       | 0.1048 | 0.0459 | 0.2670 | 0.1410 |
| CUB+I(B)   | 0.0755 | 0.0527 | 0.2303 | 0.1414 |
| CUB        | 0.0626 | 0.0393 | 0.1950 | 0.1081 |
| I(B)       | 0.0456 | 0.0358 | 0.1954 | 0.1074 |
| SOP+I(P)   | 0.1662 | 0.0829 | **0.3812** | 0.2934 |
| SOP        | 0.1725 | 0.0743 | 0.3750 | 0.2754 |
| I(P)       | 0.0940 | 0.0422 | 0.2716 | 0.1697 |
| ISC+I(C)   | 0.1090 | 0.0550 | 0.3001 | **0.5318** |
| ISC        | 0.1022 | 0.0503 | 0.2699 | 0.4581 |
| I(C)       | 0.0625 | 0.0412 | 0.2294 | 0.1446 |

ResNet18 (ImageNet 1K Pretrained)

|            | CAR    | CUB    | SOP    | ISC    |
|------------|--------|--------|--------|--------|
| CAR+I(V)   | 0.8610 | 0.1131 | 0.4104 | 0.2133 |
| CAR        | **0.8680** | 0.1008 | 0.3966 | 0.1931 |
| I(V)       | 0.4210 | 0.1698 | 0.4618 | 0.2507 |
| CUB+I(B)   | 0.3474 | 0.5289 | 0.4745 | 0.2171 |
| CUB        | 0.3476 | **0.5366** | 0.4872 | 0.2527 |
| I(B)       | 0.3771 | 0.3400 | **0.5062** | 0.2278 |
| SOP+I(P)   | 0.4073 | 0.1565 | 0.4775 | 0.2827 |
| SOP        | 0.3802 | 0.1499 | 0.4827 | 0.3261 |
| I(P)       | 0.4003 | 0.2076 | 0.4838 | 0.2569 |
| ISC+I(C)   | 0.2420 | 0.0976 | 0.4616 | 0.7098 |
| ISC        | 0.2130 | 0.0847 | 0.4550 | **0.7115** |
| I(C)       | 0.3738 | 0.2227 | 0.4994 | 0.2457 |

ResNet50 (ImageNet 1K Pretrained)

|            | CAR    | CUB    | SOP    | ISC    |
|------------|--------|--------|--------|--------|
| CAR+I(V)   | **0.9081** | 0.1268 | 0.4192 | 0.1805 |
| CAR        | 0.9078 | 0.1020 | 0.3945 | 0.1673 |
| I(V)       | 0.4013 | 0.1648 | 0.4815 | 0.2330 |
| CUB+I(B)   | 0.2831 | 0.5657 | 0.4580 | 0.1895 |
| CUB        | 0.3075 | **0.5778** | 0.4794 | 0.2203 |
| I(B)       | 0.3212 | 0.3337 | 0.4781 | 0.1846 |
| SOP+I(P)   | 0.4662 | 0.2264 | **0.6367** | 0.3702 |
| SOP        | 0.4666 | 0.2200 | 0.6276 | 0.3700 |
| I(P)       | 0.3547 | 0.2208 | 0.4602 | 0.2337 |
| ISC+I(C)   | 0.2301 | 0.1207 | 0.5376 | **0.8718** |
| ISC        | 0.2230 | 0.1274 | 0.5390 | 0.8710 |
| I(C)       | 0.3655 | 0.2311 | 0.5167 | 0.2413 |

Table 2: Table results of performance for Expr. V.

ResNet18 (Randomly Initialized)

|          | CAR    | CUB    | SOP    | ISC    |
|----------|--------|--------|--------|--------|
| D        | 0.2383 | 0.1089 | **0.3790** | **0.4887** |
| D+I(Sub) | **0.3922** | 0.1205 | 0.3720 | 0.4870 |
| D+I      | 0.3074 | **0.1404** | 0.3591 | 0.4532 |

ResNet50 (Randomly Initialized)

|          | CAR    | CUB    | SOP    | ISC    |
|----------|--------|--------|--------|--------|
| D        | 0.2067 | 0.0393 | **0.3750** | 0.4581 |
| D+I(Sub) | **0.3280** | 0.0527 | 0.3812 | **0.5318** |
| D+I      | 0.3276 | **0.0968** | 0.3726 | 0.4992 |

ResNet18 (ImageNet 1K Pretrained)

|          | CAR    | CUB    | SOP    | ISC    |
|----------|--------|--------|--------|--------|
| D        | **0.8680** | **0.5366** | **0.4827** | **0.7115** |
| D+I(Sub) | 0.8610 | 0.5289 | 0.4775 | 0.7098 |
| D+I      | 0.7604 | 0.5357 | 0.4766 | 0.6897 |

ResNet50 (ImageNet 1K Pretrained)

|          | CAR    | CUB    | SOP    | ISC    |
|----------|--------|--------|--------|--------|
| D        | 0.9078 | **0.5778** | 0.6276 | 0.8710 |
| D+I(Sub) | **0.9081** | 0.5657 | **0.6367** | **0.8718** |
| D+I      | 0.7603 | 0.4689 | 0.6360 | 0.8481 |

Table 3: Expr. VII from (Randomly Initialized)

ResNet18 (Randomly Initialized)

| Test | Train | Treatment | Random | Δ | Total |
|------|-------|-----------|--------|------|-------|
| CAR Test | CAR | 35.26 | 34.49 | 0.77 | 23.83 |
| | I(V) | 29.02 | 26.96 | 2.06 | 11.17 |
| | CAR+I(V) | 49.27 | 46.73 | 2.54 | 39.22 |
| | In | 39.51 | 36.48 | 3.03 | 25.70 |
| CUB Test | CUB | 20.01 | 18.49 | 1.52 | 10.89 |
| | I(B) | 16.36 | 14.75 | 1.61 | 6.40 |
| | CUB+I(B) | 19.58 | 18.39 | 1.19 | 12.05 |
| | In | 32.16 | 28.74 | 3.42 | 21.49 |
| ISC Test | ISC | 60.64 | 59.45 | 1.19 | 48.87 |
| | I(C) | 60.93 | 57.78 | 3.15 | 18.23 |
| | ISC+I(C) | 59.59 | 59.11 | 0.48 | 48.70 |
| | In | 45.01 | 46.92 | -1.91 | 24.75 |
| SOP Test | SOP | 43.58 | 42.96 | 0.62 | 37.90 |
| | I(P) | 49.76 | 48.45 | 1.31 | 31.29 |
| | SOP+I(P) | 42.57 | 42.12 | 0.45 | 37.20 |
| | In | 51.84 | 54.03 | -2.19 | 38.82 |
| Average Improvement | | | | 1.20 | |
| Success Rate | | | | **0.875** | |

ResNet50 (Randomly Initialized)

| Test | Train | Treatment | Random | Δ | Total |
|------|-------|-----------|--------|------|-------|
| CAR Test | CAR | 30.45 | 29.95 | 0.50 | 20.67 |
| | I(V) | 24.49 | 22.16 | 2.33 | 10.48 |
| | CAR+I(V) | 42.25 | 42.67 | -0.42 | 32.80 |
| | In(CAR) | 51.69 | 42.39 | 9.30 | 30.06 |
| CAR Test | CUB | 13.24 | 15.84 | -2.60 | 3.93 |
| | I(B) | 21.20 | 16.65 | 4.55 | 3.58 |
| | CUB+I(B) | 16.30 | 13.66 | 2.64 | 5.27 |
| | In | 48.10 | 39.59 | 8.51 | 28.06 |
| CAR Test | ISC | 60.63 | 59.41 | 1.22 | 45.81 |
| | I(C) | 53.67 | 51.22 | 2.45 | 14.46 |
| | ISC+I(C) | 67.46 | 66.88 | 0.58 | 53.18 |
| | In | 44.60 | 44.85 | -0.25 | 22.85 |
| CAR Test | SOP | 44.02 | 43.34 | 0.68 | 37.50 |
| | I(P) | 44.93 | 45.22 | -0.29 | 27.16 |
| | SOP+I(P) | 43.51 | 43.70 | -0.19 | 38.12 |
| | In | 59.49 | 59.47 | 0.02 | 42.93 |
| Average Improvement | | | | 1.81 | |
| Success Rate | | | | **0.6875** | |

Table 4: Expr. VII (ImageNet 1K Pretrained)

ResNet18 (ImageNet 1K Pretrained)

| Test | Train | Treatment | Random | Δ | Total |
|---|---|---|---|---|---|
| CAR Test | CAR | 90.90 | 90.33 | 0.57 | 86.80 |
| | I(V) | 64.51 | 65.03 | -0.52 | 42.10 |
| | CAR+I(V) | 90.06 | 88.79 | 1.27 | 86.10 |
| | In(CAR) | 71.77 | 73.08 | -1.31 | 26.00 |
| CAR Test | CUB | 66.19 | 63.12 | 3.07 | 53.66 |
| | I(B) | 48.67 | 46.90 | 1.77 | 34.00 |
| | CUB+I(B) | 64.48 | 63.89 | 0.59 | 52.89 |
| | In | 44.95 | 39.30 | 5.65 | 30.32 |
| CAR Test | ISC | 78.81 | 77.15 | 1.66 | 71.15 |
| | I(C) | 70.48 | 66.47 | 4.01 | 24.57 |
| | ISC+I(C) | 78.58 | 77.35 | 1.23 | 70.98 |
| | In | 32.65 | 35.78 | -3.13 | 13.85 |
| CAR Test | SOP | 52.45 | 51.81 | 0.64 | 48.27 |
| | I(P) | 66.72 | 66.81 | -0.09 | 48.38 |
| | SOP+I(P) | 51.34 | 51.01 | 0.33 | 47.75 |
| | In | 46.31 | 46.95 | -0.64 | 30.66 |
| Average Improvement | | | | 0.94 | |
| Success Rate | | | | **0.6875** | |

ResNet50 (ImageNet 1K Pretrained)

| Test | Train | Treatment | Random | Δ | Total |
|---|---|---|---|---|---|
| CAR Test | CAR | 93.78 | 93.57 | 0.21 | 90.77 |
| | I(V) | 70.12 | 63.34 | 6.78 | 40.13 |
| | CAR+I(V) | 94.45 | 93.34 | 1.11 | 90.81 |
| | In(CAR) | 84.20 | 77.43 | 6.77 | 32.51 |
| CAR Test | CUB | 71.44 | 68.51 | 2.93 | 57.78 |
| | I(B) | 47.78 | 46.19 | 1.59 | 33.37 |
| | CUB+I(B) | 70.63 | 67.15 | 3.48 | 56.56 |
| | In | 75.96 | 62.32 | 13.64 | 35.53 |
| CAR Test | ISC | 91.35 | 90.49 | 0.86 | 87.10 |
| | I(C) | 68.62 | 71.90 | -3.28 | 24.13 |
| | ISC+I(C) | 91.60 | 90.59 | 1.01 | 87.18 |
| | In | 39.54 | 35.39 | 4.15 | 8.68 |
| CAR Test | SOP | 68.40 | 68.07 | 0.33 | 62.75 |
| | I(P) | 66.24 | 64.09 | 2.15 | 64.02 |
| | SOP+I(P) | 68.83 | 68.40 | 0.43 | 63.66 |
| | In | 59.94 | 54.78 | 5.16 | 28.87 |
| Average Improvement | | | | 2.96 | |
| Success Rate | | | | **0.9375** | |

## G. Additional Notations

The operator $\mathrm{diag}(\cdot)$ creates a matrix with the elements of the input vector placed along the diagonal. Let $\mathbf{1}_{\mathrm{condition}}$ be 1 if the condition is true and 0 otherwise. Let $m!$ be factorials of $m$. Let $n!!$ be double factorial. We define $(-1)!! = 0!! = 1$. For $o_{\mathbb{P}}, O_{\mathbb{P}}, \Theta_{\mathbb{P}}$ notations we follow Moniri et al. (2024) $\|\cdot\|_F$ is the Frobenius norm. $\|\cdot\|_\infty$ is the infinity norm. $\|\cdot\|_{\psi_2}$ is orlicz-2 norm $e^{(i)}$ Standard basis vector with 1 at position $i$. $\lfloor n/2 \rfloor$ denotes the floor of $n/2$. $\Gamma(z)$ is the Gamma function.

**Additional information of Hermite Polynomials** We employ the probabilist's Hermite polynomials (Szegő, 1975; Bienstman, 2023; Moniri et al., 2024). We denote $H_k(x)$ as $k$-th Hermite polynomial.

The $n$-th Hermite polynomials, $H_n(\cdot)$, are defined by the recurrence relation: $H_{n+1}(x) = xH_n(x) - nH_{n-1}(x)$, for $n \geq 1$, with the initial conditions $H_0(x) = 1, H_1(x) = x$. Using this recurrence, we have $H_2(x) = x^2 - 1, H_3(x) = x^3 - 3x, \cdots$.

Hermite polynomials can be represented as the following explicit form:

$$H_n(x) = (-1)^n e^{\frac{x^2}{2}} \frac{d^n}{dx^n} e^{-\frac{x^2}{2}}.$$

for $n \in \mathbb{N}_0$. Lastly, there are another expression:

$$H_n(x) = n! \sum_{m=0}^{\lfloor \frac{n}{2} \rfloor} \frac{(-1)^m}{m!(n-2m)!} \frac{x^{n-2m}}{2^m}$$

The probabilist's Hermite polynomials form an orthogonal set with respect to the standard normal weight function $\phi(x) = \frac{1}{\sqrt{2\pi}} e^{-\frac{x^2}{2}}$ on the interval $(-\infty, \infty)$. Their orthogonality condition is given by:

$$\int_{-\infty}^{\infty} H_m(x) H_n(x) \frac{1}{\sqrt{2\pi}} e^{-\frac{x^2}{2}} dx = n! \mathbf{1}_{\mathrm{m=n}}.$$

## H. hermite coef of shifted ReLU

One of the activation function that satisfy our condition 2.1 is shifted ReLU,

$$\sigma(x) = \max(0, x) - \frac{1}{\sqrt{2\pi}}.$$

This allow hermite decomposition with coefficient is calculated as

$$c_n = \frac{1}{n!} \mathbb{E}_z[\sigma(z) H_n(z)].$$

Then for the zero-th coefficient is calculated as

$$c_0 = \mathbb{E}_z[\sigma(z) \times 1] = \mathbb{E}_z[\max(0, x)] - \frac{1}{\sqrt{2\pi}}$$

$$= \int_0^\infty x\phi(x)dx - \frac{1}{\sqrt{2\pi}} = 0 \tag{10}$$

By the way, if $n \neq 0$, $\mathbb{E}[\frac{1}{\sqrt{2\pi}} \times H_n] = \frac{1}{\sqrt{2\pi}} \mathbb{E}[1 \times H_n] = \frac{1}{\sqrt{2\pi}} \mathbb{E}[H_0 \times H_n] = 0$ by orthogonality. Thus, shift is only effects on $n = 0$.

The coefficient $c_n$ of the expansion of Shifted-ReLU is defined as:

$$c_n = \begin{cases} 0, & \text{if } n = 0, \\ \sum_{m=0}^{\lfloor n/2 \rfloor} \frac{(-1)^m \cdot 2^{\frac{n-2m}{2}-m} \cdot \Gamma\left(\frac{n-2m+2}{2}\right)}{m! \cdot (n-2m)! \cdot \sqrt{2\pi}}, & \text{otherwise.} \end{cases} \tag{11}$$

We directly calculated equation 11 and obtained the following result in Figure 38.

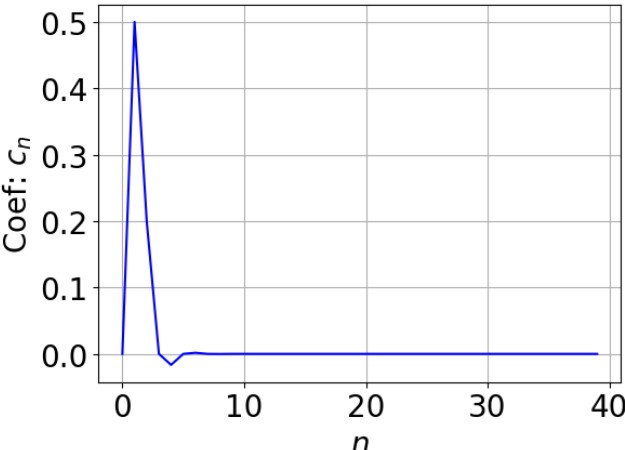

Figure 38: Hermite Coefficient of Shifted ReLU

# I. Proof of Theorem 3.1

In this section, we follow the proof structure of Ba et al. (2022) to decompose gradient in our classification learning setting. Unlike their assumption of centered Gaussian training data, we consider non-centered Sub-Gaussian data distributions. In this process, we apply a novel approach involving the concentration of the operator norm on a random matrix. Also, since our framework is not in a teacher-student setting, we use class labels instead of a teacher function.

We will omit the subscript $ij$ since it does not cause any confusion untill equation 35. The following statements hold for $\forall ij$. For the aforementinoed $\mathbb{A}$, $\mathbb{B}$, and $\mathbb{C}$, we obtain bounds for each operator norm as follows

**Lemma I.1.**

$$\mathbb{P}\left(\|\mathbb{A}\| \le C(\frac{1}{\sqrt{\mathbf{N}}} - C\frac{\sqrt{\mathbf{d}}}{\sqrt{\mathbf{nN}}})\right) \le 2\left(e^{-c\mathbf{N}} + e^{-c\mathbf{n}}\right)$$

$$\mathbb{P}\left(\|\mathbb{B}\| \ge \frac{C}{\mathbf{n}\sqrt{\mathbf{Nd}}}(\sqrt{\mathbf{n}} + \sqrt{\mathbf{d}})(\sqrt{\mathbf{n}} + \sqrt{\mathbf{N}})\log \mathbf{N}\right) \le C\left(e^{-c\mathbf{N}} + e^{-c\mathbf{d}} + \mathbf{N}e^{-c\log^2 \mathbf{n}} + e^{-(\sqrt{\mathbf{n}}+\sqrt{\mathbf{d}})^2}\right) \quad (12)$$

$$\mathbb{P}\left(\|\mathbb{C}\| \ge \frac{C}{\sqrt{\mathbf{nN}}}(2\sqrt{\mathbf{d}} + \sqrt{\mathbf{n}})\log \mathbf{n}\log \mathbf{N}\right) \le 2\left(\mathbf{n}e^{-c\mathbf{d}} + \mathbf{n}e^{-c\log^2 \mathbf{n}} + \mathbf{N}e^{-c\log^2 \mathbf{n}}\right).$$

*Proof of Lemma I.1 ($\mathbb{A}$).* We obtain

$$\mathbb{A} = \frac{c_1}{\mathbf{n}\sqrt{\mathbf{N}}}X^\top y a^\top. \quad (13)$$

Then, we can find an explicit notation of the norm as

$$\|\mathbb{A}\| = \frac{c_1}{\mathbf{n}\sqrt{\mathbf{N}}}\|X^\top y a^\top\| = \frac{c_1}{\mathbf{n}\sqrt{\mathbf{N}}}\|X^\top y\|_2\|a\|_2 = \frac{c_1}{\mathbf{n}\sqrt{\mathbf{N}}}\left(y^\top XX^\top y\right)^{1/2}\|a\|_2 \quad (14)$$

$\|a\|_2$ **study** By definition, $a \sim \mathbf{N}(0, \frac{1}{\mathbf{N}})$, so $\sqrt{\mathbf{N}}\alpha[i]$ is a sub-Gaussian. Use Thm 3.3.1 in Vershynin (2018),

$$\mathbb{P}\left(\left|\|\sqrt{\mathbf{N}}\alpha\| - \sqrt{\mathbf{N}}\right| \ge t\right) \le 2e^{-ct^2} \quad \text{let } t = \sqrt{\mathbf{N}}$$

$$\mathbb{P}(\|\alpha\|_2 \le 1) \le 2e^{-c\mathbf{N}} \quad (15)$$

$\left(y^\top XX^\top y\right)^{1/2}$ **study** Note that the $U, V$ matrices resulting from the SVD belong to the $O$-group, so there is no length transformation.

$$y^\top XX^\top y = \|X^\top y\|_2^2 = \|U\Sigma V^\top y\|_2^2 = \|\Sigma V^\top y_1\|$$

$$= \sum_i \sigma_i^2|V^\top y_i|^2 \ge \sigma_{\min}^2 \sum_i |V^\top y_i|^2 = \sigma_{\min}^2\|y\|_2^2 = \mathbf{n}\sigma_{\min}^2 \quad (16)$$

We get $\left(y^\top XX^\top y\right)^{1/2} \geq \sqrt{\mathbf{n}}\sigma_{\min}$. $\sigma_{\min}$ is singular value of $X$ which is a anistropic sub-Gaussian matrix. With the result of Remark 1.2 in Liaw et al. (2016),

$$\mathbb{P}\sigma_{\min} \leq (\sqrt{\mathbf{n}} - c\sqrt{\mathbf{d}})) \leq e^{-\mathbf{n}}. \tag{17}$$

Therefore, $\mathbb{P}(\|\mathbb{A}\| \leq C(\frac{1}{\sqrt{\mathbf{N}}} - C\frac{\sqrt{d}}{\sqrt{\mathbf{nN}}})) \leq 2(e^{-c\mathbf{N}} + e^{-c\mathbf{n}}).$ $\qquad\square$

*Fact* I.2 (from Ba et al. (2022)). For $m \in \mathbb{R}^m, n \in \mathbb{R}^n, M \in \mathbb{R}^{m \times n}$,

$$mn^\top \odot M = \mathrm{diag}(m)M\mathrm{diag}(n)$$
$$\|mn^\top \odot M\| \leq \|\mathrm{diag}(m)\|\,\|M\|\,\|\mathrm{diag}(n)\| = \|m\|_\infty\|M\|\|n\|_\infty \tag{18}$$

**Lemma I.3.** *For Sub-Gaussian R.V. $a$,*

$$\mathbb{P}(\|a\|_\infty \leq t/\sqrt{\mathbf{N}}) \geq 1 - 2\mathbf{N}e^{-ct^2}$$

*Proof.* We use the Hoeffding inequality such that

$$\mathbb{P}(\|a\|_\infty \geq \frac{t}{\sqrt{\mathbf{N}}}) = \mathbb{P}\left(\max_i |a_i| \geq \frac{t}{\sqrt{\mathbf{N}}}\right) \leq \mathbb{P}\left(\bigcup_i\{|a_i| \geq \frac{t}{\sqrt{\mathbf{N}}}\}\right) \leq \sum_i \mathbb{P}\left(|a_i| \geq \frac{t}{\sqrt{\mathbf{N}}}\right)$$
$$\overset{\text{i.i.d.}}{=} \mathbf{N}\mathbb{P}\left(|a_i| \geq \frac{t}{\sqrt{\mathbf{N}}}\right) = \mathbb{P}(|\sqrt{\mathbf{N}}a_i| \geq t) \leq 2\mathbf{N}\exp(-ct^2) \tag{19}$$

$\qquad\square$

*Fact* I.4. Let a sub-Gaussian random variable $v$ s.t. $\|v\|_{\psi_2} \leq k$, and bounded function $\sigma$, then $\sigma(v)$ is Sub-Gaussian, i.e. $\|\sigma(v)\|_{\psi_2} \leq \|\lambda\|_{\psi_2} < \infty$.

*Proof of Lemma I.1 ($\mathbb{B}$).*

$$\mathbb{B} = \frac{1}{\mathbf{n}\sqrt{\mathbf{N}}}X^\top ya^\top \odot \sigma'_\perp(XW_0) \tag{20}$$

$$\|\mathbb{B}\| \leq \frac{1}{\mathbf{n}\sqrt{\mathbf{N}}}\|X\|\,\|ya^\top \odot \sigma'_\perp(XW_0)\|$$
$$\leq \frac{1}{\mathbf{n}\sqrt{\mathbf{N}}}\|X\|\|ya^\top \odot \sigma'_\perp(XW_0)\|$$
$$\leq \frac{1}{\mathbf{n}\sqrt{\mathbf{N}}}\|X\|\|y\|_\infty\,\|\sigma'_\perp(XW_0)\|\,\|a\|_\infty \tag{21}$$
$$= \frac{1}{\mathbf{n}\sqrt{\mathbf{N}}}\|X\|\,\|\sigma'_\perp(XW_0)\|\|a\|_\infty$$

$\|\sigma'_\perp(XW_0)\|$ **study** Use the result of D.4 in Fan & Wang (2020), which is hold for orthogonal columns. $X$ is sampled from continuous support distribution $c_1, c_2$. The first vector is linearly independent with probability 1 due to the continuous support of its distribution. For the second vector, which is drawn independently, the probability that it lies in the span of the first vector is 0, as it also has a continuous density. This reasoning extends to n vectors, implying that, with high probability, they are orthogonal or nearly orthogonal because no vector falls into the span of the others. Thus, $\forall B > 0$ following is hold.

$$\mathbb{P}(\{\|\sigma'_\perp\| \geq C(\sqrt{\mathbf{n}} + \sqrt{\mathbf{N}})\lambda_\sigma B\}, \mathscr{A}_B) \leq 2e^{-c\mathbf{N}}$$
$$\mathscr{A}_B = \{\{\|W_0\| \leq B\} \cup \{\sum_{i=1}^{\mathbf{N}}(\|W_{0,i}\|^2 - 1)^2 \leq B^2\}\}. \tag{22}$$

Therefore,

$$\mathbb{P}(\|\sigma'_\perp\| \geq C(\sqrt{\mathbf{n}} + \sqrt{\mathbf{N}})\lambda_\sigma B) \leq 2e^{-c\mathbf{N}} + \mathbb{P}(\mathscr{A}_B^c) \tag{23}$$

$\mathbb{P}(\mathscr{A}_B)$ **study** We choose $t = C\sqrt{\frac{\mathbf{d}}{\mathbf{N}}}, B = C\sqrt{\frac{\mathbf{d}}{\mathbf{N}}}$.

**case of** $\|W_{0,i}\| \le B$   By Lemma L.3,

$$\mathbb{P}(\|\sqrt{\mathbf{N}}W_0\| \ge 2\sqrt{\mathbf{N}} + \sqrt{\mathbf{d}}) \le 2e^{-c\mathbf{N}} \Rightarrow \quad \mathbb{P}(\|W_0\| \ge C\sqrt{\frac{\mathbf{d}}{\mathbf{N}}}) \le 2e^{-c\mathbf{N}} \tag{24}$$

Therefore, $\|W_0\| \le B$ at least w.p. $1 - 2e^{-c\mathbf{N}}$

**case of** $\sum_{i=1}^{N}(\|W_{0,i}\|^2 - 1)^2 \le B^2$   By definition, $\|W_{0,i}\|^2 = 1$, so $0 \le B^2$, trivialy.

We know $\mathbb{P}(\mathscr{A}_B^c) \le 2e^{-cN}$.

$$\mathbb{P}(\|\sigma'_\perp\| \ge C(\sqrt{\mathbf{n}} + \sqrt{\mathbf{N}})\sqrt{\frac{\mathbf{d}}{\mathbf{N}}}) \le 2e^{-c\mathbf{N}} \tag{25}$$

Use Lemma I.3, and L.3,

$$\|\sigma'_\perp\| \le C\left(\sqrt{\frac{\mathbf{nN}}{\mathbf{d}}} + \sqrt{\frac{\mathbf{N}^2}{\mathbf{d}}}\right) \qquad\qquad \text{w.p. } 1 - C(e^{-c\mathbf{N}} + e^{-c\mathbf{d}}) \tag{26}$$

$$\|a\|_\infty \le \frac{t}{\sqrt{\mathbf{N}}} \qquad\qquad \text{w.p. } 1 - 2\mathbf{N}e^{-ct^2} \tag{27}$$

$$\|X\| \le \sqrt{\mathbf{n}} + \sqrt{\mathbf{d}} + t' \qquad\qquad \text{w.p. } 1 - 2e^{-ct'^2}. \tag{28}$$

In summary, we get

$$\|\mathbb{B}\| \le \frac{C}{\mathbf{n}\sqrt{\mathbf{N}}}(\sqrt{\mathbf{n}} + \sqrt{\mathbf{d}} + t')\left(\sqrt{\frac{\mathbf{nN}}{\mathbf{d}}} + \sqrt{\frac{\mathbf{N}^2}{\mathbf{d}}}\right)\frac{t}{\sqrt{\mathbf{N}}}$$

$$\text{let } t = \log\mathbf{n}, t' = \sqrt{\mathbf{n}} + \sqrt{\mathbf{d}} \tag{29}$$

$$\mathbb{P}(\|\mathbb{B}\| \ge \frac{C}{\mathbf{n}\sqrt{\mathbf{Nd}}}(\sqrt{\mathbf{n}} + \sqrt{\mathbf{d}})(\sqrt{\mathbf{n}} + \sqrt{\mathbf{N}})\log N) \le C\left(e^{-c\mathbf{N}} + e^{-c\mathbf{d}} + \mathbf{N}e^{-c\log^2\mathbf{n}} + e^{-(\sqrt{\mathbf{n}}+\sqrt{\mathbf{d}})^2}\right).$$

This compelete the proof. $\qquad\qquad\qquad\qquad\qquad\qquad\qquad\qquad\qquad\qquad\qquad\qquad\qquad\qquad\qquad\square$

*Proof of Lemma I.1 ($\mathbb{C}$).* We know that $\sigma'$ is bounded, so $\|\sigma'\|_F \le \lambda_\sigma\sqrt{\mathbf{nN}}$

$$\mathbb{C} = -\frac{1}{\mathbf{nN}}X^\top\sigma(XW_0)(aa^\top) \odot \sigma'(XW_0), \tag{30}$$

ans we can bound the norm as follows

$$\|\mathbb{C}\| \le \frac{1}{\mathbf{nN}}\|X\|\|\sigma aa^\top \odot \sigma'\|$$

$$\le \frac{1}{\mathbf{nN}}\|X\|\|\sigma a\|_\infty\|a\|_\infty\|\sigma'\|_F \tag{31}$$

$$\le \frac{\lambda_\sigma}{\sqrt{\mathbf{nN}}}\|X\|\|\sigma a\|_\infty\|a\|_\infty$$

**Control of** $\|\sigma a\|_\infty$   Let $t = \sqrt{\mathbf{d}}$. Given $X$ s.t. $\mathbb{P}(|X_i - \sqrt{\mathbf{d}}| \ge \sqrt{\mathbf{d}}) \le 2e^{-ct^2}$, consider one element $\sigma(X_j^\top W_0)a = \sum_i^{\mathbf{N}} a_i\sigma(X_j^\top W_0[i])$.

We know $a_i, \sqrt{\mathbf{n}}W_{0,i}$ is an independent centered sub-Gaussian, and use Fact I.4, then $\sigma\left(\frac{X_j^\top}{\sqrt{\mathbf{N}}}\sqrt{\mathbf{N}}W_0\right)a$ is sub-exponential and mean is zero, since $\|a\sigma(x_j^\top W_{0,i})\|_{\psi_1} \le \|a\|_{\psi_2}\|\sigma(x_j^\top W_{0,i})\|_{\psi_2} < \infty$. Apply the Bernstein inequality for the sub-exponential,

$$\mathbb{P}(|\sigma(X_j^\top a)| \ge \log\mathbf{n} \text{ given } \{|X_j - \sqrt{\mathbf{d}}| \ge \sqrt{\mathbf{d}}\}) \le 2e^{-c\log^2\mathbf{n}}. \tag{32}$$

For every element $\|\sigma(XW_0)a\|_\infty \le \log\mathbf{n}$ w.p. $1 - [2\mathbf{n}e^{-c\log^2\mathbf{n}+2\mathbf{n}e^{-c\mathbf{d}}}]$

By Lemma I.3 $\mathbb{P}(\|a\|_\infty \leq t/\sqrt{\mathbf{N}}) \geq 1 - 2\mathbf{N}e^{-ct^2}$, and Lemma L.3 with $t = \sqrt{\mathbf{d}}$

$$\mathbb{P}\left(\|\mathbb{C}\| \geq \frac{C}{\sqrt{\mathbf{nN}}}(2\sqrt{\mathbf{d}} + \sqrt{\mathbf{n}})\log\mathbf{n}\log\mathbf{N}\right) \leq 2\left(\mathbf{n}e^{-c\mathbf{d}} + ne^{-c\log^2\mathbf{n}} + \mathbf{N}e^{-c\log^2\mathbf{n}}\right). \tag{33}$$

$\square$

*Remark* I.5. In the proportional regime, as $\mathbf{n}, \mathbf{d}, \mathbf{N} \to \infty$, these quantities can be interchanged to a constant. Thus, Lemma I.1 is reformulated as follows

$$\mathbb{P}(\|\mathbb{A}\| \leq \kappa/\sqrt{\mathbf{n}}) \leq Ce^{-c\mathbf{n}})$$

$$\mathbb{P}\left(\|\mathbb{B}\| \geq \frac{C\log\mathbf{N}}{\mathbf{n}}\right) \leq C(e^{-c\mathbf{n}} + \mathbf{n}e^{-c\log^2\mathbf{n}}) \tag{34}$$

$$\mathbb{P}\left(\|\mathbb{C}\| \geq \frac{C\log^2\mathbf{N}}{\mathbf{n}}\right) \leq C(\mathbf{n}e^{-c\mathbf{n}} + \mathbf{n}e^{-c\log^2\mathbf{n}})$$

Also, for gradient, we have

$$\|G\| = \|\mathbb{A} + \mathbb{B} + \mathbb{C}\| \leq \|\mathbb{A}\| + \|\mathbb{B}\| + \|\mathbb{C}\| = O_{\mathbb{P}}(\frac{1}{\sqrt{\mathbf{n}}} + \frac{\log\mathbf{n}}{\mathbf{n}} + \frac{\log^2\mathbf{n}}{\mathbf{n}}) = O_{\mathbb{P}}(\frac{1}{\sqrt{\mathbf{n}}}) \tag{35}$$

Now we denote subscript $ij$ for summary.

*Proof of Theorem 3.1.* Using $\|G_{ij} - \mathbb{A}_{ij}\| = \|\mathbb{B}_{ij} + \mathbb{C}_{ij}\| \leq \|\mathbb{B}_{ij}\| + \|\mathbb{C}_{ij}\|$ and Lemma I.5

$$\mathbb{P}\left(\|G_{ij} - \mathbb{A}_{ij}\| \geq C\frac{\log^2\mathbf{n}}{\mathbf{n}}\right) \leq \mathbb{P}\left(\|G_{ij} - \mathbb{A}_{ij}\| \geq C(\frac{\log n}{n} + \frac{\log^2\mathbf{n}}{\mathbf{n}})\right) \leq Cne^{-c\log^2\mathbf{n}}. \tag{36}$$

Therefore, almost surely, in the proportional limit,

$$\|G_{ij} - \mathbb{A}_{ij}\| \leq C\frac{\log^2\mathbf{n}}{\mathbf{n}} = \frac{\kappa}{\sqrt{\mathbf{n}}}\frac{C}{\kappa}\frac{\log^2\mathbf{n}}{\sqrt{\mathbf{n}}} \leq \|\mathbb{A}_{ij}\|\frac{C}{\kappa}\frac{\log^2\mathbf{n}}{\sqrt{\mathbf{n}}} \leq \kappa'\frac{\log^2\mathbf{n}}{\sqrt{\mathbf{n}}}\left(\|G_{ij}\| + \|G_{ij} - \mathbb{A}_{ij}\|\right). \tag{37}$$

We get $(1 - \kappa'\frac{log^2\mathbf{n}}{\sqrt{\mathbf{n}}})\|G_{ij} - \mathbb{A}\| \leq \kappa'\frac{log^2\mathbf{n}}{\sqrt{\mathbf{n}}}\|G_{ij}\|$. For large enough $\mathbf{n}$ for $1 - \kappa'\frac{log^2\mathbf{n}}{\sqrt{\mathbf{n}}} \geq \frac{1}{2}$,

$$\|G_{ij} - \mathbb{A}_{ij}\| \leq \kappa'\frac{\log^2\mathbf{n}}{\sqrt{\mathbf{n}}}\|G_{ij}\| \leq C\frac{\log^2\mathbf{n}}{\mathbf{n}}$$

Sum up for $\forall ij$,

$$\|G - \sum_{i<j}\mathbb{A}_{ij}\| = \|\sum_{i<j}G_{ij} - \mathbb{A}_{ij}\| \leq \sum_{i<j}\|G_{ij} - \mathbb{A}_{ij}\| \leq C\frac{\log^2\mathbf{n}}{\mathbf{n}}$$

$\square$

## J. Proof of Theorem 3.3

**Lemma J.1.** *The following facts will be used in subsequent proofs. Remark* $\beta_{ij} \triangleq \frac{1}{n}X_{ij}^T y$ *in Theorem 3.2.*

A. $\|X_{ij}\| = O_{\mathbb{P}}(\sqrt{\mathbf{n}})$, $\|y\| = O_{\mathbb{P}}(\sqrt{\mathbf{n}})$, $\|\beta_{ij}\| = O_{\mathbb{P}}(1)$

B. $\|X_{ij}\beta_{ij}a_{ij}\| = \|X\beta_{ij}\|_2\|a_{ij}\|_2 = O_{\mathbb{P}}(\sqrt{n})$

C. $\|W_0\| = O_{\mathbb{P}}(1)$, $\|W\| = \|W_0 + G\| \leq \|W_0\| + \|G\| = O_{\mathbb{P}}(1)$

D. $\|X_{ij}G\| = O_{\mathbb{P}}(\sqrt{n})$

E. $M_a \triangleq \|a\|_\infty = \max_{1 \leq i \leq \mathbf{N}}|a_i| \leq \frac{C\log^{1/2}\mathbf{n}}{\sqrt{\mathbf{n}}}$ *w.p* $1 - 2ne^{-c\log\mathbf{n}}$

*F.* $M_b \triangleq \|X\beta\|_\infty = \max_{1 \leq i \leq \mathbf{n}} |<\tilde{X}[i], \beta>| \leq C \log^{1/2} \mathbf{n}$, *w.p.* $1 - 2\mathbf{n}e^{-c \log \mathbf{n}}$

*G.* $M_{W_0} \triangleq \sup_{k \geq 1} \|(W_0 W_0^\top)^{\circ k}\| \leq C$ *w.p.* $1 - o(1)$

*H.* $\|A^{\circ k}\| \leq \|A\|^k$

*Proof.* It is evident from Lemma L.3, equation 15 in the proportional regime, that A, B, C, and D hold. To proof E, F, and G, we employ proof techniques adapted from Moniri et al. (2024). For E, by Lemma I.3, with $t = \log^{\frac{1}{2}} \mathbf{n}$, $M_a \leq \frac{C \log^{\frac{1}{2}} n}{\sqrt{n}}$, w.p. $1 - o(1)$.

For F,

$$\begin{aligned}
\mathbb{P}(C|x^T\beta| \geq t) &= \mathbb{P}(C|x^T\beta - Ex^T\beta + Ex^T\beta| \geq t) \\
&\leq \mathbb{P}(C|x^T\beta - Ex^T\beta| \geq t - C|Ex^T\beta|) \leq 2\exp(-ct^2).
\end{aligned} \tag{38}$$

Then, $\mathbb{P}(|x^T\beta| \geq t) \leq 2\exp(-c(t - Ex^T\beta)^2) \leq 2\exp(-ct^2)$.

Therefore, $M_b \leq C \log^{\frac{1}{2}} n$, w.p. $1 - o(1)$ with $t = \log^{\frac{1}{2}} \mathbf{n}$.

For G, refer Moniri et al. (2024). For H, refer Bai & Silverstein (2010) Corollary A.21. $\qquad\square$

**Corollary J.2** (Corollary of Theorem 3.1). *By Lemma J.1, we have w.p.* $1 - o(1)$,

$$\|\tilde{X}G - c_1\tilde{X}\sum_{i<j}\beta_{ij}a_{ij}^T\| = O(\frac{\log^2 \mathbf{n}}{\mathbf{n}} \cdot \sqrt{\mathbf{n}}) = O(\frac{\log^2 \mathbf{n}}{\sqrt{\mathbf{n}}}) \tag{39}$$

*Remark* J.3. $W_1 = W_0 + G$, so $\tilde{X}W_1 = \tilde{X}W_0 + \tilde{X}G$. $\tilde{X}$ is i.i.d. copy of training data $X$

We generalize Corollary J.2 i.e. monomial approximation of data-gradient product in polynomial form as Lemma J.4 .

**Lemma J.4** (Polynomial Approximation of Data-Gradient Product). *For any* $k \in \mathbb{N}$, *sufficiently large* $\mathbf{n}$, *and w.p. 1 - o(1),*

$$\|(\tilde{X}G)^{\circ k} - c_1^k(\tilde{X}\sum_{i<j}\beta_{ij}a_{ij}^T)^{\circ k}\| = O(\mathbf{n}^{-\frac{k}{2}}\log^{2k}\mathbf{n}) \tag{40}$$

*Proof of Lemma J.4.* $k = 1$ is trivial Corollary J.2. We follow Moniri et al. (2024) for $k \geq 2$. We need to show $\exists C > 0$, w.p. 1-o(1)

$$\|(\tilde{X}G)^{\circ k} - c_1^k(\tilde{X}\sum_{i<j}\beta_{ij}a_{ij}^T)^{\circ k}\| \leq C\mathbf{n}^{-\frac{k}{2}}\log^{2k}\mathbf{n} \tag{41}$$

$$\begin{aligned}
(\tilde{X}G)^{\circ k} &= (\tilde{X}G - c_1\tilde{X}\sum_{i<j}\beta_{ij}a_{ij}^T + c_1\tilde{X}\sum_{i<j}\beta_{ij}a_{ij}^T)^{\circ k} \\
&= \sum_{j=1}^k (k_j)(\tilde{X}G - c_1\tilde{X}\sum_{i<j}\beta_{ij}a_{ij}^T)^{\circ j} \odot (c_1\tilde{X}\sum_{i<j}\beta_{ij}a_{ij}^T)^{\circ(k-j)} + c_1^k(\tilde{X}\sum_{i<j}\beta_{ij}a_{ij}^T)^{\circ k}
\end{aligned} \tag{42}$$

Thus,

$$\begin{aligned}
&(\tilde{X}G)^{\circ k} - c_1^k(\tilde{X}\sum_{i<j}\beta_{ij}a_{ij}^T)^{\circ k} \\
&= \sum_{j=1}^k \binom{k}{j}(\tilde{X}G - c_1\tilde{X}\sum_{i<j}\beta_{ij}a_{ij}^T)^{\circ j} \odot c_1^{k-j}(\sum_{i<j}(\tilde{X}\beta_{ij}a_{ij}^T))^{\circ(k-j)}
\end{aligned} \tag{43}$$

Now we will show

$$\|(\tilde{X}G - c_1\tilde{X}\sum_{i<j}\beta_{ij}a_{ij}^T)^{\circ j} \odot c_1^{k-j}(\sum_{i<j}(\tilde{X}\beta_{ij}a_{ij}^T))^{\circ(k-j)}\| = O_\mathbb{P}(\log^{k+j}\mathbf{n} \cdot \mathbf{n}^{-\frac{1}{2}k}).$$

$$\|(\tilde{X}G - c_1\tilde{X}\sum_{i<j}\beta_{ij}a_{ij}^T)^{oj} \odot c_1^{k-j}(\sum_{i<j}(\tilde{X}\beta_{ij}a_{ij}^T))^{o(k-j)}\|$$

$$\leq C\|(\tilde{X}G - c_1\tilde{X}\sum_{i<j}\beta_{ij}a^T)^{oj} \odot (\tilde{X}\beta a^T)^{o(k-j)}\|$$

$$\leq C\|\text{diag}(\tilde{X}\beta)^{ok-j}\|_{op}\|(\tilde{X}G^T - c_1\tilde{X}\sum_{i<j}\beta_{ij}a^T)^{oj}\|_{op}\|\text{diag}(a)^{ok-j}\| \tag{44}$$

$$\leq C(M_aM_b)^{k-j}\|(\tilde{X}G - c_1\tilde{X}\sum_{i<j}\beta_{ij}a^T)^{oj}\|^j$$

$$\leq C(n^{-\frac{1}{2}(k-j)}\log^{k-j}\mathbf{n})\log^{2j}\mathbf{n}\cdot\mathbf{n}^{-\frac{1}{2}j}$$

$$= O_{\mathbb{P}}(\mathbf{n}^{-\frac{1}{2}k}\log^{k+j}\mathbf{n})$$

Therefore,

$$\|(\tilde{X}G)^{ok} - c_1^k(\tilde{X}\sum_{i<j}\beta_{ij}a_{ij}^T)^{ok}\| = O_{\mathbb{P}}(\mathbf{n}^{-\frac{k}{2}}\log^{2k}\mathbf{n}) \tag{45}$$

$\square$

**Lemma J.5.** *Following condition in section 2, Assume event* $\Omega = \sup_{k\geq 1}\|(W_0W_0^T)^{ok}\|_{op} \leq C$ *occur, following statement holds.*

$$\|H_j(\tilde{X}W_0)\|_{op} = O_{\mathbb{P}}(\sqrt{n}\log^{\frac{3}{2}}n\sqrt{j!})$$

**Lemma J.6.** *Given random matrix $A$, Following statement holds,*

$$\mathbb{P}(\|A\|_{op} \geq t) \leq \mathbb{P}(\|\frac{1}{n}AA^T - EAA^T\|_{op} \geq \frac{t^2}{n} - \|EAA^T\|_{op})$$

*Proof of Lemma J.6.*

$$\mathbb{P}(\|A\|_{op} \geq t) = \mathbb{P}(\|A\|_{op}^2 \geq t^2) = \mathbb{P}(\|\frac{1}{n}AA^T\|_{op} \geq \frac{t^2}{n})$$

$$= \mathbb{P}(\|\frac{1}{n}AA^T - EAA^T + EAA^T\|_{op} \geq \frac{t^2}{n})$$

$$\leq \mathbb{P}(\|\frac{1}{n}AA^T - EAA^T\|_{op} + \|EAA^T\|_{op} \geq \frac{t^2}{n}) \tag{46}$$

$$= \mathbb{P}(\|\frac{1}{n}AA^T - E(AA^T)\|_{op} \geq \frac{t^2}{n} - E\|AA^T\|_{op})$$

$\square$

**Lemma J.7.** *Following condition of Lemma J.5,*

$$E\|H_j(\tilde{X}W_0)H_j(\tilde{X}W_0)^\top\|_{op} \leq Cj!$$

*Proof of Lemma J.7.* For non-centered Sub Gaussian random variable $X$ with mean $\mu$,

$$E(e^{(X-\mu)t}) \leq e^{\frac{k^2}{2}t^2}$$

$$Ee^{Xt} \leq e^{\frac{k^2}{2}t^2+\mu t} \tag{47}$$

Firstly, we proof $\mu = 0$ case. For centered Sub Gaussian vector $g$, let $z = g^\top u, z' = g^\top v, \rho$-correlated. s.t. $\|u\|^2 = \|v\|^2 = 1, u^Tv = \rho$, then by equation 47

$$\mathbb{E}\exp(sz + tz') \leq \exp(\frac{k^2}{2}\|u\|^2s^2 + k^2 <\vec{u},\vec{v}> st + \frac{k^2}{2}\|v\|^2t^2)$$

$$\leq \exp(\frac{k^2}{2}(s^2 + 2\rho st + t^2))$$

Dividing by $\exp(\frac{k^2}{2}(s^2 + t^2))$, then

$$E[\exp(sz - \frac{k^2}{2}s^2)\exp(tz' - \frac{k^2}{2}t^2)] \leq \exp(\rho st) = \sum_{j=0}^{\infty} \frac{\rho^j}{j!}s^j t^j$$

Using proof techniques similar to those in Lemma M.1, one can acquire

$$EH_j(u^T g)H_k(v^T g) \leq j!\rho^j \mathbf{1}_{j=k} \tag{48}$$

For $\mu \neq 0$ case, considering non-centered Sub Gaussian Random vector $g$ with mean $\mu$ and centered Sub Gaussian Random vector $\xi$ s.t. $g = \xi + \mu$. We use proof techniques similar to those in Theorem M.11.

Denote $\nu = \min(j, k)$. Considering $u^\top g, v^\top g$,

$$\mathbb{E}[H_j(u^T \mu + u^T \xi)H_k(v^T \mu + v^T \xi)]$$

$$= \mathbb{E}[\{\sum_{i=0}^{j}\binom{j}{i}(u^T\mu)^i H_{j-i}(u^T\xi)\} \cdot \{\sum_{h=0}^{k}\binom{k}{h}(v^T\mu)^h H_{k-h}(v^T\xi)\}]$$

$$= \mathbb{E}[\sum_{q=0}^{\nu}\binom{\nu}{q}^2 (u^T\mu)^{j-q}(v^T\mu)^{k-q}H_q(u^T\xi)H_q(v^T\xi)] \quad \text{by equation 48} \tag{49}$$

$$\leq \sum_{q=0}^{\nu}\binom{\nu}{q}^2 (u^T\mu)^{j-q}(v^T\mu)^{k-q} \cdot \nu!\rho^\nu$$

$$\leq C\min(j, k)!$$

$\square$

*Proof of Lemma J.5.* Let $A = H_j(\tilde{X}W_0)$, then

$$\mathbb{P}(||A||_{op} \geq t) \leq \mathbb{P}\left(||\frac{1}{n}AA^T - EAA^T||_{op} \geq \frac{t^2}{n} - ||EAA^T||_{op}\right) \quad \text{(by Lemma J.6)}$$

$$\leq \frac{1}{\frac{t^2}{n} - ||EAA^T||_{op}}E\left[||\frac{1}{n}AA^T - EAA^T||_{op}\right] \quad \text{(by Markov's inequality)}$$

$$\leq \left[\frac{t^2}{n} - E\left[||AA^T||_{op}\right]\right]^{-1}\delta\max\left(\sqrt{||EAA^T||_{op}}, \delta\right) \quad \text{(by Theorem 5.48 in Vershynin (2010))}$$

$$\leq \left[\frac{t^2}{n} - E\left[||AA^T||_{op}\right]\right]^{-1}\delta\max\left(\sqrt{E\left[||AA^T||_{op}\right]}, \delta\right) \quad \text{(by Jensen's inequality)}.$$

Let $M = E\max_i ||H_j(W_0\tilde{x}_i)||^2$ and $\delta = C\sqrt{\frac{M\log n}{N}}$. Moreover, we note that $\frac{||\tilde{x}_i||^{2j}}{N}$ is sub-weibull random variable and bound of (Kuchibhotla & Chakrabortty, 2022) proposition A.6 can be applied.

Use property of $\frac{||\tilde{x}_i||^{2j}}{N}$, $W_0$ and hermite polynomials, we have

$$M \leq c_j E\max_i ||(W_0\tilde{x}_i)^{\circ j}||_2^2 \leq c_j E\max_i ||x||^{2j} \leq c_j N(\log n)^{\frac{1}{2}}.$$

Therefore, $\delta \leq C\log n$. Let $t^2 = n \cdot Q_n E||AA^T||_{op}$ s.t. $Q_n$ is positive and increasing. Building on the result derived

above, we can continue expanding the expression as follows:

$$\left[\frac{t^2}{n} - E\left[||AA^T||_{op}\right]\right]^{-1}\delta\max\left(\sqrt{E\left[||AA^T||_{op}\right]},\delta\right)$$

$$\leq [\frac{t^2}{n} - E||AA^T||_{op}]^{-1}C\log n\max(\sqrt{E||AA^T||_{op}},\log n)$$

$$= [E||AA^T||_{op}(Q_n - 1)]^{-1}C\log n\max(\sqrt{E||AA^T||_{op}},\log n) \tag{50}$$

$$\leq C\frac{\log n\max(\sqrt{E||AA^T||_{op}},\log n)}{E||AA^T||_{op}Q_n}$$

Choosing $Q_n = \log^3 n$, and using Lemma J.7, we conclude the proof. $\qquad\square$

*Fact* J.8. For any vector $u, v$ and any matrix $A, B$

A. $||uv^T||_{op} = ||u||_2||v||_2$

B. $||u||_\infty \leq ||u||_2 \leq \sqrt{n}||u||_\infty$

C. $||u^{\circ k}|| \leq ||u||^k$

D. $||u^{\circ k}||_2 \leq \sqrt{n}||u^{\circ k}||_\infty \leq \sqrt{n}\max_i(|u_i^k|) = \sqrt{n}(\max_i|u_i|)^k = \sqrt{n}||u||_\infty^k$

E. Schur product theorem

$$||A \circ B||_{op} = \sup_{||x||=1} tr(A^T\text{diag}(x)B\text{diag}(x)) \leq ||A||_{op} \cdot ||B||_{op}$$

Next, let $L = O(\log n)$.

Denote $\sigma_L(z) = \sum_{k=1}^L c_k H_k(z)$, $F^L = \sigma_L(\tilde{X}W)$ and $F_0^L = \sigma_L(\tilde{X}W_0)$.

Then, $F = F^L + (\sigma - \sigma_L)(\tilde{X}W)$.

Using Lemma J.5, $w$ in assumption 2.1, w.p. $1 - o(1)$

$$||E[(\sigma - \sigma_L)(W_0X)(\sigma - \sigma_L)(W_0X)^T]||$$

$$\leq C\sum_{k=L+1}^\infty k!c_k^2 \leq C\sum_{k=L+1}^\infty k^{-3-w} \leq C\int_L^\infty k^{-\frac{3}{2}-w}dk \leq CL^{-2-w}. \tag{51}$$

Therefore, following same proof technique as Lemma J.5, J.6, J.7,

$$||(\sigma - \sigma_L)(\tilde{X}W_0)||_{op} = o_{\mathbb{P}}(\sqrt{n\log^3 n} \cdot L^{-2-w}) = o_{\mathbb{P}}(\sqrt{n}) \tag{52}$$

Also, because $||W||_{op} = O(1)$,

$$||(\sigma - \sigma_L)(\tilde{X}W)||_{op} = o(\sqrt{n\log^3 n} \cdot L^{-2-w}) = o_{\mathbb{P}}(\sqrt{n}) \tag{53}$$

Finally, we proof Theorem 3.3.

*Proof of Theorem 3.3.* We write $F^L + F_0^L = F^L + F_0^L$, then $F^L = F_0^L + \sum_{k=1}^L c_k(H_k(\tilde{X}W) - H_k(\tilde{X}W_0))$. We have to study $H_k(\tilde{X}W) - H_k(\tilde{X}W_0)$ term.

$$H_k(\tilde{X}W) - H_k(\tilde{X}W_0)$$

$$= H_k(\tilde{X}W_0^T + \tilde{X}G^T) - H_k(\tilde{X}W_0)$$

$$= (\tilde{X}G)^{\circ k} + \sum_{j=1}^{k-1}\binom{k}{j}H_{k-j}(\tilde{X}W_0) \circ (\tilde{X}G)^{\circ j} \tag{54}$$

Thus,

$$F^L = F_0^L + \sum_{k=1}^{L} c_k (\tilde{X}G)^{ok} + \sum_{k=1}^{L}\sum_{j=1}^{k-1} c_k \binom{k}{j} H_{k-j}(XW_0) \circ (\tilde{X}G)^{\circ j}$$

$$= F_0^L + \sum_{k=1}^{L} c_1^k c_k (\tilde{X}\sum_{i<j} \beta_{ij} a_{ij}^T)^{ok}$$

$$\Delta_1 \left[ \begin{array}{l} - \sum_{k=1}^{L} c_1^k c_k (\tilde{X}\sum_{i<j} \beta_{ij} a_{ij}^T)^{ok} \\[2mm] + \sum_{k=1}^{L} c_k (\tilde{X}G)^{ok} \end{array} \right. \tag{55}$$

$$\Delta_2 \left[ \begin{array}{l} + \sum_{k=1}^{L}\sum_{j=1}^{k-1} c_k \binom{k}{j} H_{k-j}(\tilde{X}W_0) \circ (\tilde{X}G)^{\circ j} \\[2mm] - \sum_{k=1}^{L}\sum_{j=1}^{k-1} c_1^j c_k \binom{k}{j} H_{k-j}(\tilde{X}W_0) \circ [\tilde{X}\sum_{i<j} \beta_{ij} a_{ij}^T]^{\circ j} \end{array} \right.$$

$$\Delta_3 \left[ + \sum_{k=1}^{L}\sum_{j=1}^{k-1} c_1^j c_k \binom{k}{j} H_{k-j}(\tilde{X}W_0) \circ [\tilde{X}\sum_{i<j} \beta_{ij} a_{ij}^T]^{\circ j} \right.$$

$\|F_0^L\| = \Theta(\sqrt{n})$ by Moniri et al. (2024).

$\|\sum_{k=1}^{L} c_1^k c_k (\tilde{X}\sum_{i<j} \beta_{ij} a_{ij}^T)^{ok}\|$ is bigger than $\sqrt{n}$.

For $\Delta_1, \Delta_2, \Delta_3$, it is derived as follows

$$\begin{aligned} \|\Delta_1\| &\leq \sum_{k=1}^{L} c_k \|(\tilde{X}G)^{ok} - c_1^k (\tilde{X}\sum_{i<j} \beta_{ij} a_{ij}^T)^{ok}\| \\ &\leq C \sum_{K=1}^{L} \log^{2k} n \cdot n^{-\frac{k}{2}} = O(\frac{\log^2 n}{\sqrt{n}}) = o(1) \end{aligned} \tag{56}$$

$$\begin{aligned} \|\Delta_2\| &\leq \sum_{k=1}^{L}\sum_{j=1}^{k-1} c_k \binom{k}{j} \|H_{k-j}(\tilde{X}W_0^T) \circ [(\tilde{X}G^T)^{\circ j} - c_1^j[\tilde{X}\sum_{i<j} \beta_{ij} a_{ij}^T]^{\circ j}]\| \\ &\leq C \sum_{k=1}^{L}\sum_{j=1}^{k-1} \|H_{k-j}(\tilde{X}W_0^T)\| \|(\tilde{X}G^T)^{\circ j} - c_1^j (\tilde{X}\sum_{i<j} \beta_{ij} a_{ij}^T)^{\circ j}\| \\ &\leq C \sum_{k=1}^{L}\sum_{j=1}^{k-1} \sqrt{n}\log^{\frac{3}{2}} n \sqrt{j!} \cdot n^{-\frac{j}{2}} \log^{2j} n \\ &\leq C \sum_{k=1}^{L}\sum_{j=1}^{k-1} \frac{\sqrt{n}\sqrt{j!}\log^{\frac{3}{2}+2j} n}{\sqrt{n}^j} = O(\log^{\frac{7}{2}} n) \end{aligned} \tag{57}$$

$$\|\Delta_3\| \le C \sum_{k=1}^{L} \sum_{j=1}^{k-1} \|H_{k-j}(\tilde{X}W_0) \circ [\tilde{X}\sum_{i<j}\beta_{ij}a_{ij}^T]^{\circ j}\|$$

$$\le C \sum_{k=1}^{L} \sum_{j=1}^{k-1} \|\mathrm{diag}(\tilde{X}\beta)^{\circ j}\| \|H_{k-j}(\tilde{X}W_0)\| \|\mathrm{diag}(a)^{\circ j}\|$$

$$\le C \sum_{k=1}^{L} \sum_{j=1}^{k-1} (M_a M_b)^j \|H_{k-j}(\tilde{X}W_0)\| \tag{58}$$

$$\le C \sum_{k=1}^{L} \sum_{j=1}^{k-1} n^{-\frac{1}{2}j} \log^j n \sqrt{n} \log^{\frac{3}{2}} = O(\log^{\frac{5}{2}} n)$$

Therefore, we conclude the proof. $\qquad\square$

## K. Proof of Clustering Risk Analysis in two-classes case

**Definition K.1.** Given $N, d$, let

$$
\begin{aligned}
S_{d,k}^{(1)} &= \mathbb{E}_{w \sim Unif(\mathbb{S}^{d-1})}[(w^T e_1)^k] \in \mathbb{R}_+ \\
S_{d,k,k'}^{(2)} &= E_w[(w^T \hat{\mu}_1)^k (w^T \hat{\mu}_2)^{k'}] \\
\rho_{k,k'}^{(1)} &= N S_{d,k+k'}^{(1)} \mathbf{1}_{\text{k+k' is even}} \in \mathbb{R}_+ \\
\rho_{k,k'}^{(2)}(cos(\mu_1,\mu_2)) &= N S_{d,k,k'}^{(2)} \mathbf{1}_{\text{k+k' is even}} \in \mathbb{R}_+ \\
\rho_{k,k',r}^{(3)} &= \frac{c_1^k S_{d,k'}^{(1)}}{N^{\frac{k}{2}-1}} \binom{k}{r}(r-1)!!(k-1)!! \mathbf{1}_{\text{k,k',r is even}} \in \mathbb{R}_+ \\
\rho_{k,k',r,r'}^{(4)} &= \frac{2c_1^{k+k'} S_{d,k}^{(1)}}{N^{\frac{k'}{2}-1}} \binom{k'}{r'}(r'-1)!!(k'-1)!! \mathbf{1}_{\text{k,k',r' is even}} \in \mathbb{R}_+
\end{aligned}
\tag{59}
$$

For $S_{d,k,k'}^{(2)}$, it depends on $\cos(\mu_1,\mu_2)$. As $\cos(\mu_1,\mu_2)$ increases, $S_{d,k,k'}^{(2)}$ grows, while it decreases as $\cos(\mu_1,\mu_2)$ decreases. e.g. when $\mu_1 = \mu_2$, $S_{d,k,k'}^{(2)} = S_{d,k+k'}^{(1)}$, and when $\mu_1 = -\mu_2 = -S_{d,k+k'}^{(1)}$.

**Lemma K.2.** *Let $C_{d,k} \triangleq \mathbb{E}_\omega[(\omega^\top e_1)^k]$ s.t. $\omega \sim Unif(\mathbb{S}^{d-1})$, then*

$$\mathbb{E}_\omega[(\omega^\top \mu)^k] = \|\mu\|^k S_{d,k}^{(1)} \mathbf{1}_{\text{k is even}} \tag{60}$$

*Proof of K.2.* The uniform distribution on the sphere is origin-symmetric. Therefore, when $k$ is odd, Expectation is zero. In the other case, also use isotropic property of uniform sphere,

$$E_\omega[(\omega^\top \mu)^k] = \|\mu\|^k E_\omega[(\omega^\top e_1)^k] = \|\mu\|^k S_{d,k}^{(1)}$$

$\qquad\square$

In the proof below, we utilize the results of Corollary M.12, Corollary M.13, and Lemma K.2.

**Lemma K.3.** *Given vector $a \in \mathbb{R}^N$ $\beta \in \mathbb{R}^d$ and Gaussian Random vector $x \sim \mathcal{N}(\mu, I)$. Let $b = x^\top \beta \sim \mathcal{N}(\mu^\top \beta, \|\beta\|^2)$, then*

$$\mathbb{E}_x(x^\top \beta a^\top)^{\circ k} = \sum_{r=0}^{k} \binom{k}{r}(\mu^\top \beta)^{k-r}\|\beta\|^r (r-1)!! \mathbf{1}_{\text{r is even}} a^{\circ k\top} \tag{61}$$

$$\mathbb{E}_a a^{\circ k} = \frac{(k-1)!! \mathbf{1}_{\text{k is even}}}{N^{\frac{k}{2}}} \mathbb{1} \tag{62}$$

$$\mathbb{E}_a a^{\circ k\top} a^{\circ k'} = \frac{(k+k'-1)!! \mathbf{1}_{\text{k+k' is even}}}{N^{\frac{k+k'}{2}-1}} \tag{63}$$

*Proof.* This follows directly from Corollary M.12. □

*Proof of Proposition 4.5.* Let *cohesion* of initialized feature as

$$coh_0 = \mathbb{E}_{W_0}[\mathbb{E}_{x\sim c_1}F_0^L(x)^T\mathbb{E}_{x'\sim c_1}F_0^L(x')] \tag{64}$$

Let *cohesion* of feature after training as

$$coh_1 = \mathbb{E}_{W_0,a}[\mathbb{E}_{x\sim c_1}F_L(x)^T\mathbb{E}_{x'\sim c_1}F_L(x')] \tag{65}$$

**Calculate** $coh_0$    By Lemma K.2,

$$
\begin{aligned}
coh_0 &= \mathbb{E}_{W_0}[\mathbb{E}_{x\sim c_1}[\sum_{k=1}^{L}c_kH_k(W_0^Tx)]^T\mathbb{E}_{x'\sim c_1}[\sum_{k'=1}^{L}c_{k'}H_{k'}(W_0^Tx)]]\\
&= \sum_{k=1,k'=1}^{L}c_kc_{k'}\mathbb{E}_{W_0}[\sum_{q=1}^{N}(W_0[q]^T\mu_1)^{k+k'}]\\
&= N\sum_{k=1,k'=1}^{L}c_kc_{k'}(\|\mu_1\|^{k+k'}S_{d,k+k'}^{(1)})\mathbf{1}_{(k+k')\text{even}}\\
&= \sum_{k=1,k'=1}^{L}c_kc_{k'}\rho_{k,k'}^{(1)}\|\mu\|^{k+k'}
\end{aligned}
\tag{66}
$$

**Calculate** $coh_1$

$$
\begin{aligned}
coh_1 &= \mathbb{E}_{W_0,a}[\mathbb{E}_{x\sim c_1}[\sum_{k=1}^{L}(c_kH_k(W_0^Tx)+c_kc_1^k(x^T\beta a)^{ok}]^T\mathbb{E}_{x'\sim c_1}[\sum_{k'=1}^{L}(c_{k'}H_{k'}(W_0^Tx)+c_1^k(x^T\beta a)^{ok}]]\\
&= \mathbb{E}_{W_0,a}[\sum_{k,k'=1}^{L}c_kc_{k'}[\mathbb{E}_x H_k(W_0^Tx)^T\mathbb{E}_{x'}H_{k'}(W_0^Tx\prime)\\
&\qquad\qquad + 2\mathbb{E}_x H_k(W_0^Tx)^T\mathbb{E}_{x'}c_1^{k'}(x^T\beta a)^{ok\prime}+c_1^{k+k'}\mathbb{E}_x(x^T\beta a)^{ok^T}\mathbb{E}_{x'}(x\prime^T\beta a)^{ok\prime}]]\\
&= coh_0 + 2\sum_{k,k'=1}^{L}c_kc_{k'}c_1^{k'}\mathbb{E}_{W_0}\mathbb{E}_x H_k(W_0^Tx)^T\mathbb{E}_a\mathbb{E}_{x'}(x'^\top\beta a)^{ok'}\\
&\quad + \sum_{k,k'=1}^{L}c_kc_{k'}c_1^{k+k'}\mathbb{E}_a[\mathbb{E}_x(x^\top\beta a)^{ok^T}\mathbb{E}_{x'}(x'^\top\beta a)^{ok}]\\
&= coh_0 + 2N\sum_{k,k'=1}^{L}c_kc_{k'}c_1^{k'}(\|\mu_1\|^kS_{d,k}^{(1)})(\frac{1}{N^{\frac{k'}{2}}}\sum_{r'=0}^{k'}\binom{k'}{r'}(\mu_1^T\beta)^{k'-r'}\|\beta\|^{r'}(r\prime-1)!!(k\prime-1)!!\mathbf{1}_{k,k',r'\text{is even}}\\
&\quad + \sum_{k,k'=1}^{L}\frac{c_kc_{k'}c_1^{k+k'}}{N^{\frac{k+k'}{2}}-1}\sum_{r=0}^{k}\sum_{r'=0}^{k'}\binom{k}{r}\binom{k'}{r'}(\mu_1^T\beta)^{k+k'-r-r'}\|\beta\|^{r+r'}(r-1)!!(r'-1)!!\mathbf{1}_{k+k',r,r'\text{is even}}
\end{aligned}
$$

□

*Proof of Proposition 4.6.* Let *separability* of initialized feature as

$$sep_0 = -\mathbb{E}_{W_0}[\mathbb{E}_{x\sim c_1}F_0^L(x)^T\mathbb{E}_{x'\sim c_2}F_0^L(x')] \tag{67}$$

Let *separability* of feature after training as

$$sep_1 = -\mathbb{E}_{W_0,a}[\mathbb{E}_{x\sim c_1}F_L(x)^T\mathbb{E}_{x'\sim c_2}F_L(x')] \tag{68}$$

**Calculate** $sep_0$    By Lemma K.2,

$$
sep_0 = - \sum_{k=1,k'=1}^{L} c_k c_{k'} \mathbb{E}_{W_0} \Big[ \sum_{q=1}^{N} (W_0[q]^T \mu_1)^k (W_0[q]^T \mu_2)^{k'} \Big]
$$

$$
= -N \sum_{k=1,k'=1}^{L} c_k c_{k'} \mathbb{E}_{w \sim Unif(\mathbb{S}^{d-1})} [(w^T \mu_1)^k (w^T \mu_2)^{k'}]
$$

$$
= -N \sum_{k=1,k'=1}^{L} c_k c_{k'} \|\mu_1\|^k \|\mu_2\|^{k'} E_w[(w^T \hat{\mu}_1)^k (w^T \hat{\mu}_2)^{k'}] \tag{69}
$$

$$
= -N \sum_{k=1,k'=1}^{L} c_k c_{k'} \|\mu_1\|^k \|\mu_2\|^{k'} S_{d,k,k'}^{(2)} \mathbf{1}_{k+k' \text{ is even}}
$$

$$
= - \sum_{k=1,k'=1}^{L} c_k c_{k'} \|\mu_1\|^k \|\mu_2\|^{k'} \rho_{k,k'}^{(1)}
$$

**Calculate** $sep_1$

$$
sep_1 = - \sum_{k,k'=1}^{L} c_k c_{k'} \mathbb{E}_{W_0,a}
\begin{bmatrix}
\mathbb{E}_{x \sim c_1} H_k(W_0^T x)^T \mathbb{E}_{x' \sim c_2} H_{k'}(W_0^T x') \\
+ \mathbb{E}_{x \sim c_1} H_k(W_0^T x)^T \mathbb{E}_{x' \sim c_2} c_1^{k'}(x'^T \beta a)^{ok'} \\
+ \mathbb{E}_{x \sim c_1} c_1^k (x^T \beta a)^{ok^T} \mathbb{E}_{x' \sim c_2} H_{k'}(W_0^T x) \\
+ c_1^{k+k'} \mathbb{E}_{x \sim c_1} (x^T \beta a)^{ok^T} \mathbb{E}_{x' \sim c_2} (x'^T \beta a)^{ok'}
\end{bmatrix}
$$

$$
= sep_0 - \sum_{k,k'=1}^{L} c_k c_{k'}
\begin{bmatrix}
c_1^{k'} (\|\mu_1\|^k S_{d,k}^{(1)}) \dfrac{1}{N^{\frac{k'}{2}-1}} \sum_{r'=0}^{k'} \binom{k'}{r'} (\mu_2^T \beta)^{k'-r'} \|\beta\|^{r'} (r'-1)!! (k'-1)!! \mathbf{1}_{k,k',r' \text{ is even}} \\[2mm]
+ c_1^k (\|\mu_2\|^{k'} S_{d,k'}^{(1)}) \dfrac{1}{N^{\frac{k}{2}-1}} \sum_{r=0}^{k} \binom{k}{r} (\mu_1^T \beta)^{k-r} \|\beta\|^r (r-1)!! (k-1)!! \mathbf{1}_{k,r,k' \text{ is even}} \\[2mm]
+ c_1^{k+k'} \sum_{r=0}^{k} \sum_{r'=0}^{k} \binom{k}{r} \binom{k'}{r'} (\mu_1^T \beta)^{k-r} (\mu_2^T \beta)^{k'-r'} \|\beta\|^{r+r'} (r-1)!! (r'-1)!! \\[2mm]
\hspace{3cm} \dfrac{1}{N^{\frac{k+k'}{2}-1}} (k+k'-1)!! \mathbf{1}_{k+k',r,r' \text{ is even}}
\end{bmatrix}
$$

$\square$

## L. Additional Lemmas of Sub-Gaussian Distribution

For more detailed explanation and well known results of Sub-Gaussian we used, please refer to Vershynin (2010; 2018). We show below that the truncated Gaussian distribution, utilized in our synthetic data experiments, is a sub-Gaussian distribution.

**Lemma L.1.** *Truncated Gaussian distribution which have support on* $(a, b)$ *s.t.* $a, b \in (-\infty, \infty)$ *is Sub-Gaussian.*

*Proof.* Denote $\mathcal{N}_{(a,b)}(0, \sigma^2)$ is Truncated Gaussian distribution which have support on $(a, b)$ s.t. $a, b \in (-\infty, \infty)$. support $(\mathcal{N}_{(a,b)}(0, \sigma^2)) \subset \mathbb{R}^d$. Therefore, $\mathbb{P}(|X| \geq t)$ s.t. $X \sim \mathcal{N}_{(a,b)}(0, \sigma^2)$ have same tail behavior with Gaussian and Gaussian is Sub-Gaussian. $\square$

### L.1. Generalization of centered Sub-Gaussian results toward non-centered

We verify below that the results on centered sub-Gaussian distributions from Vershynin (2018) can be extended to non-centered sub-Gaussian distributions.

**Lemma L.2.** *Sum of non-centered Sub-Gaussian random variable is Sub-Gaussian.*

*Proof.* If the Orlicz 2 norm is bounded $||X||_{\psi_2} < \infty$, then X is Sub-Gaussian. Also, $||\mathbb{E}X||_{\psi_2} \leq C||X||_{\psi_2}$, and Sum of centered Sub-Gaussian random variable is Sub-Gaussian. We show $||\sum X_i||_{\psi_2} < \infty$, s.t. X is non-centered Sub-Gaussian.

$$
\begin{aligned}
||\sum X_i||_{\psi_2} &\leq ||\sum(X_i - \mathbb{E}X_i)||_{\psi_2} + ||\sum \mathbb{E}X_i||_{\psi_2} \\
&\leq ||\sum(X_i - \mathbb{E}X_i)||_{\psi_2} + \sum ||\mathbb{E}X_i||_{\psi_2} \\
&\leq ||\sum(X_i - \mathbb{E}X_i)||_{\psi_2} + C \sum ||X_i||_{\psi_2} < \infty
\end{aligned}
\tag{70}
$$

$\square$

**Lemma L.3.** *(Operator norm bound for non-centered Sub-Gaussian matrix, generalization of 4.4.5 in* Vershynin (2018)*) let* $A \in \mathbb{R}^{m \times n}$, $A[i][j]$ *is independent, non-centered Sub-Gaussian.* $\forall t > 0$,

$$
\begin{aligned}
||A|| &\leq CK(\sqrt{m} + \sqrt{n} + t) \text{ w.p. } 1 - \exp(-t^2) \\
\text{Alternatively, } ||A|| &\leq CK(\sqrt{m+n} + t) \text{ w.p. } 1 - \exp(-t^2)
\end{aligned}
\tag{71}
$$

$K = \max_{i,j} ||A[i][j]||_{\psi_2}$

**Lemma L.4.** *(Expectation of operator norm for non-centered Sub-Gaussian matrix generalization of 4.4.6 in* Vershynin (2018)*)*

$$
\begin{aligned}
\mathbb{E}||A|| &\leq CK(\sqrt{m} + \sqrt{n}) \\
\text{Alternatively, } \mathbb{E}||A|| &\leq CK(\sqrt{m+n}), \quad \text{and, } \mathbb{E}||A||^2 \leq C(m+n)
\end{aligned}
\tag{72}
$$

*Proof of Lemma L.3 and Lemma L.4.* Based on the result of Lemma L.2, one can follow the same proof process of Vershynin (2018)

$\square$

# M. Additional Results of Expectation of Hermite Polynomials

The non standard gaussian expectation of the product of two Hermite polynomials is computed as follows. It is an generalization of results of standard Gaussian distributions in O'Donnell (2021); Moniri et al. (2024) into a generalized multivariate Gaussian. We start with previously known facts, and derive our generalized results. These findings provide a useful analysis tool for Hermite polynomials, and may offer a foundation for broader applications in future works involving nonlinear activations decomposable into Hermite polynomials under the assumption of a multivariate Gaussian distribution.

## M.1. Expectation of a product of two Hermite polynomials

Here is the result of the expectation of the product of two Hermite polynomials, utilizing the orthogonality of Hermite polynomials.

**Lemma M.1** (Orthogonality of Hermite polynomials from Lemma C.1 Moniri et al. (2024))**.** *See also derivation in Chapter 11.2* O'Donnell (2021)*.*

*Let* $(Z_1, Z_2)$ *be jointly Gaussian with* $\mathbb{E}[Z_1] = \mathbb{E}[Z_2] = 0$, $\mathbb{E}[Z_1^2] = \mathbb{E}[Z_2^2] = 1$, *and* $\mathbb{E}[Z_1 Z_2] = \rho$. *Then for any* $k_1, k_2 \in \{0, 1, \cdots, \}$

$$
\mathbb{E}[H_{k_1}(Z_1) H_{k_2}(Z_2)] = k_1! \rho^{k_1} \mathbf{1}_{k_1 = k_2}
$$

*In the other form, for* $d \in \mathbb{N}$, $Z \sim \mathcal{N}(0, I_d)$, $a, b \in \mathbb{S}^{d-1}$,

$$
\mathbb{E}[H_{k_1}(Z^\top a) H_{k_2}(Z^\top b)] = k_1! (a^\top b)^{k_1} \mathbf{1}_{k_1 = k_2}
$$

*Fact* M.2. Let $W \in \mathbb{R}^{d \times N}$ s.t. $\forall i\ W[i] \in \mathbb{S}^{d-1}$. For $Z \sim \mathcal{N}(0, I)$,

$$
\mathbb{E}_{Z \sim \mathcal{N}(0,1)}[H_j(W^\top Z) H_k(W^\top Z)^\top] = k! (W^\top W)^{\circ j} \mathbf{1}_{j=k}
\tag{73}
$$

$$
\mathbb{E}_{Z \sim \mathcal{N}(0,1)}[H_j(W^\top Z)^\top H_k(W^\top Z)] = k! \sum ||W[i]||^{2j} \mathbf{1}_{j=k} = k! N \mathbf{1}_{j=k}
\tag{74}
$$

*Proof.* We apply $H_j$ element-wise. By Lemma M.1, we can acquire the above result. □

The following remark presents a modified condition of Lemma M.1 for the case where $a, b \notin \mathbb{S}^{d-1}$ in Lemma M.1. In this case, the variances of $Z^\top a$ and $Z^\top b$ are not equal to 1, and the covariance may exceed the bounds $[-1, 1]$. Under this condition, we will compute the expectation of the product of two Hermite polynomials as in Lemma M.1.

*Remark* M.3 (the modified condition of Lemma M.1). For $d \in \mathbb{N}$, $u, v \in \mathbb{R}^d$, $Z \sim \mathcal{N}(0, I_d)$,

$$Z_1 = \langle u, Z \rangle \sim \mathcal{N}(0, ||u||_2^2), Z_2 = \langle v, Z \rangle \sim \mathcal{N}(0, ||v||_2^2).$$

Then, $Z_1, Z_2$ is $\rho \stackrel{\triangle}{=} \langle \frac{u}{||u||}, \frac{v}{||v||} \rangle$ - correlated

$$
\begin{aligned}
corr(Z_1, Z_2) &= \frac{\mathbb{E}[Z_1 Z_2]}{\sqrt{V(Z_1)}\sqrt{V(Z_2)}} &= \frac{\mathbb{E}_Z \langle u, Z \rangle \langle v, Z \rangle}{||u|| \, ||v||} \\
&= \frac{\mathbb{E}_g \sum_i \sum_j u_i v_j Z_i Z_j}{||u|| \, ||v||} &= \frac{\sum_i \sum_j u_i v_j \mathbb{E}_Z[Z_i Z_j]}{||u|| \, ||v||} \\
&= \frac{\langle u, v \rangle}{||u|| \, ||v||}
\end{aligned}
\tag{75}
$$

Additionally,

$$
\begin{pmatrix} Z_1 \\ Z_2 \end{pmatrix} \sim n \left( \begin{pmatrix} 0 \\ 0 \end{pmatrix}, \begin{pmatrix} ||u||^2 & \langle u, v \rangle \\ \langle v, u \rangle & ||v||^2 \end{pmatrix} \right)
\tag{76}
$$

We first introduce Isserlis' theorem, which is essential for the proof. This theorem allows the expectation of the product of centered Gaussian random variables to be expressed as a product of covariances, making the computation feasible.

**Theorem M.4** (Isserlis' Theorem (Isserlis, 1918; Vignat, 2011)). *Let $X = (X_1, \cdots, X_d)$ Gaussian random vector s.t. $\mathbb{E}[X] = 0$, and let $A = \{\alpha_1, \cdots, \alpha_N\}$ be set of integers s.t. $1 \le \alpha_i \le d, \forall i$. Denote $X_A = \prod_{\alpha_i \in A} X_{\alpha_i}$, and $X_\emptyset = 1$. Let $\prod(A)$ denote partitions of $A$ into disjoint pairs and $\sigma \in \prod(A)$ is pair.*

$$
\mathbb{E}[X_A] = \sum_{\sigma \in \prod(A)} \prod_{(i,j) \in \sigma} \mathbb{E}[X_{\alpha_i} X_{\alpha_j}] \mathbf{1}_{d \text{ is even}}.
\tag{77}
$$

Now, we generalize the assumptions from the previous works so that Lemma M.1 holds for arbitrary vectors as Remark M.3. This could allow the weights of the networks to become analyzable when they go beyond the assumption of lying on the unit spheres.

**Theorem M.5** (Generalization of Lemma M.1 for centered Gaussian distribution). *For $d \in \mathbb{N}$, $u, v \in \mathbb{R}^d$, $g \sim \mathcal{N}(0, I_d)$, $\langle u, g \rangle \sim \mathcal{N}(0, ||u||_2^2)$, $\langle v, g \rangle \sim \mathcal{N}(0, ||v||_2^2)$.*

$$
\begin{aligned}
&\mathbb{E}_g[H_j(u^\top g) H_k(v^\top g)] \\
&= \frac{j! \langle u, v \rangle^j}{||u||^2 ||v||^2} \mathbf{1}_{j=k} - \frac{(||u||^2 - 1)(||v||^2 - 1)}{||u||^2 ||v||^2} \mathbb{E}_g[(v^\top g)^k (u^\top g)^j] \\
&\quad + \frac{(||v||^2 - 1)}{||v||^2} \mathbb{E}_g[H_j(u^\top g)(v^\top g)^k] + \frac{(||u||^2 - 1)}{||u||^2} \mathbb{E}_g[H_k(v^\top g)(u^\top g)^j]
\end{aligned}
\tag{78}
$$

*Remark* M.6. The same results can be derived as in Lemma M.1 when the variance is 1 in Thm. M.5.

*Proof of Theorem M.5.* (Generalize Chapter 11.2 O'Donnell (2021)'s derivation to non unit variance)

$\mathbb{E}_{z \sim n(0, \sigma^2)}[e^{tz}]$ **study**

First, we study about $\mathbb{E}_{g \sim n(0,\sigma^2)}[e^{tg}]$ in order to analysis non unit variance case.

$$\begin{aligned}
\mathbb{E}_{g \sim n(0,\sigma^2)}[e^{tg}] &= \frac{1}{\sqrt{2\pi}\sigma} \int e^{tg} e^{-\frac{g^2}{2\sigma^2}} \, dg \\
&= \frac{1}{\sqrt{2\pi}\sigma} e^{\frac{1}{2}t^2} \int \exp(-\frac{(g - \sigma^2 t)^2}{2\sigma^2}) \quad \text{complete square} \\
&= e^{\frac{1}{2}t^2}
\end{aligned} \tag{79}$$

$\mathbb{E}_{Z,Z'}[\exp(sZ + tZ')]$ **study**

Studying $\mathbb{E}_{Z,Z'}[\exp(sZ + tZ')]$, we can derive what we need to show.

$$\begin{aligned}
\mathbb{E}_{Z,Z'}[\exp(sZ + tZ')] &= \mathbb{E}_{g \sim n(0,I)}[\exp(s\langle u, g \rangle) + \exp(t\langle v, g \rangle)] \\
&= \prod_i \mathbb{E}_{g \sim n(0,1)}[\exp((su_i + tv_i)g_i)] \quad \text{Use equation 79} \\
&= \prod_i \exp(\frac{1}{2}(su_i + tv_i)^2) = \prod_i \exp(\frac{1}{2}s^2||u||^2 + \langle u, v \rangle st + \frac{1}{2}t^2||v||^2)
\end{aligned} \tag{80}$$

Therefore,

$$\exp(\langle u, v \rangle st) = \mathbb{E}_g[\exp(su^\top g - \frac{1}{2}s^2||u||^2)\exp(tv^\top g - \frac{1}{2}t^2||v||^2)].$$

*Fact* M.7. One can verify below propositions with simple calculations.
Let $P_j(z) + z^j = H_j(z)$, $C_u = ||u||^2 - 1$, $a > 0$.
Let $f(s) = \exp(sz - \frac{1}{2}s^2)$, $\bar{f}(s) = \exp(sz - \frac{1}{2}as^2)$, then

A. By Taylor expansion, $\exp(\langle u, v \rangle st) = \sum_{j=0}^{\infty} \frac{1}{j!}\langle u, v \rangle^j s^j t^j$.

B. By Taylor expansion, $\bar{f}(s) = \sum_{j=0}^{\infty} \frac{1}{j!}\bar{f}^{(n)}(0)s^j$

C. $\bar{f}^{(n)}(0) = H_n(z) + C_u P_n(z)$

By using the fact that $\exp(\langle u, v \rangle st) = \mathbb{E}_g[\exp(su^\top g - \frac{1}{2}s^2||u||^2)\exp(tv^\top g - \frac{1}{2}t^2||v||^2)]$, we can eliminate the different orders of $s\,t$ by a Taylor expansion and equating all monomials of the resulting polynomials.

$$\begin{aligned}
j!\langle u, v \rangle^j \mathbf{1}_{j=k} &= \mathbb{E}_g\Big[(H_j(u^\top g) + P_j(u^\top g)C_u)(H_j(v^\top g) + P_j(v^\top g)C_v)\Big] \\
&= \mathbb{E}_g\Big[(H_j(u^\top g) + (H_j(u^\top g) - (u^\top g)^j)C_u)(H_j(v^\top g) + (H_j(v^\top g) - (v^\top g)^j)C_v)\Big] \\
&= ||u||^2||v||^2\mathbb{E}_g[H_j(u^\top g)H_j(v^\top g)] + (||u||^2 - 1)(||v||^2 - 1)\mathbb{E}_g[(v^\top g)^j(u^\top g)^j] \\
&\quad - ||u||^2(||v||^2 - 1)\mathbb{E}_g[H_j(u^\top g)(v^\top g)^j] - ||v||^2(||u||^2 - 1)\mathbb{E}_g[H_j(v^\top g)(u^\top g)^j]
\end{aligned} \tag{81}$$

Therefore,

$$\begin{aligned}
&\mathbb{E}_g[H_j(u^\top g)H_j(v^\top g)] \\
&= \frac{j!\langle u, v \rangle^j}{||u||^2||v||^2}\mathbf{1}_{j=k} - \frac{(||u||^2 - 1)(||v||^2 - 1)}{||u||^2||v||^2}\mathbb{E}_g[(v^\top g)^j(u^\top g)^j] \\
&\quad + \frac{(||v||^2 - 1)}{||v||^2}\mathbb{E}_g[H_j(u^\top g)(v^\top g)^j] + \frac{(||u||^2 - 1)}{||u||^2}\mathbb{E}_g[H_j(v^\top g)(u^\top g)^j]
\end{aligned} \tag{82}$$

Note that the result of Lemma M.8 can be applied for concrete calculation, and conclude the proof. □

**Lemma M.8.** *For* $d \in \mathbb{N}$, $u, v \in \mathbb{R}^d$, $g \sim \mathcal{N}(0, I_d)$, $\bar{Z}_1 = \langle u, g \rangle$, $\bar{Z}_2 = \langle v, g \rangle$.

$$\begin{pmatrix} \bar{Z}_1 \\ \bar{Z}_2 \end{pmatrix} \sim n\left( \begin{pmatrix} 0 \\ 0 \end{pmatrix}, \begin{pmatrix} ||u||^2 & \langle u, v \rangle \\ \langle v, u \rangle & ||v||^2 \end{pmatrix} \right) \tag{83}$$

$X_{\alpha_i}$ *is defined at Thm.* M.4

$$
\mathbb{E}_{\bar{Z}_1,\bar{Z}_2}[H_j(\bar{Z}_1)\bar{Z}_2^k] = j! \sum_{m=0}^{\lfloor\frac{j}{2}\rfloor} \frac{(-1)^m}{m!(j-2m)!2^m} \sum_{\sigma \in \prod(\{\{\bar{Z}_1\}\times j-2m\}\cup\{\{\bar{Z}_2\}\times k\}\})} \prod_{(p,q)\in\sigma} \mathbb{E}[X_{\alpha_p}X_{\alpha_q}]\mathbf{1}_{\text{j+k-2m is even}}
\tag{84}
$$

$$
\mathbb{E}_{\bar{Z}_1,\bar{Z}_2}[\bar{Z}_1^j\bar{Z}_2^k] = \sum_{\sigma \in \prod(\{\{\bar{Z}_1\}\times j\}\cup\{\{\bar{Z}_2\}\times k\}\})} \prod_{(p,q)\in\sigma} \mathbb{E}[X_{\alpha_p}X_{\alpha_q}]\mathbf{1}_{\text{j+k is even}}
$$

*Proof.* By explicit formula of Hermite polynomials

$$
\mathbb{E}_{\bar{Z}_1,\bar{Z}_2}[H_j(\bar{Z}_1)(\bar{Z}_2)^k] = j! \sum_{m=0}^{\lfloor\frac{j}{2}\rfloor} \frac{(-1)^m}{m!(j-2m)!2^m} \mathbb{E}_{\bar{Z}_1,\bar{Z}_2}[\bar{Z}_1^{j-2m}\bar{Z}_2^k]
\tag{85}
$$

Therefore, we need to figure out $\mathbb{E}_{\bar{Z}_1,\bar{Z}_2}[\bar{Z}_1^p\bar{Z}_2^q]$. We know $\bar{Z}_1, \bar{Z}_2$ is mean zero Gaussian, so we can apply Thm. M.4 with $A = \{\{\bar{Z}_1\}\times p\}\cup\{\{\bar{Z}_2\}\times q\}\}, \mathbb{E}[\bar{Z}_1^p\bar{Z}_2^q] = \sum_{\sigma\in\prod(A)}\prod_{(\tau,\upsilon)\in\sigma}\mathbb{E}[X_{\alpha_\tau}X_{\alpha_\upsilon}].\mathbf{1}_{\text{p+q is even}}$

$\square$

**Corollary M.9** (Corollary of Lemma M.8). *Remark* $Z_1 \sim \mathcal{N}(0, \|u\|^2)$ *For the case* $k = 0$,

$$
\mathbb{E}_{\bar{Z}_1}[\bar{Z}_1^j] = \|u\|^j(j-1)!!\mathbf{1}_{\text{j is even}}
\tag{86}
$$

*Proof.*

$$
\mathbb{E}_{\bar{Z}_1,\bar{Z}_2}[\bar{Z}_1^j\bar{Z}_2^k] = \mathbb{E}_{\bar{Z}_1}[\bar{Z}_1^j] = \sum_{\sigma\in\prod(\{\bar{Z}_1\}\times j)} \prod_{(p,q)\in\sigma} \mathbb{E}[X_{\alpha_p}X_{\alpha_q}]\mathbf{1}_{\text{j is even}}
$$

$$
= \sum_{\sigma\in\prod(\{\bar{Z}_1\}\times j)} \prod_{(p,q)\in\sigma} \|u\|^2\mathbf{1}_{\text{j is even}} = \sum_{\sigma\in\prod(\{\bar{Z}_1\}\times j)} \|u\|^j\mathbf{1}_{\text{j is even}} = (j-1)!!\|u\|^j\mathbf{1}_{\text{j is even}}
\tag{87}
$$

$\square$

### M.2. Expectation of a product of two Hermite polynomials—Generalization toward non-centered Gaussian

We will change Theorem M.5 and Lemma M.8 to adopt a generalized Gaussian assumption with a mean of zero.

**Lemma M.10** (Taylor expansion of Hermite polynomials from Lemma C.2 Moniri et al. (2024)). *For any* $k_1, k_2 \in \{0, 1, \cdots, \}$ *and* $x, y \in \mathbb{R}$,

$$
H_k(x + y) = \sum_{j=0}^{k} \binom{k}{j} x^j H_{k-j}(y).
\tag{88}
$$

**Theorem M.11** (Generalization of Thm. M.5 for any Gaussian distribution). *For* $d \in \mathbb{N}$, $u, v \in \mathbb{R}^d$, $\xi \sim \mathcal{N}(0,1)$, $g \sim \mathcal{N}(\mu, \Sigma)$, $Z_1 = \langle u, g\rangle \sim \mathcal{N}(\mu^\top u, u^\top\Sigma u)$, $Z_2 = \langle v, g\rangle \sim \mathcal{N}(\mu^\top v, v^\top\Sigma v)$.

$$
\mathbb{E}_g[H_j(Z_1)H_k(Z_2)]
$$
$$
= \sum_{\alpha=0}^{j}\sum_{\beta=0}^{k} \binom{j}{\alpha}\binom{k}{\beta}(u^\top\mu)^\alpha(v^\top\mu)^\beta
$$
$$
\times \left[ \frac{(j-\alpha)!(u^\top\Sigma v)^{j-\alpha}}{u^\top\Sigma u v^\top\Sigma v}\mathbf{1}_{j-\alpha=k-\beta} - \frac{(u^\top\Sigma u - 1)(v^\top\Sigma v - 1)}{u^\top\Sigma u v^\top\Sigma v}\mathbb{E}_g[(\sqrt{u^\top\Sigma u}\xi)^{j-\alpha}(\sqrt{v^\top\Sigma v}\xi)^{k-\beta}] \right.
\tag{89}
$$
$$
\left. + \frac{(v^\top\Sigma v - 1)}{v^\top\Sigma v}\mathbb{E}_g[H_{j-\alpha}(\sqrt{u^\top\Sigma u}\xi)(\sqrt{v^\top\Sigma v}\xi)^{k-\beta}] + \frac{(u^\top\Sigma u - 1)}{u^\top\Sigma u}\mathbb{E}_g[(\sqrt{u^\top\Sigma u}\xi)^{j-\alpha}H_{k-\beta}(\sqrt{v^\top\Sigma v}\xi)] \right]
$$

*Proof of Theorem M.11.* By reparametrization i.e. $Z_1 = \sqrt{u^\top \Sigma u}\xi + u^\top \mu$, $Z_2 = \sqrt{v^\top \Sigma v}\xi + v^\top \mu$, and Lemma M.10,

$$H_j(\sqrt{u^\top \Sigma u}\xi + u^\top \mu) = \sum_{\alpha=0}^{j} \binom{j}{\alpha}(u^\top \mu)^\alpha H_{j-\alpha}(\sqrt{\mu^\top \Sigma u}\xi). \tag{90}$$

$$\begin{aligned} \mathbb{E}_g[H_j(u^\top g)H_k(v^\top g)] &= \mathbb{E}_\xi[H_j(\sqrt{u^\top \Sigma u}\xi + u^\top \mu)H_k(\sqrt{v^\top \Sigma v}\xi + v^\top \mu)] \\ &= \mathbb{E}_\xi\Big[ \sum_{\alpha=0}^{j} \binom{j}{\alpha}(u^\top \mu)^\alpha H_{j-\alpha}(\sqrt{\mu^\top \Sigma u}\xi)\Big]\Big[ \sum_{\beta=0}^{k} \binom{k}{\beta}(v^\top \mu)^\beta H_{k-\beta}(\sqrt{\mu^\top \Sigma v}\xi)\Big] \\ &= \sum_{\alpha=0}^{j} \sum_{\beta=0}^{k} \binom{j}{\alpha}\binom{k}{\beta}(u^\top \mu)^\alpha (v^\top \mu)^\beta \mathbb{E}_\xi[H_{j-\alpha}(\sqrt{\mu^\top \Sigma u}\xi)H_{k-\beta}(\sqrt{\mu^\top \Sigma v}\xi)] \end{aligned} \tag{91}$$

Use same proof technique Thm. M.5, with $\binom{\sqrt{u^\top \Sigma u}\xi}{\sqrt{v^\top \Sigma v}\xi} \sim n\left( \binom{0}{0}, \begin{pmatrix} u^\top \Sigma u & u^\top \Sigma v \\ v^\top \Sigma u & v^\top \Sigma v \end{pmatrix} \right)$

$$\begin{aligned} &\mathbb{E}_\xi[H_{j-\alpha}(\sqrt{u^\top \Sigma u}\xi)H_{k-\beta}(\sqrt{v^\top \Sigma v}\xi)] \\ &= \frac{(j-\alpha)!(u^\top \Sigma v)^{j-\alpha}}{u^\top \Sigma u v^\top \Sigma v}\mathbf{1}_{j-\alpha=k-\beta} - \frac{(u^\top \Sigma u - 1)(v^\top \Sigma v - 1)}{u^\top \Sigma u v^\top \Sigma v}\mathbb{E}_g[(\sqrt{u^\top \Sigma u}\xi)^{j-\alpha}(\sqrt{v^\top \Sigma v}\xi)^{k-\beta}] \\ &+ \frac{(v^\top \Sigma v - 1)}{v^\top \Sigma v}\mathbb{E}_g[H_{j-\alpha}(\sqrt{u^\top \Sigma u}\xi)(\sqrt{v^\top \Sigma v}\xi)^{k-\beta}] + \frac{(u^\top \Sigma u - 1)}{u^\top \Sigma u}\mathbb{E}_g[(\sqrt{u^\top \Sigma u}\xi)^{j-\alpha}H_{k-\beta}(\sqrt{v^\top \Sigma v}\xi)] \end{aligned} \tag{92}$$

In summary,

$$\begin{aligned} &\mathbb{E}_g[H_j(u^\top g)H_k(v^\top g)] \\ &= \sum_{\alpha=0}^{j} \sum_{\beta=0}^{k} \binom{j}{\alpha}\binom{k}{\beta}(u^\top \mu)^\alpha (v^\top \mu)^\beta \\ &\times \Bigg[ \frac{(j-\alpha)!(u^\top \Sigma v)^{j-\alpha}}{u^\top \Sigma u v^\top \Sigma v}\mathbf{1}_{j-\alpha=k-\beta} - \frac{(u^\top \Sigma u - 1)(v^\top \Sigma v - 1)}{u^\top \Sigma u v^\top \Sigma v}\mathbb{E}_\xi[(\sqrt{u^\top \Sigma u}\xi)^{j-\alpha}(\sqrt{v^\top \Sigma v}\xi)^{k-\beta}] \\ &+ \frac{(v^\top \Sigma v - 1)}{v^\top \Sigma v}\mathbb{E}_\xi[H_{j-\alpha}(\sqrt{u^\top \Sigma u}\xi)(\sqrt{v^\top \Sigma v}\xi)^{k-\beta}] + \frac{(u^\top \Sigma u - 1)}{u^\top \Sigma u}\mathbb{E}_\xi[(\sqrt{u^\top \Sigma u}\xi)^{j-\alpha}H_{k-\beta}(\sqrt{v^\top \Sigma v}\xi)]\Bigg] \end{aligned} \tag{93}$$

$\square$

The following Corollary which calculates the Expectation of the Power of a Gaussian Random Variable can be derived using the binomial expansion with the reparametrization technique and Corollary M.9. It corresponds to the case $k = 0$ in Lemma M.8.

**Corollary M.12** (Corollary of Lemma M.8). *Given $\omega \in \mathbb{R}^d$, let Gaussian Random Variable $Z \sim \mathcal{N}(\mu^\top \omega, \|\omega\|^2)$, then*

$$\begin{aligned} \mathbb{E}_Z(Z)^k &= \sum_{t=0}^{k} \binom{k}{t}(\mu^\top \omega)^{k-t}\mathbb{E}_{\bar{Z} \sim n(0,\|\omega\|^2)}[\bar{Z}^t] \\ &= \sum_{t=0}^{k} \binom{k}{t}(\mu^\top \omega)^{k-t}(t-1)!! \cdot \|\omega\|^t \mathbf{1}_{t\ is\ even}. \end{aligned} \tag{94}$$

The following corollary, which computes the Gaussian expectation of Hermite polynomials, is derived from the explicit form of Hermite polynomials and Corollary M.9. It corresponds to the case $k = 0$ in Theorem M.11.

**Corollary M.13.** *Given $\omega \in \mathbb{S}^{d-1}$, let Gaussian Random Variable $Z \sim \mathcal{N}(\mu^\top \omega, 1)$, then*

$$\mathbb{E}_x[H_k(\omega^\top x)] = \mathbb{E}_{\xi \sim \mathcal{n}(0,1)}[H_k(\omega^\top \mu + \xi)]$$

$$= \sum_{j=0}^{k} \binom{k}{j} (\omega^\top \mu)^{\circ j} E[H_k(\xi) H_0(\xi)] = (\omega^\top \mu)^k \tag{95}$$

# N. Information of ImageNet subset used in Experiments

Table 5: Configuration of Expr. V

|  | Vehicle | Bird | Product | Clothing |
|---|---|---|---|---|
| D | 98 | 100 | 11316 | 3985 |
| D+I(Sub) | 138 | 159 | 11568 | 4031 |
| D+I | 1098 | 1100 | 12316 | 4985 |

Table 6: Configuration of Expr. VI

|  | Step 0 | Step 1 | Step 2 | Step 3 |
|---|---|---|---|---|
| Vehicle | 25 | 50 | 75 | 98 |
| Bird | 25 | 50 | 75 | 100 |
| Product | 2829 | 5658 | 8487 | 11316 |
| Clothing | 996 | 1992 | 2989 | 3985 |

In this section, we present the criteria used to select classes for constructing the ImageNet subsets. We manually verified the label information to select the classes. The ImageNet subsets corresponding to the base fine-grained datasets were constructed as follows: I(V), I(B), I(P), and I(C), representing the Vehicle, Bird, Product, and Clothing subsets, respectively. These subsets consist of 59, 40, 353, and 46 classes, respectively. To balance the number of samples per class with those in the base fine-grained datasets, we extracted 82, 58, 5, and 6 samples per class for I(V), I(B), I(P), and I(C), respectively.

### N.1. I(V): The Vehicle classes chosen in ImageNet

Total 40 classes.

ambulance, cab, convertible, fire engine, forklift, freight car, garbage truck, go-kart, golfcart, half track, harvester, horse cart, jeep, jinrikisha, limousine, minibus, minivan, Model T, moped, motor scooter, mountain bike, moving van, oxcart, passenger car, pickup, police van, racer, recreational vehicle, school bus, snowmobile, snowplow, sports car, streetcar, tank, tow truck, tractor, trailer truck, tricycle, trolleybus, unicycle

### N.2. I(B): The bird classes chosen in ImageNet

Total 59 classes.

cock, hen, ostrich, brambling, goldfinch, house finch, junco, indigo bunting, robin, bulbul, jay, magpie, chickadee, water ouzel, bald eagle, vulture, great grey owl, black grouse, ptarmigan, ruffed grouse, prairie chicken, peacock, quail, partridge, African grey, macaw, sulphur-crested cockatoo, lorikeet, coucal, bee eater, hornbill, hummingbird, jacamar, toucan, drake, red-breasted merganser, goose, black swan, tusker, white stork, black stork, spoonbill, flamingo, little blue heron, American egret, bittern, crane, limpkin, European gallinule, American coot, bustard, ruddy turnstone, red-backed sandpiper, redshank, dowitcher, oystercatcher, pelican, king penguin, albatross

### N.3. I(P): The Product classes chosen in ImageNet

Total 353 classes.

abacus, accordion, acoustic guitar, altar, analog clock, apiary, ashcan, assault rifle, backpack, balance beam, balloon, ballpoint, Band Aid, banjo, barbell, barber chair, barometer, barrel, barrow, baseball, basketball, bassinet, bassoon, bathing cap, bath towel, bathtub, beach wagon, beacon, beaker, bearskin, beer bottle, beer glass, bell cote, bib, bicycle-built-for-two, binder, binoculars, bobsled, bolo tie, bonnet, bookcase, bottlecap, bow tie, brass, breakwater, broom, bucket, buckle, bulletproof vest, caldron, candle, cannon, canoe, can opener, car mirror, carousel, carpenter's kit, carton, car wheel, cash machine, cassette, cassette player, CD player, cello, cellular telephone, chain, chain saw, chest, chiffonier, chime, china cabinet, cleaver, clog, cocktail shaker, coffee mug, coffeepot, coil, combination lock, computer keyboard, confectionery, corkscrew, cornet, cradle, crash helmet, crate, crib, Crock Pot, croquet ball, crutch, dam, desk, desktop computer, dial telephone, digital clock, digital watch, dining table, dishrag, dishwasher, disk brake, dogsled, doormat, drum, drumstick, dumbbell, Dutch oven, electric fan, electric guitar, electric locomotive, envelope, espresso maker, face powder, feather boa,

file, fire screen, flagpole, flute, folding chair, football helmet, fountain pen, four-poster, French horn, frying pan, gasmask, gas pump, goblet, golf ball, gondola, gong, grand piano, grille, guillotine, hair slide, hair spray, hammer, hamper, hand blower, hand-held computer, handkerchief, hard disc, harmonica, harp, hatchet, holster, honeycomb, hook, horizontal bar, hourglass, iPod, iron, jack-o'-lantern, jigsaw puzzle, joystick, knot, ladle, lampshade, laptop, lawn mower, lens cap, letter opener, lighter, lipstick, lotion, loudspeaker, loupe, magnetic compass, mailbox, maraca, marimba, matchstick, maypole, measuring cup, medicine chest, microphone, microwave, milk can, mixing bowl, modem, monitor, mountain tent, mousetrap, muzzle, nail, neck brace, necklace, nipple, notebook, oboe, ocarina, odometer, oil filter, organ, oscilloscope, oxygen mask, packet, paddle, paddlewheel, padlock, paintbrush, paper towel, parachute, parallel bars, park bench, parking meter, pay-phone, pedestal, pencil box, pencil sharpener, perfume, Petri dish, photocopier, pick, picket fence, piggy bank, pill bottle, pillow, ping-pong ball, plastic bag, plate rack, plow, plunger, Polaroid camera, pole, pool table, pop bottle, pot, potter's wheel, power drill, prayer rug, printer, prison, projectile, projector, puck, punching bag, purse, quill, quilt, racket, radiator, radio, radio telescope, rain barrel, reel, reflex camera, refrigerator, remote control, revolver, rifle, rocking chair, rotisserie, rubber eraser, rugby ball, rule, safe, safety pin, saltshaker, sax, scabbard, scale, scoreboard, screen, screw, screwdriver, seat belt, sewing machine, shield, shopping basket, shopping cart, shovel, shower cap, shower curtain, ski, sleeping bag, sliding door, slot, snorkel, soap dispenser, soccer ball, sock, solar dish, soup bowl, space bar, space heater, spatula, spider web, spindle, spotlight, steel drum, stethoscope, stole, stopwatch, stove, strainer, stretcher, studio couch, sunscreen, swab, switch, syringe, table lamp, tape player, teapot, teddy, television, tennis ball, theater curtain, thimble, thresher, throne, tile roof, toaster, tobacco shop, toilet seat, torch, totem pole, tray, tripod, trombone, tub, turnstile, typewriter keyboard, umbrella, vacuum, vase, vault, velvet, vending machine, violin, volleyball, waffle iron, wall clock, wallet, wardrobe, washbasin, washer, water bottle, water jug, water tower, whiskey jug, whistle, window screen, window shade, wine bottle, wing, wok, wooden spoon, comic book, crossword puzzle, street sign, traffic light, book jacket, menu, plate

**N.4. I(C): The Clothing classes chosen in ImageNet**

Total 46 classes.

abaya, academic gown, apron, bikini, brassiere, breastplate, cardigan, chain mail, Christmas stocking, cloak, cowboy boot, cowboy hat, cuirass, diaper, fur coat, gown, hoopskirt, jean, jersey, kimono, knee pad, lab coat, Loafer, mailbag, mask, military uniform, miniskirt, mitten, overskirt, pajama, poncho, running shoe, sandal, sarong, ski mask, sombrero, suit, sunglass, sunglasses, sweatshirt, swimming trunks, trench coat, vestment, wig, Windsor tie, wool

# O. Rotation Matrix Generation Process of *Setup 2*

To generate a set of rotation matrices with diverse magnitudes of rotation, we constructed an algorithm that samples $k = 300$ random matrices, each formed by adding i.i.d. Gaussian noise matrix of varying variance to the identity matrix $I$. The process ensures the generation of rotation matrices with varying extents of rotation, from slight to more substantial deviations from the identity matrix.

The rotation matrices are generated as follows:

1. A matrix is initialized as $I + \epsilon \cdot M$, where $M$ is a i.i.d. standard random Gaussian matrix.

2. Using the QR decomposition, we orthogonalize this matrix to ensure it forms a valid rotation matrix.

3. Finally, if the determinant of the resulting matrix is negative, we flip the sign of the first column to maintain a determinant of $+1$, ensuring it is a valid rotation.

In summary, this method provides a collection of matrices that progressively deviate from $I$, allowing us to observe and sample rotations of increasing magnitude. Please refer Algorithm 3

---

**Algorithm 3** Gaussian-Sampled Random Rotation Matrix Generation

---

**Input:** Number of dimensions $n$, number of matrices $k$
**Output:** Stack of random rotation matrices
Initialize empty list $\mathcal{Q}$
Set $\epsilon \leftarrow 0.5$
**for** $i \leftarrow 0$ **to** $k - 1$ **do**
  **if** $i \mod \left(\frac{k}{16}\right) = 0$ and $i \neq 0$ **then**
    $\epsilon \leftarrow \epsilon \times 0.22360679775$
  **end if**
  Generate random matrix $M$: $M \sim \mathcal{N}(0, 1)^{n \times n}$
  Compute perturbed matrix: $A \leftarrow I_n + \epsilon \times M$
  Compute QR decomposition: $Q, R \leftarrow \text{QR}(A)$
  **if** $\det(Q) < 0$ **then**
    Flip first column of $Q$: $Q[:, 0] \leftarrow -Q[:, 0]$
  **end if**
  Add $Q$ to $\mathcal{Q}$
**end for**
**return** $\mathcal{Q}$

---

