# OpenReview forum: "How Classifiers Extract General Features for Downstream Tasks: An Asymptotic Analysis in Two-Layer Models"
_ICML.cc/2025/Conference — Submitted to ICML 2025_

### Official Review · Reviewer_RnQR · 2025-03-01

**Overall Recommendation:** 2

**Summary:**

The paper investigates how classifiers learn general features that can be directly applied to new tasks without further training. It considers a two-layer neural network trained with a single gradient descent step on a mean‐squared error loss. In an asymptotic regime—where the number of samples, input dimension, and network width all grow proportionally—the authors decompose the learned feature representation into the initial random features and a “spike” signal term introduced by training. They show that when the distribution of unseen classes is similar to that of the training data, the extracted features exhibit strong intra-class cohesion and inter-class separability. A key finding is that if two unseen classes both align with the same training class, separability decreases even when overall similarity is high.

**Claims And Evidence:**

The claims are supported by simplified theoretical analysis and several empirical validations.

**Essential References Not Discussed:**

This is my primary concern regarding this paper. There already exists extensive theoretical literature on feature learning, encompassing both linear and nonlinear features, and covering various regimes such as the concentration and proportional regimes. Although I am most familiar with regression settings, classification scenarios should inherently share similar insights, making results transferable. Additionally, earlier theoretical work on classification likely exists, and many papers have also addressed transfer learning extensively. I would appreciate a thorough comparison with the following references: nonlinear feature learning and transfer learning in regression cases (https://arxiv.org/pdf/2311.13774, https://arxiv.org/pdf/2411.17201), linear feature learning and transfer learning in regression (https://arxiv.org/abs/2206.15144), and linear feature learning in the proportional regime using random matrix theory (https://arxiv.org/abs/2410.18938). Moreover, there should be additional relevant references along these lines. Given this context, I am not entirely sure about the novelty of your paper.

**Experimental Designs Or Analyses:**

Yes, see above.

**Methods And Evaluation Criteria:**

The analysis is based on an idealized model—a two-layer network undergoing a single, large gradient descent update—selected for analytical convenience. Very similar analyses should already appear in previous studies, and I will elaborate on this point below. The evaluation includes synthetic experiments designed to control similarity measures, along with empirical tests on standard image datasets. A nearest-neighbor retrieval metric is employed to assess clustering quality, serving as a reasonable proxy for evaluating feature transferability.

**Other Comments Or Suggestions:**

Some notation choices are unusual. For example, in Theorem 3.3, the notation "spike_L" is non-standard and should be avoided in a formal theorem statement.

**Other Strengths And Weaknesses:**

No.

**Questions For Authors:**

No.

**Relation To Broader Scientific Literature:**

This paper should be related to the feature learning literature in understanding deep learning.

**Theoretical Claims:**

Yes. The techniques are standard.

---

> ### Author Rebuttal · Authors · 2025-03-31
>
> Attachment link: anonymous.4open.science/r/icmlrebuttal-3B6F
>
> Thank you for recognizing the theory as standard and for positively evaluating the experiments. We understand that you wanted us to **clarify the relationship between the relevant studies and ours** in order to highlight the **novelty**.
>
> Our work focuses on the **feature transfer of networks trained via classifiers**, particularly in settings akin to metric learning, which has previously been used for performance *without understanding the underlying principles.* In contrast to many previous studies that analyze on feature learning phenomenon of training data or test errors.
>
> For theoretical contributions, we agree as you mentioned, “classification scenarios should inherently share similar insights, making results transferable”. However, this adoption is challenging due to difficulties in dealing with multiple non-identical and arbitrarily labeled class conditional distributions. Such modifications on the problem setup require the more generalized assumption on data distribution, i.e., **a non-centered Sub-Gaussian**. Thus we provide mathematical tools to analyze non-centered sub-Gaussian distribution with an Hermite expandable activation, which are novel in the existing works (Section L, M). These technical contributions generalize the previous data assumption, and we believe future research can build on ours.
>
> **First, we reviewed the requested studies** (Wang23, Fu24, Damian22, Dandi24) and noted them by the first author and year.
>
> *Note on Damian 22, Wang 23, Fu 24*
>
> These studies analyze the sample complexity for neural networks to learn the internal structure while regressing a teacher in the form of $g(x^\top Ax)$ where this form becomes progressively more complex (here, $x$ comes from a centered distribution.). These studies focus on regression problems using a teacher, whereas we assume only **arbitrarily assigned classification labels without using a teacher**. At the same time, the transfer learning part analyzes the sample complexity for learning when function head $g$ is changed while maintaining the internal structure. In contrast, we conduct research on feature transfer when new distribution inputs are introduced without additional learning.
>
> *Note on Dandi24*
>
> Dandi24 analyzed the learned feature extractor using an equivalent model to characterize the test error for *regression* and analyze the spectral tail behavior of the covariance matrix of the feature extractor. This study dealt with the phenomenon where spikes appear in the spectrum, making the tail heavy-tailed. Similarly, we derive an equivalent model suited to our *classification* setup and analyze the characteristics of clustering error for unseen distributions (Section 4.1). Then, we analyze how the spike term of the feature extractor operates (Section 4.2).
>
> Additionally, we have provided a comparison table in Table 2 of the above anonymous link, which we plan to include in the Appendix to offer further explanation to the readers. We sincerely appreciate the references you shared.
>
> **Secondly, regarding the existing research on classification and transfer learning**, we respond as follows
>
> We’ve already covered existing studies on feature transferability in the Additional Related Works section, which focuses on intuitive explanations of feature transfer (L662) and feature learning phenomena, Neural Collapse, mainly for training data (L672). Since you reviewed only the mathematical part of the Supplementary Material, we suggest reviewing Section A.
>
> Some additional classification studies were not cited as they either **differ from our framework or focus mainly on training data**. However, we plan to discuss them in our paper for readers' understanding.
>
> For example, there are papers addressing the alignment phenomenon between the network and the training data in neural network classification tasks (arxiv.org/abs/2307.12851), studies on the increased separability and cohesion of training data features in classification settings (arxiv.org/abs/2012.10424, arxiv.org/abs/1909.06930), and research showing that classifier networks outperform linear classifiers through feature learning (arxiv.org/abs/2206.01717, arxiv.org/abs/2202.07626, arxiv.org/abs/2102.11742).
>
> Additionally, there are empirical studies on classifier *transfer learning* (not *feature transfer* like ours, which doesn’t require learning) (arxiv.org/abs/2212.12206) and theoretical studies (arxiv.org/abs/1809.10374, arxiv.org/abs/2006.11650).
>
> Furthermore, there are theoretical papers studying optimization properties like implicit margin maximization (arxiv.org/abs/2110.13905, arxiv.org/abs/2305.11788) and interpolation (arxiv.org/abs/2012.02409, arxiv.org/abs/2306.09955).
>
> **Finally**, we will standardize spike_L to s_L.
>
> Once again, we truly appreciate your thoughtful review and constructive suggestions, and if you have any additional concerns, let us know. We will do our best to address your concerns.

---

### Official Review · Reviewer_Y7aR · 2025-03-03

**Overall Recommendation:** 1

**Summary:**

This paper studies how a two-layer classifier, trained by mean-squared error for multi-class problems, learns a features that can cluster unseen data. The main theoretical result is an exact characterization of a single-step gradient update of the network features, derived under proportional asymptotics (sample size, width, and data dimension diverging at constant rate) and mixture of sub-Gaussian covariate distribution. More precisely, they show that after a single gradient step, the feature matrix can be asymptotically approximated by a low-rank spiked matrix. From this result, the authors draw the following conclusions:

- **Multi-class spikes**: the feature matrix depends mostly on a linear combination of classifier weights aligned with data, implying that “off-angle” or unrelated training directions have negligible effect.
- **Cohesion and separability**: By looking at the population risk for binary-class unseen data, this characterization implies that train-unseen similarity drives intra-class cohesion and inter-class separability – but separability degrades if new classes map to the same training label.

Numerical experiments on both real and synthetic data are provided to illustrate the theoretical results.

**Claims And Evidence:**

Most of the claims in this work are mathematical statements, which are supported by rigorous proofs in the appendix.

**Essential References Not Discussed:**

While some works in this literature as acknowledged, I have found some relevant omissions - most importantly [Dandi et al., 2024], which proves results which are complementary to (Damien et al., 2022; Ba et al., 2022) and precedes (Moniri et al., 2024).

Also relevant to the regime studied here are (Cui et al., 2024; Dandi et al., 2024), who provided an exact characterization of the gradient step in the critical learning rate regime which is complementary to (Moniri et al., 2024) results, that hold in the sub-critical regime.

Finally, a related recent work is (Demir & Dogan 2024), who generalized this discussion to mixture distributions, in a similar spirit to this work. The authors should provide a comparison of their technical contributions with this work, since it is possible there are some technical overlap.


- [Dandi et al., 2024] Dandi, Yatin, Florent Krzakala, Bruno Loureiro, Luca Pesce, and Ludovic Stephan. "How two-layer neural networks learn, one (giant) step at a time." JMLR 2024.
- [Cui et al., 2024] Hugo Cui, Luca Pesce, Yatin Dandi, Florent Krzakala, Yue Lu, Lenka Zdeborova, Bruno Loureiro. "Asymptotics of feature learning in two-layer networks after one gradient-step." ICML 2024.
- [Dandi et al., 2024] Dandi, Yatin, Luca Pesce, Hugo Cui, Florent Krzakala, Yue M. Lu, and Bruno Loureiro. "A random matrix theory perspective on the spectrum of learned features and asymptotic generalization capabilities." arXiv preprint arXiv:2410.18938 (2024).
- [Demir & Dogan 2024] Demir, S., & Dogan, Z. "Asymptotic Analysis of Two-Layer Neural Networks after One Gradient Step under Gaussian Mixtures Data with Structure." ICLR 2024.

**Experimental Designs Or Analyses:**

Several numerical experiments are presented, which seem mostly to agree with the theory. I did not check them in detail, but I think this is not the main point of the paper.

**Methods And Evaluation Criteria:**

N/A.

**Other Comments Or Suggestions:**

- The scales in Fig. 2 and 4 are very big. It would be better to plot these quantities with the correct normalization with respect to the dimensions $d,N,n$ to have more meaningful numbers.
- Although you do it implicitly, would be nice to give the explicit definition of the matrix $\mathbb{A}$ in eq. (3), which is relevant to the result that follows.
- It is not easy to keep track of which quantities here are matrices/vectors or components of matrices/vectors. I suggest the authors to stress this difference, for instance by putting the former in bold.
- There is an inconsistency in the usage of bold and not bold letters throughout the manuscript, for instance in the dimensions $N,p,d$.
- There are many sentences which seem incomplete in the text. For example, In the end of page 2: "The number of problem \#_{P} \overset{\Delta}{=} ...$. To improve readability, I would encourage the authors to go through the text and complete these.
- Page 2, L090, left-column: "sementic"
- I don't understand the limit $\sum_{i<j}^{c}$ in Eq. (1) - is this correct? Is $c=N/n$?

**Other Strengths And Weaknesses:**

The motivation of this work and the phenomenology are very interesting, and this could have been a potentially nice paper if it was more clearly written.

However, the heavy (and sometimes redundant) notation, combined with many typos and confusing writing makes the reading quite challenging, even for a reader familiar to this line of work. The manuscript would definitively benefit from an extensive rewriting. Below I give a few concrete suggestions.

**Questions For Authors:**

My main concerns were listed in the previous points.

However, I have one clarification for the authors:
- If I understood correctly, the learning rate in the gradient step is exactly $1$? Accounting for the different choices of normalization, how does this compare with the scale of the learning rate of (Ba et al., 2022; Dandi et al., 2024; Moniri et al., 2024)? More precisely: is this choice sub-critical (as in Moniri et al., 2024) or critical (as in Cui et al., 2024)?

**Relation To Broader Scientific Literature:**

This paper belongs to a recent wave in the machine learning theory literature in looking at the benefits of feature learning after a few (or here, a single) large gradient step from initialization (Damien et al., 2022; Ba et al., 2022; Dandi et al., 2024). The key result, which is the asymptotic characterization of the network feature matrix after a single step, heavily builds on the analysis developed in these works, adapting it to the case of an input data mixture distribution.

**Theoretical Claims:**

I skimmed through the proofs in the appendix. The key steps are mostly adapting previous results by (Ba et al., 2022; Dandi et al., 2024; Moniri et al., 2024) to the multi-class classification setting studied here.

Honestly, the lack of text highlighting the main ideas make it challenging to parse the proof, even for a reader familiar with the works cited above. Therefore, even after skimming through it I am not in a position to saying the proofs are correct, and I would strongly encourage the authors to rewrite the appendix having the readability in mind.

---

> ### Author Rebuttal · Authors · 2025-03-31
>
> Attachment link: anonymous.4open.science/r/icmlrebuttal-3B6F
>
> Thank you for your comments. We also appreciate your feedback that the motivation of our work and the phenomenology are very interesting.
>
> After reviewing your comments, we found that you suggested improvements for readability, understood the paper's core focus as theoretical, requested a discussion on its relation to certain references, and faced difficulties in comprehending the proof section.
>
> Through this rebuttal, we aim to 1. inform you of the revisions made based on your suggestions, 2. clarify that our contributions beyond theory, 3. discuss the gap between our work and the suggested references, 4. assist in your understanding of our theory, and 5,6. rebuttal to some misunderstandings
>
> 1. We made the following revisions based on your feedback.
>
> First, for readability, we fix typos you suggested. Second, for better clarity of theory, we included the main idea before the proof and the proof structure as in the above link's Table 3-8 and Fig. 4–8. Also, we placed the proofs of main Theorems and Propositions upfront, while moving the lemmas back. Third, we included references to studies that you suggested.
>
> 2. Contrary to comment on experiments that “I think this is not the main point of the paper.”, **the results in Section 4 and experiments are the major contributions of our study**.
>
> This work offers intuition based on theory to data collection, with experiments on settings not explored in metric learning literature. It contributes by leveraging existing theories with empirical adaption, **enhancing the understanding of feature transfer, and establish widely accepted intuitions (as in *L662*) on a rigorous foundation.** We hope this is reflected in your evaluation.
>
> 3. As stated in the right column on L84, **the research we present is not a direct extension of the feature learning literatures including your suggestions**.
>
> The studies you mentioned focus on *teacher-student regression*. On the other hand, our research analyzes feature transfer and clustering tasks in a *classification setups*. Therefore, we primarily cited baselines that are required for the proof. However, since the distribution we are dealing with is non-centered Sub-Gaussian, which generalizes to all data setups in your suggestions. Additionally, you can find the comparisons between the papers you mentioned and our approach in Table 2 of the attachment link.
>
> 4. It seems you missed the logical flow of the proof, so we summarize and clarify its implications as follows.
>
> Lemmas (I.xx, J.xx) resemble those in Ba et al. (2022) and Moniri et al. (2024) which deal regression, but our classifier setting **need to deal with multiple non-identical distributions**. This requires novel proof techniques in Section I, J and a novel generalized distributions be dealt within Sections L and M.
> There, **we develop mathematical tools for analyzing sub-Gaussian distributions and Hermite-expandable activations**, extending previous data assumptions. We trust this enhances your comprehension of our contributions.
>
> 5. We address sub-Gaussian distributions rather than Gaussian mixture distributions, which is more general distribution set.
>
> **We did not use the word "mixture" in the text. Thus, could you clarify why you mentioned this in the Summary and in referencing Scientific Literature?** After reviewing Demir & Dogan 2024, we found that it is contemporary work. They assume a Gaussian mixture for the data, propose a Conditional Gaussian Equivalence and stochastically approximate a 2-layer network. This might seems similar to our Theorem 3.3, but **our proof does not require conditional approximates nor constructing a stochastically equivalent model** since we only utilize Sub-Gaussian property. This provides an advantage in analyzing the features of new data in a simple form.
>
> 6. Regarding the rejection, we understand that it was due to a lack of clarity in the proof of the Theorem.
>
> We would like to appreciate if you clarify which part was difficult to understand, leading to your statement, "I am not in a position to say the proofs are correct"?
> **Your concrete suggestions mainly focused on typographical errors**, the distinction between vectors and matrices, etc. **We do not believe these significantly hinder the understanding.** Also, We appreciate your understanding in novel analysis induce unfamiliar notations, unavoidably.
>
> **Answer for Question**:
>
> The learning rate is exactly 1. Normalization follows Moniri et al., 2024. The learning rate corresponds to the case where $\alpha = 0$ in Moniri et al., 2024 or $\Theta(\sqrt{n})$ in Cui et al., 2024 In this setup, we make $||G||, ||W_0||, ||W|| = O_p(1)$. We will add the assumption that $\eta = \Theta(1)$ for clarity.
>
> Finally, the constant $c$ in Eq. 1 is #_cls. We will revise this.
>
> Looking forward to your response and discussion. Thank you.

---

> > ### Comment · Reviewer_Y7aR · 2025-04-04
> >
> > I thank the authors for clarifying some points in their rebuttal and for the updates.
> >
> > To clarify, my score is not based on a single factor (e.g. clarity) but on a combination of factors, which in my judgement suggests the paper would benefit from substantial rewritting justifying a resubmission. Coming to the specific points of your rebuttal.
> >
> > > the results in Section 4 and experiments are the major contributions of our study.
> >
> > I find rather striking that you highlight the experiments as the major contribution of your work.
> >
> > First, this does not translate in the writting: both in the abstract and introduction, the experiments are refered as a "*validation*" of the theoretical findings, and not the opposite.
> >
> > Second, I honestly find the plots you present in Section 5 illisible. There are too many plots, with small fonts and curves that do not convey a clear message. I don't think it is normal I have to zoom in to understand them. Moreover, in most of the plots the y-axis is a quantity that is scales badly with the problem dimensions (and in some plots also the x-axis). As a constructive suggestion, I would suggest you to focus on less plots that convey stronger and clearer conclusions.
> >
> > > We did not use the word "mixture" in the text.
> >
> > Maybe I misunderstood something. A mixture distribution $p(x)$ is a distribution of the form:
> > $$
> > p(x) = \sum\limits_{c\in\mathcal{C}} \alpha_{c} p_{c}(x)
> > $$
> > for a countable set $\mathcal{C}$, a sequence of $\alpha_{c}\in[0,1]$ with $\sum_{c\in\mathcal{C}}\alpha_{c}=1$ and $p_{c}(x)$ a family of distributions indexed by $c\in\mathcal{C}$. Can you clarify how Assumption 2.2 is different from a mixture of subGaussian distributions?
> >
> > > We would like to appreciate if you clarify which part was difficult to understand, leading to your statement, "I am not in a position to say the proofs are correct"?
> >
> > The way the theoretical part of the appendix is written is hard to parse. For concretness, consider for instance "*Appendix I. Proof of Theorem 3.1*". The first Lemma is a bunch of symbolic mathematical statements with no text. Assumptions are not stated. You talk about "aforementinoed" matrices $\mathbb{A},\mathbb{B},\mathbb{C}$, but where have you defined them?
> >
> > The rest of the Appendix follow pretty much a similar structure: a bunch of mathematical formulas, almost no text and context and with the assumptions almost always implicit in the statements of the mathematical results.

---

> > > ### Author Response · Authors · 2025-04-08
> > >
> > > Thank you for staying engaged in the discussion. Following the initial rebuttal, we consider issues raised by R1, R3, R4, R5 (rebuttal index), and Question1 to be resolved, as no further objections were made.
> > > In this round, although described as a combined basis for rejection, we believe some have been clarified in revision without rewriting, while others are addressed through rebuttals:
> > >
> > > 1. Sub-Gaussian vs. mixture: The question seems to conflate basic distributional concepts, whereas **this assumption serves as a central enabler** of our contribution.
> > > 2. Revisit R2: We clarified a few sentences to more explicitly emphasize that Sections 4 and 5 are key contributions, as **already stated throughout the paper**.
> > > 3. Section 5 figure clarity: **Now newly raised; We respectfully contest**: the figures are sufficiently **clear and minimal** for supporting our claims.
> > > 4. Revisit R6: We note that **this focuses on style rather than the correctness** of the theory. We have revised the text accordingly for clarity.
> > >
> > > **First, Sub-Gaussian vs. mixture**: The question on Assumption 2.2 seems to reflect a misunderstanding of its role. We assume class-conditional distributions are non-centered sub-Gaussian (defined via tail behavior; Vershynin, 2018), enabling generalization over prior works and forming the theoretical basis of our deterministic feature analysis for classifier in Sections 3 and 4. Since mixtures of sub-Gaussian remain sub-Gaussian due to the tail, mentioning “mixture” is redundant. We suspect the confusion stems from “class-conditional,” but clarify that unconditioned distributions (possibly representated as mixture) are not the object of analysis.
> > >
> > > **Second**, while you comment that the role of Sections 4 and 5 as key contributions is not clearly conveyed, we note that this point is emphasized consistently throughout the paper, where we repeatedly highlight the practical perspectives:
> > >
> > > - L30 (Abstract): “demonstrate practical applicability”
> > > - L46 (Intro): Motivated from the underexplored nature of transferable conditions
> > > - L408 (Conclusion): Offers empirical insights and implications for transfer tasks
> > >
> > > Nonetheless, we revised a few sentences to clarify that “validate” (L87, L107) refers to theory-driven practical explanation, not just validation of approximation.
> > >
> > > **Third**, the opinion that figures in Section 5 are illegible may not be appropriate:
> > >
> > > - Number of plots: The number of plots is justified by the need to support each claim with a minimal set—one summary figure per experiment, mostly.
> > > - Need to Zoom in: We disagree with your concern about figure size. The current scale is already sufficient—for instance, in Figure 7, the text is roughly 3/4 the size of the caption, which is not atypical. Moreover, the key message (trend direction) remains clearly readable without zooming.
> > > - Scale: This issue appears only in Figures 7 and 8 due to unnormalized high-dimensional features. Normalizing would complicate the setup explanation, so we respectfully disagree.
> > >
> > > **Also, we note that UifN carefully reviewed experimental results and provided constructive feedback, and neither RnQR nor UifN raised concerns about legibility.**
> > >
> > > **Fourth**, the renewed concern about the readability of the theoretical section appears to be a matter of presentation style—including (a) Formula-centric presentation, (b) implicit definitions, and (c) implicit assumptions/contexts—rather than a substantive issue. We note that no specific errors were raised or suspected. With RnQR’s positive evaluation, we claim that our theory is structurally and formally sound. **Minor improvements such as restating important assumptions or adding brief context have made in revision, but we do not believe these issues warrant a full rewrite.**
> > >
> > > Responses to specific points:
> > >
> > > - The remark on “bunch of symbolic math with no text” reflects a stylistic preference e.g. Lemma I.1. This is acceptable in theoretical papers; see, e.g., Lemma 14/15 of Ba et al. (2022) for a format.
> > > - The notations A,B,C were already explicitized in response to earlier feedback. As you reviewed the theory closely, we hope Eq. 13, 20, and 30 were not overlooked.
> > > - We explicitly state that our main results rely on Assumptions 2.1/2.2 and Condition 4.4, so we did not repeat them in every lemma. Additionally, whenever we use external results, we already ensured citation for context (e.g., L2243, L2255).
> > >
> > > Moreover, we believe this work is timely for publication with few revisions. The references you and RnQR provided (Table 2 in Attachment) suggest that the field is entering a mature phase, with recent studies extending to more theoretical generalizations. *Our work goes beyond commonly proposed complex assumptions and shows that this line of research can extend to practical tasks like classification and feature transfer.* It demonstrates how theory can inform real-world problems, and we believe sharing this idea to the community now can **notably stimulate future progress.**

---

### Official Review · Reviewer_UifN · 2025-03-14

**Overall Recommendation:** 4

**Summary:**

This paper explores how classifiers extract general features for transfer to new distributions. It analyzes a two - layer network in the proportional regime, decomposing features into components like random initialization and spikes related to training classes. In binary classification, train - unseen similarity affects cohesion and separability. Higher similarity increases cohesion, and for separability, it depends on class assignment. In multi-class classification, non-orthogonal spikes to the input contribute to feature extraction. Experiments on synthetic and real datasets demonstrate the theoretical findings, showing that semantic similarity between training and test data improves clustering performance.

**Claims And Evidence:**

Yes.

**Essential References Not Discussed:**

No.

**Experimental Designs Or Analyses:**

Yes. Experimental design in Expr.V seems a little unfair. Accoding to the supp.N, the number of related images between sub In1k and subsampled whole In1k is different, it is hard to get the conclusion in L411.

**Methods And Evaluation Criteria:**

Yes. Althogh the paper starts the anaylze from a two-layer network, it consider the network width and dataset size in a similar scale, it aligns with common practices in model scaling. Experimental results on Resnet50 also show consistent results with the analyzing.

**Other Comments Or Suggestions:**

None.

**Other Strengths And Weaknesses:**

Strengths:
1. Offers a rigorous theoretical analysis of feature learning in two-layer neural networks using Hermite decomposition.
2. Uses both synthetic and real-world datasets to demonstrate the effectiveness.
3. Provides insights into feature transferability.

Weaknesses:
1. Unfair comparsion. Experimental design in Expr.V seems a little unfair. Accoding to the supp.N, the number of related images between sub In1k and subsampled whole In1k is different, it is hard to get the conclusion in L411.
2. The experimental setup on ImgNet can not well support their claims. This paper uses four semantic categories in ImgNet to show ""adding semantically relevant classes to the training set leads to performance gains", however, these semantic categories have different classes and have different internal similarity. In other words, these semantic categories have different diversity. For example, the semantic similarity between cock and hen (birds) is surely closer to that between guitar and analog clock. I guess a better way is to establish four semantic categories from four finegrained datasets. Or, consider the internal similarity when conducting the training set.
3. Organization problem of the paper. Some important experimental setup (which showcase the fair comparison) should be included in the main text to avoid readers from turning pages repeatedly.

**Questions For Authors:**

Please refer to weaknesses.

**Relation To Broader Scientific Literature:**

The paper provide a theoretical analysis method for understanding the transfer learning problem in deep learning, which is beneficial for downstream applications.

**Theoretical Claims:**

No.

---

> ### Author Rebuttal · Authors · 2025-03-31
>
> Attachment link: anonymous.4open.science/r/icmlrebuttal-3B6F
>
> Your thoughtful feedback is a great encouragement and reaffirms our commitment to furthering this research.
> We understand that your primary concern lies in fairness during experimental validation.
>
> Thus, the primary purpose of this rebuttal is **to dispel any misunderstandings that this setup is unfair**,and to clarify that fairness has indeed been ensured. Moreover, *in response to your point about the difficulty of verifying fairness* as **W3**, we included **the experimental setup mentioned at Appendix N in the main text.**
>
> After carefully reviewing your comments, we respond to your concerns and propose the following revisions:
>
> **First, We will address the concern on W1 about the unfairness of the claim on L411**, i.e., Expr V. It seems to stem from the following sentences.
>
> _L355_: we performed experiments on the whole classes ImageNet by sampling $100$ instances per class (say subsampled whole In1k).
>
> _L411_: We find that adding classes from the entire ImageNet dataset during training, rather than including only related classes,  does not significantly improve clustering
>
> _Appendix N_: To balance the number of samples per class with those in the base fine-grained datasets, we extracted $82$, $58$, $5$, and $6$ samples per class for I(V), I(B), I(P), and I(C), respectively.
>
> We sincerely apologize for the omission of some information in _L355_. To be more precise, in _L355_, we only specified the setup for Expr. VII, which used the subsampled whole In1k. In this case, as _L355_, we extracted $100$ instances per class from the entire ImageNet dataset. Since this experiment is related to the case of duplicate assignment and we use subsampled whole In1k alone, the situation you were concerned about did not occur.
> **On the other hand**, for Expr. V, we did not perform sampling in the same way. We already agree with your concerns, and we took them into account when designing the experiment. In Expr. V, to ensure fairness, we extracted $82$, $58$, $5$, and $6$ instances per class from the I+D case datasets and performed the experiment accordingly. We sincerely apologize for the omission of this detail when explaining the setup.
>
> To ensure the accuracy of this experiment, we _re-inspect_ the code for extracting $82$, $58$, $5$, and $6$ instances per class for the D+I case using subsampled whole In1k in Expr. V (Listing 1 in the above anonymous link) and then _conducted the experiment again_. As a result, we obtained nearly identical performance to the original results (with only minor performance variations due to seed differences, which had no impact on the claims). This can be confirmed in Figure 1 and Table 2  in the above anonymous link.
>
> **Secondly, you raised a concern on W2 that the ImageNet setup may not sufficiently support the claim that "adding semantically relevant classes to the training set leads to performance gains"** However, there is a misunderstanding regarding this point.
> We presented this claim in the _L90_ and left column of _L414_, and the experiment that supports this claim is Expr. VI. We designed Expr. VI based on the same reasoning as yours, which is why **we do not make the above claim by adding similar classes using a subset of ImageNet.**
>
> Consequently, as stated in _L430_ and as you suggested, in Expr. VI, we conducted experiments using $25$%, $50$%, $75$%, and $100$% of the domain dataset classes exclusively i.e. the four fine-grained datasets. As a result, we observed a trend where adding classes within a dataset led to performance improvements, which formed the basis of our claim.
> However, we have realized that this claim was not explicitly stated in the explanation of Expr. VI. To reduce any potential misunderstandings, we will explicitly add the statement "adding semantically relevant classes to the training set leads to performance gains" in _L429_, based on Expr. VI. Additionally, we will clarify that the domain datasets used are fine-grained datasets such as CUB, CAR, SOP, and ISC.
>
> I hope this answer will help you understand this study better and please continue to rate it positively.
>
> Thank you.

---

### Decision · Program_Chairs · 2025-05-01

**Decision:**

Reject

**Comment:**

This paper addresses an interesting problem: the theoretical and empirical understanding of feature transfer in classifier-trained neural networks. While the topic is timely and relevant to ongoing discussions in feature learning and transfer, I share the reviewers’ concerns regarding both the clarity of exposition and the novelty of the contributions. The theoretical development overlaps heavily with prior work, and the differences are not sufficiently articulated or justified in the text. Additionally, the paper’s organization and presentation, especially in the theoretical sections, make it difficult to parse, even for readers familiar with the literature.

The experimental section, while intended to support the theoretical findings, suffers from issues of clarity and scale, limiting its effectiveness in communicating the main observations. Multiple reviewers found the plots difficult to interpret and the conclusions not convincingly drawn. Overall, I agree with the reviewers that the current form of the submission does not meet the standards or maturity required for acceptance. I encourage the authors to revise the manuscript substantially to improve clarity, better position their contributions within the existing literature, and strengthen the empirical presentation.